# Solving Random Quadratic Systems of Equations Is Nearly as Easy as Solving Linear Systems

**Yuxin Chen**
Department of Statistics
Stanford University
Stanford, CA 94305
yxchen@stanfor.edu

**Emmanuel J. Candès**
Department of Mathematics and Department of Statistics
Stanford University
Stanford, CA 94305
candes@stanford.edu

## Abstract

This paper is concerned with finding a solution $x$ to a quadratic system of equations $y_i = |\langle a_i, x \rangle|^2$, $i = 1, \ldots, m$. We demonstrate that it is possible to solve unstructured random quadratic systems in $n$ variables exactly from $\mathcal{O}(n)$ equations in linear time, that is, in time proportional to reading the data $\{a_i\}$ and $\{y_i\}$. This is accomplished by a novel procedure, which starting from an initial guess given by a spectral initialization procedure, attempts to minimize a nonconvex objective. The proposed algorithm distinguishes from prior approaches by regularizing the initialization and descent procedures in an adaptive fashion, which discard terms bearing too much influence on the initial estimate or search directions. These careful selection rules—which effectively serve as a variance reduction scheme—provide a tighter initial guess, more robust descent directions, and thus enhanced practical performance. Further, this procedure also achieves a near-optimal statistical accuracy in the presence of noise. Empirically, we demonstrate that the computational cost of our algorithm is about four times that of solving a least-squares problem of the same size.

## 1 Introduction

Suppose we are given a response vector $y = [y_i]_{1 \le i \le m}$ generated from a quadratic transformation of an unknown object $x \in \mathbb{R}^n / \mathbb{C}^n$, i.e.

$$y_i = |\langle a_i, x \rangle|^2, \qquad i = 1, \cdots, m, \tag{1}$$

where the feature/design vectors $a_i \in \mathbb{R}^n / \mathbb{C}^n$ are known. In other words, we acquire measurements about the linear product $\langle a_i, x \rangle$ with all signs/phases missing. Can we hope to recover $x$ from this nonlinear system of equations?

This problem can be recast as a quadratically constrained quadratic program (QCQP), which subsumes as special cases various classical combinatorial problems with Boolean variables (e.g. the NP-complete stone problem [1, Section 3.4.1]). In the physical sciences, this problem is commonly referred to as *phase retrieval* [2]; the origin is that in many imaging applications (e.g. X-ray crystallography, diffraction imaging, microscopy) it is infeasible to record the phases of the diffraction patterns so that we can only record $|Ax|^2$, where $x$ is the electrical field of interest. Moreover, this problem finds applications in estimating the *mixture of linear regression*, since one can transform the latent membership variables into missing phases [3]. Despite its importance across various fields, solving the quadratic system (1) is combinatorial in nature and, in general, NP complete.

To be more realistic albeit more challenging, the acquired samples are almost always corrupted by some amount of noise, namely,

$$y_i \approx |\langle a_i, x \rangle|^2, \qquad i = 1, \cdots, m. \tag{2}$$

For instance, in imaging applications the data are best modeled by Poisson random variables

$$y_i \overset{\text{ind.}}{\sim} \mathsf{Poisson}\big( |\langle \boldsymbol{a}_i, \boldsymbol{x} \rangle|^2 \big), \qquad i = 1, \cdots, m, \tag{3}$$

which captures the variation in the number of photons detected by a sensor. While we shall pay special attention to the Poisson noise model due to its practical relevance, the current work aims to accommodate general—or even deterministic—noise structures.

## 1.1 Nonconvex optimization

Assuming independent samples, the first attempt is to seek the maximum likelihood estimate (MLE):

$$\text{minimize}_{\boldsymbol{z}} \quad -\sum_{i=1}^{m} \ell\left(\boldsymbol{z}; y_i\right), \tag{4}$$

where $\ell\left(\boldsymbol{z}; y_i\right)$ represents the log-likelihood of a candidate $\boldsymbol{z}$ given the outcome $y_i$. As an example, under the Poisson data model (3), one has (up to some constant offset)

$$\ell(\boldsymbol{z}; y_i) = y_i \log(|\boldsymbol{a}_i^* \boldsymbol{z}|^2) - |\boldsymbol{a}_i^* \boldsymbol{z}|^2. \tag{5}$$

Computing the MLE, however, is in general intractable, since $\ell(\boldsymbol{z}; y_i)$ is not concave in $\boldsymbol{z}$.

Fortunately, under unstructured random systems, the problem is not as ill-posed as it might seem, and is solvable via convenient convex programs with optimal statistical guarantees [4–12]. The basic paradigm is to lift the quadratically constrained problem into a linearly constrained problem by introducing a matrix variable $\boldsymbol{X} = \boldsymbol{x}\boldsymbol{x}^*$ and relaxing the rank-one constraint. Nevertheless, working with the auxiliary matrix variable significantly increases the computational complexity, which exceeds the order of $n^3$ and is prohibitively expensive for large-scale data.

This paper follows a different route, which attempts recovery by minimizing the nonconvex objective (4) or (5) directly (e.g. [2, 13–19]). The main incentive is the potential computational benefit, since this strategy operates directly upon vectors instead of lifting decision variables to higher dimension. Among this class of procedures, one natural candidate is the family of gradient-descent type algorithms developed with respect to the objective (4). This paradigm can be regarded as performing some variant of stochastic gradient descent over the random samples $\{(y_i, \boldsymbol{a}_i)\}_{1 \leq i \leq m}$ as an approximation to maximize the population likelihood $L(\boldsymbol{z}) := \mathbb{E}_{(y,\boldsymbol{a})}[\ell(\boldsymbol{z}; y)]$. While in general nonconvex optimization falls short of performance guarantees, a recently proposed approach called *Wirtinger Flow* (WF) [13] promises efficiency under random features. In a nutshell, WF initializes the iterate via a spectral method, and then successively refines the estimate via the following update rule:

$$\boldsymbol{z}^{(t+1)} = \boldsymbol{z}^{(t)} + \frac{\mu_t}{m} \sum_{i=1}^{m} \nabla \ell(\boldsymbol{z}^{(t)}; y_i),$$

where $\boldsymbol{z}^{(t)}$ denotes the $t$th iterate of the algorithm, and $\mu_t$ is the learning rate. Here, $\nabla \ell(\boldsymbol{z}; y_i)$ represents the Wirtinger derivative with respect to $\boldsymbol{z}$, which reduces to the ordinary gradient in the real setting. Under Gaussian designs, WF (i) allows exact recovery from $\mathcal{O}(n \log n)$ noise-free quadratic equations [13];[1] (ii) recovers $\boldsymbol{x}$ up to $\epsilon$-accuracy within $\mathcal{O}(mn^2 \log 1/\epsilon)$ time (or flops) [13]; and (iii) is stable and converges to the MLE under Gaussian noise [20]. Despite these intriguing guarantees, the computational complexity of WF still far exceeds the best that one can hope for. Moreover, its sample complexity is a logarithmic factor away from the information-theoretic limit.

## 1.2 This paper: Truncated Wirtinger Flow

This paper develops a novel linear-time algorithm, called *Truncated Wirtinger Flow* (TWF), that achieves a near-optimal statistical accuracy. The distinguishing features include a careful initialization procedure and a more adaptive gradient flow. Informally, TWF entails two stages:

1. **Initialization:** compute an initial guess $\boldsymbol{z}^{(0)}$ by means of a spectral method applied to a subset $\mathcal{T}_0$ of data $\{y_i\}$ that do not bear too much influence on the spectral estimates;

2. **Loop:** for $0 \leq t < T$,

$$\boldsymbol{z}^{(t+1)} = \boldsymbol{z}^{(t)} + \frac{\mu_t}{m} \sum_{i \in \mathcal{T}_{t+1}} \nabla \ell(\boldsymbol{z}^{(t)}; y_i) \tag{6}$$

for some index set $\mathcal{T}_{t+1} \subseteq \{1, \cdots, m\}$ over which $\nabla \ell(\boldsymbol{z}^{(t)}; y_i)$ are well-controlled.

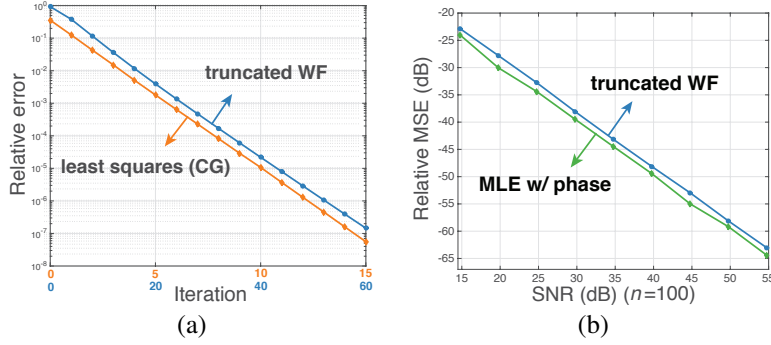

Figure 1: (a) Relative errors of CG and TWF vs. iteration count, where $n = 1000$ and $m = 8n$. (b) Relative MSE vs. SNR in dB, where $n = 100$. The curves are shown for two settings: TWF for solving quadratic equations (blue), and MLE had we observed additional phase information (green).

We highlight three aspects of the proposed algorithm, with details deferred to Section 2.

(a) In contrast to WF and other gradient descent variants, we regularize both the initialization and the gradient flow in a more cautious manner by operating only upon some iteration-varying index sets $\mathcal{T}_t$. The main point is that enforcing such careful selection rules lead to tighter initialization and more robust descent directions.

(b) TWF sets the learning rate $\mu_t$ in a far more liberal fashion (e.g. $\mu_t \equiv 0.2$ under suitable conditions), as opposed to the situation in WF that recommends $\mu_t = \mathcal{O}(1/n)$.

(c) Computationally, each iterative step mainly consists in calculating $\{\nabla\ell(\boldsymbol{z}; y_i)\}$, which is inexpensive and can often be performed in linear time, that is, in time proportional to evaluating the data and the constraints. Take the real-valued Poisson likelihood (5) for example:

$$\nabla\ell(\boldsymbol{z}; y_i) = 2\left\{\frac{y_i}{|\boldsymbol{a}_i^\top \boldsymbol{z}|^2}\boldsymbol{a}_i\boldsymbol{a}_i^\top \boldsymbol{z} - \boldsymbol{a}_i\boldsymbol{a}_i^\top \boldsymbol{z}\right\} = 2\left(\frac{y_i - |\boldsymbol{a}_i^\top \boldsymbol{z}|^2}{\boldsymbol{a}_i^\top \boldsymbol{z}}\right)\boldsymbol{a}_i, \quad 1 \le i \le m,$$

which essentially amounts to two matrix-vector products. To see this, rewrite

$$\sum_{i \in \mathcal{T}_{t+1}} \nabla\ell(\boldsymbol{z}^{(t)}; y_i) = \boldsymbol{A}^\top \boldsymbol{v}, \qquad v_i = \begin{cases} 2\frac{y_i - |\boldsymbol{a}_i^\top \boldsymbol{z}^{(t)}|^2}{\boldsymbol{a}_i^\top \boldsymbol{z}^{(t)}}, & i \in \mathcal{T}_{t+1}, \\ 0, & \text{otherwise}, \end{cases}$$

where $\boldsymbol{A} := [\boldsymbol{a}_1, \cdots, \boldsymbol{a}_m]^\top$. Hence, $\boldsymbol{A}\boldsymbol{z}^{(t)}$ gives $\boldsymbol{v}$ and $\boldsymbol{A}^\top \boldsymbol{v}$ the desired truncated gradient.

## 1.3 Numerical surprises

The power of TWF is best illustrated by numerical examples. Since $\boldsymbol{x}$ and $e^{-j\phi}\boldsymbol{x}$ are indistinguishable given $\boldsymbol{y}$, we evaluate the solution based on a metric that disregards the global phase [13]:

$$\text{dist}(\boldsymbol{z}, \boldsymbol{x}) := \min_{\varphi \in [0, 2\pi)} \|e^{-j\varphi}\boldsymbol{z} - \boldsymbol{x}\|. \tag{7}$$

In the sequel, TWF operates according to the Poisson log-likelihood (5), and takes $\mu_t \equiv 0.2$.

We first compare the computational efficiency of TWF for solving quadratic systems with that of conjugate gradient (CG) for solving least square problems. As is well known, CG is among the most popular methods for solving large-scale least square problems, and hence offers a desired benchmark. We run TWF and CG respectively over the following two problems:

(a)  find $\boldsymbol{x} \in \mathbb{R}^n$      s.t. $b_i = \boldsymbol{a}_i^\top \boldsymbol{x},$      $1 \le i \le m,$

(b)  find $\boldsymbol{x} \in \mathbb{R}^n$      s.t. $b_i = |\boldsymbol{a}_i^\top \boldsymbol{x}|,$      $1 \le i \le m,$

where $m = 8n$, $\boldsymbol{x} \sim \mathcal{N}(\boldsymbol{0}, \boldsymbol{I})$, and $\boldsymbol{a}_i \stackrel{\text{ind.}}{\sim} \mathcal{N}(\boldsymbol{0}, \boldsymbol{I})$. This yields a well-conditioned design matrix $\boldsymbol{A}$, for which CG converges extremely fast [21]. The relative estimation errors of both methods are reported in Fig. 1(a), where TWF is seeded by 10 power iterations. The iteration counts are plotted in different scales so that 4 TWF iterations are tantamount to 1 CG iteration. Since each iteration of CG and TWF involves two matrix vector products $\boldsymbol{A}\boldsymbol{z}$ and $\boldsymbol{A}^\top \boldsymbol{v}$, the numerical plots lead to a suprisingly positive observation for such an unstructured design:

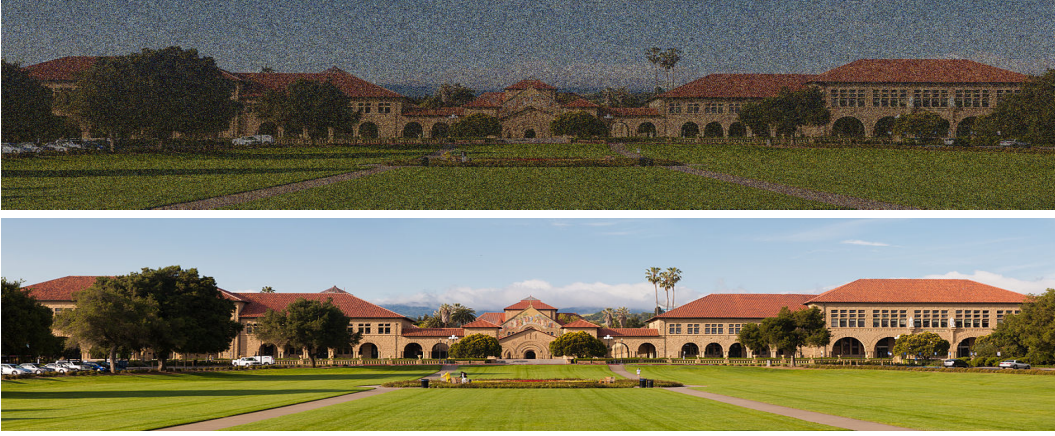

Figure 2: Recovery after (top) truncated spectral initialization, and (bottom) 50 TWF iterations.

*Even when all phase information is missing, TWF is capable of solving a quadratic system of equations only about 4 times[2] slower than solving a least squares problem of the same size!*

The numerical surprise extends to noisy quadratic systems. Under the Poisson data model, Fig. 1(b) displays the relative mean-square error (MSE) of TWF when the signal-to-noise ratio (SNR) varies; here, the relative MSE and the SNR are defined as[3]

$$\text{MSE} := \text{dist}^2(\hat{\boldsymbol{x}}, \boldsymbol{x}) \,/\, \|\boldsymbol{x}\|^2 \qquad \text{and} \qquad \text{SNR} := 3\|\boldsymbol{x}\|^2, \tag{8}$$

where $\hat{\boldsymbol{x}}$ is an estimate. Both SNR and MSE are displayed on a dB scale (i.e. the values of $10\log_{10}(\text{SNR})$ and $10\log_{10}(\text{MSE})$ are plotted). To evaluate the quality of the TWF solution, we compare it with the MLE applied to an *ideal* problem where the phases (i.e. $\{\varphi_i = \text{sign}(\boldsymbol{a}_i^\top \boldsymbol{x})\}$) are revealed *a priori*. The presence of this precious side information gives away the phase retrieval problem and allows us to compute the MLE via convex programming. As illustrated in Fig. 1(b), TWF solves the quadratic system with nearly the best possible accuracy, since it only incurs an extra 1.5 dB loss compared to the ideal MLE with all true phases revealed.

To demonstrate the scalability of TWF on real data, we apply TWF on a $320 \times 1280$ image. Consider a type of physically realizable measurements called coded diffraction patterns (CDP) [22], where

$$\boldsymbol{y}^{(l)} = |\boldsymbol{F}\boldsymbol{D}^{(l)}\boldsymbol{x}|^2, \qquad 1 \le l \le L, \tag{9}$$

where $m = nL$, $|\boldsymbol{z}|^2$ denotes the vector of entrywise squared magnitudes, and $\boldsymbol{F}$ is the DFT matrix. Here, $\boldsymbol{D}^{(l)}$ is a diagonal matrix whose diagonal entries are randomly drawn from $\{1, -1, j, -j\}$, which models signal modulation before diffraction. We generate $L = 12$ masks for measurements, and run TWF on a MacBook Pro with a 3 GHz Intel Core i7. We run 50 truncated power iterations and 50 TWF iterations, which in total cost 43.9 seconds for each color channel. The relative errors after initialization and TWF iterations are 0.4773 and $2.2 \times 10^{-5}$, respectively; see Fig. 2.

## 1.4 Main results

We corroborate the preceding numerical findings with theoretical support. For concreteness, we assume TWF proceeds according to the Poisson log-likelihood (5). We suppose the samples $(y_i, \boldsymbol{a}_i)$ are independently and randomly drawn from the population, and model the random features $\boldsymbol{a}_i$ as

$$\boldsymbol{a}_i \sim \mathcal{N}\left(\boldsymbol{0}, \boldsymbol{I}_n\right). \tag{10}$$

To start with, the following theorem confirms the performance of TWF under noiseless data.

**Theorem 1** (**Exact recovery**). *Consider the noiseless case (1) with an arbitrary $\boldsymbol{x} \in \mathbb{R}^n$. Suppose that the learning rate $\mu_t$ is either taken to be a constant $\mu_t \equiv \mu > 0$ or chosen via a backtracking line search. Then there exist some constants $0 < \rho, \nu < 1$ and $\mu_0, c_0, c_1, c_2 > 0$ such that with probability exceeding $1 - c_1 \exp(-c_2 m)$, the TWF estimates (Algorithm 1) obey*

$$\text{dist}(\boldsymbol{z}^{(t)}, \boldsymbol{x}) \leq \nu(1 - \rho)^t \|\boldsymbol{x}\|, \quad \forall t \in \mathbb{N}, \tag{11}$$

*provided that $m \geq c_0 n$ and $\mu \leq \mu_0$. As discussed below, we can take $\mu_0 \approx 0.3$.*

Theorem 1 justifies two intriguing properties of TWF. To begin with, TWF recovers the ground truth exactly as soon as the number of equations is on the same order of the number of unknowns, which is information theoretically optimal. More surprisingly, TWF converges at a geometric rate, i.e. it achieves $\epsilon$-accuracy (i.e. $\text{dist}(\boldsymbol{z}^{(t)}, \boldsymbol{x}) \leq \epsilon \|\boldsymbol{x}\|$) within at most $\mathcal{O}(\log 1/\epsilon)$ iterations. As a result, the time taken for TWF to solve the quadratic systems is proportional to the time taken to read the data, which confirms the linear-time complexity of TWF. These outperform the theoretical guarantees of WF [13], which requires $\mathcal{O}(mn^2 \log 1/\epsilon)$ runtime and $\mathcal{O}(n \log n)$ sample complexity.

Notably, the performance gain of TWF is the result of the key algorithmic changes. Rather than maximizing the data usage at each step, TWF exploits the samples at hand in a more selective manner, which effectively trims away those components that are too influential on either the initial guess or the search directions, thus reducing the volatility of each movement. With a tighter initial guess and better-controlled search directions in place, TWF is able to proceed with a more aggressive learning rate. Taken collectively these efforts enable the appealing convergence property of TWF.

Next, we turn to more realistic noisy data by accounting for a general additive noise model:

$$y_i = |\langle \boldsymbol{a}_i, \boldsymbol{x} \rangle|^2 + \eta_i, \qquad 1 \leq i \leq m, \tag{12}$$

where $\eta_i$ represents a noise term. The stability of TWF is demonstrated in the theorem below.

**Theorem 2** (**Stability**). *Consider the noisy case (12). Suppose that the learning rate $\mu_t$ is either taken to be a positive constant $\mu_t \equiv \mu$ or chosen via a backtracking line search. If*

$$m \geq c_0 n, \quad \mu \leq \mu_0, \quad and \quad \|\boldsymbol{\eta}\|_\infty \leq c_1 \|\boldsymbol{x}\|^2, \tag{13}$$

*then with probability at least $1 - c_2 \exp(-c_3 m)$, the TWF estimates (Algorithm 1) satisfy*

$$\text{dist}(\boldsymbol{z}^{(t)}, \boldsymbol{x}) \lesssim \frac{\|\boldsymbol{\eta}\|}{\sqrt{m}\|\boldsymbol{x}\|} + (1 - \rho)^t \|\boldsymbol{x}\|, \quad \forall t \in \mathbb{N} \tag{14}$$

*for all $\boldsymbol{x} \in \mathbb{R}^n$. Here, $0 < \rho < 1$ and $\mu_0, c_0, c_1, c_2, c_3 > 0$ are some universal constants.*

Alternatively, if one regards the SNR for the model (12) as follows

$$\text{SNR} := \left( \sum_{i=1}^{m} |\langle \boldsymbol{a}_i, \boldsymbol{x} \rangle|^4 \right) / \|\boldsymbol{\eta}\|^2 \approx 3m\|\boldsymbol{x}\|^4 / \|\boldsymbol{\eta}\|^2, \tag{15}$$

then we immediately arrive at another form of performance guarantee stated in terms of SNR:

$$\text{dist}(\boldsymbol{z}^{(t)}, \boldsymbol{x}) \lesssim \frac{1}{\sqrt{\text{SNR}}}\|\boldsymbol{x}\| + (1 - \rho)^t \|\boldsymbol{x}\|, \quad \forall t \in \mathbb{N}. \tag{16}$$

As a consequence, the relative error of TWF reaches $\mathcal{O}(\text{SNR}^{-1/2})$ within a logarithmic number of iterations. It is worth emphasizing that the above stability guarantee is deterministic, which holds for any noise structure obeying (13). Encouragingly, this statistical accuracy is nearly un-improvable, as revealed by a minimax lower bound that we provide in the supplemental materials.

We pause to remark that several other nonconvex methods have been proposed for solving quadratic equations, which exhibit intriguing empirical performances. A partial list includes the error reduction schemes by Fienup [2], alternating minimization [14], Kaczmarz method [17], and generalized approximate message passing [15]. However, most of them fall short of theoretical support. The analytical difficulty arises since these methods employ the same samples in each iteration, which introduces complicated dependencies across all iterates. To circumvent this issue, [14] proposes a sample-splitting version of the alternating minimization method that employs fresh samples in each iteration. Despite the mathematical convenience, the sample complexity of this approach is $\mathcal{O}(n \log^3 n + n \log^2 n \log 1/\epsilon)$, which is a factor of $\mathcal{O}(\log^3 n)$ from optimal and is empirically largely outperformed by the variant that reuses all samples. In contrast, our algorithm uses the same pool of samples all the time and is therefore practically appealing. Besides, the approach in [14] does not come with provable stability guarantees. Numerically, each iteration of Fienup's algorithm (or alternating minimization) involves solving a least squares problem, and the algorithm converges in tens or hundreds of iterations. This is computationally more expensive than TWF, whose computational complexity is merely about 4 times that of solving a least squares problem.

## 2 Algorithm: Truncated Wirtinger Flow

This section explains the basic principles of truncated Wirtinger flow. For notational convenience, we denote $A := [a_1, \cdots, a_m]^\top$ and $\mathcal{A}(M) := \{a_i^\top M a_i\}_{1 \le i \le m}$ for any $M \in \mathbb{R}^{n \times n}$.

### 2.1 Truncated gradient stage

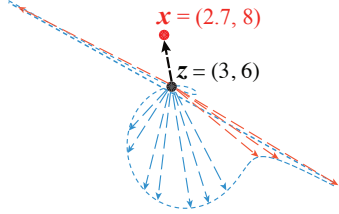

Figure 3: The locus of $-\frac{1}{2}\nabla \ell_i(z)$ for all unit vectors $a_i$. The red arrows depict those directions with large weights.

In the case of independent *real-valued* data, the descent direction of the WF updates—which is the gradient of the Poisson log-likelihood—can be expressed as follows:

$$\sum_{i=1}^m \nabla \ell(z; y_i) = \sum_{i=1}^m 2 \underbrace{\frac{y_i - |a_i^\top z|^2}{a_i^\top z}}_{:=\nu_i} a_i, \qquad (17)$$

where $\nu_i$ represents the weight assigned to each feature $a_i$.

Unfortunately, the gradient of this form is non-integrable and hence uncontrollable. To see this, consider any fixed $z \in \mathbb{R}^n$. The typical value of $\min_{1 \le i \le m} |a_i^\top z|$ is on the order of $\frac{1}{m}\|z\|$, leading to some excessively large weights $\nu_i$. Notably, an underlying premise for a nonconvex procedure to succeed is to ensure all iterates reside within *a basin of attraction*, that is, a neighborhood surrounding $x$ within which $x$ is the unique stationary point of the objective. When a gradient is unreasonably large, the iterative step might overshoot and end up leaving this basin of attraction. Consequently, WF moving along the preceding direction might not come close to the truth unless $z$ is already very close to $x$. This is observed in numerical simulations[4].

TWF addresses this challenge by discarding terms having too high of a leverage on the search direction; this is achieved by regularizing the weights $\nu_i$ via appropriate truncation. Specifically,

$$z^{(t+1)} = z^{(t)} + \frac{\mu_t}{m} \nabla \ell_{\mathrm{tr}}(z^{(t)}), \quad \forall t \in \mathbb{N}, \qquad (18)$$

where $\nabla \ell_{\mathrm{tr}}(\cdot)$ denotes the *truncated* gradient given by

$$\nabla \ell_{\mathrm{tr}}(z) := \sum_{i=1}^m 2 \frac{y_i - |a_i^\top z|^2}{a_i^\top z} a_i \mathbf{1}_{\mathcal{E}_1^i(z) \cap \mathcal{E}_2^i(z)} \qquad (19)$$

for some appropriate truncation criteria specified by $\mathcal{E}_1^i(\cdot)$ and $\mathcal{E}_2^i(\cdot)$. In our algorithm, we take $\mathcal{E}_1^i(z)$ and $\mathcal{E}_2^i(z)$ to be two collections of events given by

$$\mathcal{E}_1^i(z) := \left\{ \alpha_z^{\mathrm{lb}} \|z\| \le |a_i^\top z| \le \alpha_z^{\mathrm{ub}} \|z\| \right\}; \qquad (20)$$

$$\mathcal{E}_2^i(z) := \left\{ |y_i - |a_i^\top z|^2| \le \frac{\alpha_h}{m} \|y - \mathcal{A}(zz^\top)\|_1 \frac{|a_i^\top z|}{\|z\|} \right\}, \qquad (21)$$

where $\alpha_z^{\mathrm{lb}}$, $\alpha_z^{\mathrm{ub}}$, $\alpha_z$ are predetermined truncation thresholds. In words, we drop components whose size fall outside some confidence range—a range where the magnitudes of both the numerator and denominator of $\nu_i$ are comparable to their respective mean values.

This paradigm could be counter-intuitive at first glance, since one might expect the larger terms to be better aligned with the desired search direction. The issue, however, is that the large terms are extremely volatile and could dominate all other components in an undesired way. In contrast, TWF makes use of only gradient components of typical sizes, which slightly increases the bias but remarkably reduces the variance of the descent direction. We expect such gradient regularization and variance reduction schemes to be beneficial for solving a broad family of nonconvex problems.

### 2.2 Truncated spectral initialization

A key step to ensure meaningful convergence is to seed TWF with some point inside the basin of attraction, which proves crucial for other nonconvex procedures as well. An appealing initialization

**Algorithm 1** Truncated Wirtinger Flow.

---

**Input**: Measurements $\{y_i \mid 1 \leq i \leq m\}$ and feature vectors $\{\boldsymbol{a}_i \mid 1 \leq i \leq m\}$; truncation thresholds $\alpha_z^{\text{lb}}, \alpha_z^{\text{ub}}, \alpha_h$, and $\alpha_y$ satisfying (by default, $\alpha_z^{\text{lb}} = 0.3$, $\alpha_z^{\text{ub}} = \alpha_h = 5$, and $\alpha_y = 3$)

$$0 < \alpha_z^{\text{lb}} \leq 0.5, \quad \alpha_z^{\text{ub}} \geq 5, \quad \alpha_h \geq 5, \quad \text{and} \quad \alpha_y \geq 3. \tag{25}$$

**Initialize** $\boldsymbol{z}^{(0)}$ to be $\sqrt{\frac{mn}{\sum_{i=1}^m \|\boldsymbol{a}_i\|^2}} \lambda \tilde{\boldsymbol{z}}$, where $\lambda = \sqrt{\frac{1}{m}\sum_{i=1}^m y_i}$ and $\tilde{\boldsymbol{z}}$ is the leading eigenvector of

$$\boldsymbol{Y} = \frac{1}{m}\sum_{i=1}^m y_i \boldsymbol{a}_i \boldsymbol{a}_i^* \mathbf{1}_{\{|y_i| \leq \alpha_y^2 \lambda_0^2\}}. \tag{22}$$

**Loop: for** $t = 0 : T$ **do**

$$\boldsymbol{z}^{(t+1)} = \boldsymbol{z}^{(t)} + \frac{2\mu_t}{m}\sum_{i=1}^m \frac{y_i - \left|\boldsymbol{a}_i^* \boldsymbol{z}^{(t)}\right|^2}{\boldsymbol{z}^{(t)*} \boldsymbol{a}_i} \boldsymbol{a}_i \mathbf{1}_{\mathcal{E}_1^i \cap \mathcal{E}_2^i}, \tag{23}$$

where

$$\mathcal{E}_1^i := \left\{\alpha_z^{\text{lb}} \leq \frac{\sqrt{n}}{\|\boldsymbol{a}_i\|} \frac{|\boldsymbol{a}_i^* \boldsymbol{z}^{(t)}|}{\|\boldsymbol{z}^{(t)}\|} \leq \alpha_z^{\text{ub}}\right\}, \quad \mathcal{E}_2^i := \left\{|y_i - |\boldsymbol{a}_i^* \boldsymbol{z}^{(t)}|^2| \leq \alpha_h K_t \frac{\sqrt{n}}{\|\boldsymbol{a}_i\|} \frac{|\boldsymbol{a}_i^* \boldsymbol{z}^{(t)}|}{\|\boldsymbol{z}^{(t)}\|}\right\}, \tag{24}$$

$$\text{and} \quad K_t := \frac{1}{m}\sum_{l=1}^m \left|y_l - |\boldsymbol{a}_l^* \boldsymbol{z}^{(t)}|^2\right|.$$

**Output** $\boldsymbol{z}^{(T)}$.

---

procedure is the spectral method [14] [13], which initializes $\boldsymbol{z}^{(0)}$ as the leading eigenvector of $\widetilde{\boldsymbol{Y}} := \frac{1}{m}\sum_{i=1}^m y_i \boldsymbol{a}_i \boldsymbol{a}_i^\top$. This is based on the observation that for any fixed unit vector $\boldsymbol{x}$,

$$\mathbb{E}[\widetilde{\boldsymbol{Y}}] = \boldsymbol{I} + 2\boldsymbol{x}\boldsymbol{x}^\top,$$

whose principal component is exactly $\boldsymbol{x}$ with an eigenvalue of 3.

Unfortunately, the success of this method requires a sample complexity exceeding $n \log n$. To see this, recall that $\max_i y_i \approx 2\log m$. Letting $k = \arg\max_i y_i$ and $\tilde{\boldsymbol{a}}_k := \boldsymbol{a}_k / \|\boldsymbol{a}_k\|$, one can derive

$$\tilde{\boldsymbol{a}}_k^\top \widetilde{\boldsymbol{Y}} \tilde{\boldsymbol{a}}_k \geq \tilde{\boldsymbol{a}}_k^\top \left(m^{-1}\boldsymbol{a}_k \boldsymbol{a}_k^\top y_k\right) \tilde{\boldsymbol{a}}_k \approx (2n\log m)/m,$$

which dominates $\boldsymbol{x}^\top \widetilde{\boldsymbol{Y}} \boldsymbol{x} \approx 3$ unless $m \gtrsim n\log m$. As a result, $\tilde{\boldsymbol{a}}_k$ is closer to the principal component of $\widetilde{\boldsymbol{Y}}$ than $\boldsymbol{x}$ when $m \asymp n$. This drawback turns out to be a substantial practical issue.

This issue can be remedied if we preclude those data $y_i$ with large magnitudes when running the spectral method. Specifically, we propose to initialize $\boldsymbol{z}^{(0)}$ as the leading eigenvector of

$$\boldsymbol{Y} := \frac{1}{m}\sum_{i=1}^m y_i \boldsymbol{a}_i \boldsymbol{a}_i^\top \mathbf{1}_{\left\{|y_i| \leq \alpha_y^2 \left(\frac{1}{m}\sum_{l=1}^m y_l\right)\right\}} \tag{26}$$

followed by proper scaling so as to ensure $\|\boldsymbol{z}^{(0)}\| \approx \|\boldsymbol{x}\|$. As illustrated in Fig. 4, the empirical advantage of the truncated spectral method is increasingly more remarkable as $n$ grows.

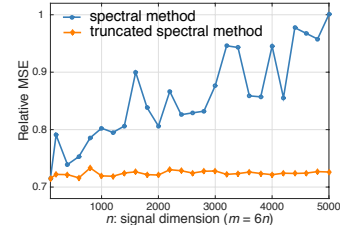

Figure 4: Relative initialization error when $\boldsymbol{a}_i \sim \mathcal{N}(\boldsymbol{0}, \boldsymbol{I})$.

## 2.3 Choice of algorithmic parameters

One important implementation detail is the learning rate $\mu_t$. There are two alternatives that work well in both theory and practice:

1. **Fixed size**. Take $\mu_t \equiv \mu$ for some constant $\mu > 0$. As long as $\mu$ is not too large, this strategy always works. Under the condition (25), our theorems hold for any positive constant $\mu < 0.28$.

2. **Backtracking line search with truncated objective**. This strategy performs a line search along the descent direction and determines an appropriate learning rate that guarantees a sufficient improvement with respect to the truncated objective. Details are deferred to the supplement.

Another algorithmic details to specify are the truncation thresholds $\alpha_h$, $\alpha_z^{\text{lb}}$, $\alpha_z^{\text{ub}}$, and $\alpha_y$. The present paper isolates a concrete set of combinations as given in (25). In all theory and numerical experiments presented in this work, we assume that the parameters fall within this range.

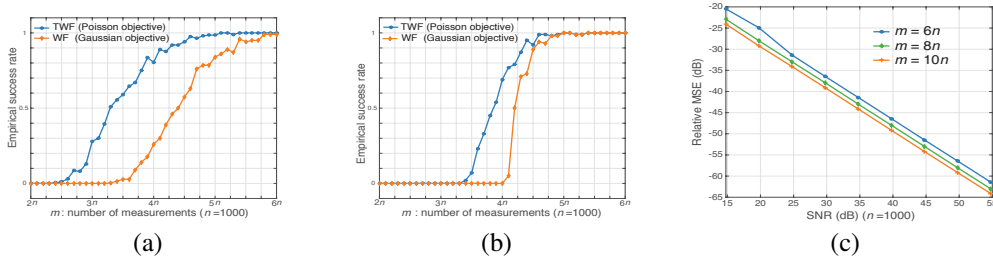

(a)  (b)  (c)

Figure 5: (a) Empirical success rates for real Gaussian design; (b) empirical success rates for complex Gaussian design; (c) relative MSE (averaged over 100 runs) vs. SNR for Poisson data.

## 3  More numerical experiments and discussion

We conduct more extensive numerical experiments to corroborate our main results and verify the applicability of TWF on practical problems. For all experiments conducted herein, we take a fixed step size $\mu_t \equiv 0.2$, employ 50 power iterations for initialization and $T = 1000$ gradient iterations. The truncation levels are taken to be the default values $\alpha_z^{\text{lb}} = 0.3$, $\alpha_z^{\text{ub}} = \alpha_h = 5$, and $\alpha_y = 3$.

We first apply TWF to a sequence of noiseless problems with $n = 1000$ and varying $m$. Generate the object $\boldsymbol{x}$ at random, and produce the feature vectors $\boldsymbol{a}_i$ in two different ways: (1) $\boldsymbol{a}_i \overset{\text{ind.}}{\sim} \mathcal{N}(\boldsymbol{0}, \boldsymbol{I})$; (2) $\boldsymbol{a}_i \overset{\text{ind.}}{\sim} \mathcal{N}(\boldsymbol{0}, \boldsymbol{I}) + j\mathcal{N}(\boldsymbol{0}, \boldsymbol{I})$. A Monte Carlo trial is declared success if the estimate $\hat{\boldsymbol{x}}$ obeys $\text{dist}(\hat{\boldsymbol{x}}, \boldsymbol{x}) / \|\boldsymbol{x}\| \leq 10^{-5}$. Fig. 5(a) and 5(b) illustrate the empirical success rates of TWF (average over 100 runs for each $m$) for noiseless data, indicating that $m \geq 5n$ are $m \geq 4.5n$ are often sufficient under real and complex Gaussian designs, respectively. For the sake of comparison, we simulate the empirical success rates of WF, with the step size $\mu_t = \min\{1 - e^{-t/330}, 0.2\}$ as recommended by [13]. As shown in Fig. 5, TWF outperforms WF under random Gaussian features, implying that TWF exhibits either better convergence rate or enhanced phase transition behavior.

Next, we empirically evaluate the stability of TWF under noisy data. Set $n = 1000$, produce $\boldsymbol{a}_i \overset{\text{ind.}}{\sim} \mathcal{N}(\boldsymbol{0}, \boldsymbol{I})$, and generate $y_i$ according to the Poisson model (3). Fig. 5(c) shows the relative mean square error—on the dB scale—with varying SNR (cf. (8)). As can be seen, the empirical relative MSE scales inversely proportional to SNR, which matches our stability guarantees in Theorem 2 (since on the dB scale, the slope is about -1 as predicted by the theory (16)).

While this work focuses on the Poisson-type objective for concreteness, the proposed paradigm carries over to a variety of nonconvex objectives, and might have implications in solving other problems that involve latent variables, e.g. matrix completion [23–25], sparse coding [26], dictionary learning [27], and mixture problems (e.g. [28, 29]). We conclude this paper with an example on estimating mixtures of linear regression. Imagine

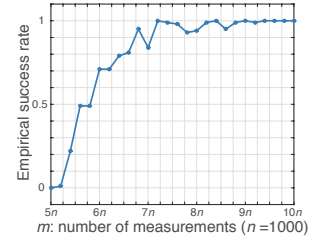

Figure 6: Empirical success rate for mixed regression ($p = 0.5$).

$$y_i \approx \begin{cases} \boldsymbol{a}_i^\top \boldsymbol{\beta}_1, & \text{with probability } p, \\ \boldsymbol{a}_i^\top \boldsymbol{\beta}_2, & \text{else,} \end{cases} \qquad 1 \leq i \leq m, \qquad (27)$$

where $\boldsymbol{\beta}_1, \boldsymbol{\beta}_2$ are unknown. It has been shown in [3] that in the noiseless case, the ground truth satisfies

$$f_i(\boldsymbol{\beta_1}, \boldsymbol{\beta_2}) := y_i^2 + 0.5\boldsymbol{a}_i^\top (\boldsymbol{\beta}_1\boldsymbol{\beta}_2^\top + \boldsymbol{\beta}_2\boldsymbol{\beta}_1^\top)\boldsymbol{a}_i - \boldsymbol{a}_i^\top (\boldsymbol{\beta}_1 + \boldsymbol{\beta}_2) y_i = 0, \qquad 1 \leq i \leq m,$$

which forms a set of quadratic constraints (in particular, if one further knows $\boldsymbol{\beta}_1 = -\boldsymbol{\beta}_2$, then this reduces to the form (1)). Running TWF with a nonconvex objective $\sum_{i=1}^m f_i^2(\boldsymbol{z_1}, \boldsymbol{z_2})$ (with the assistance of a 1-D grid search proposed in [29] applied right after truncated initialization) yields accurate estimation of $\boldsymbol{\beta}_1, \boldsymbol{\beta}_2$ under minimal sample complexity, as illustrated in Fig. 6.

#### Acknowledgments

E. C. is partially supported by NSF under grant CCF-0963835 and by the Math + X Award from the Simons Foundation. Y. C. is supported by the same NSF grant.

## Footnotes

[1] $f(n) = \mathcal{O}(g(n))$ or $f(n) \lesssim g(n)$ (resp. $f(n) \gtrsim g(n)$) means there exists a constant $c > 0$ such that $|f(n)| \leq c|g(n)|$ (resp. $|f(n)| \geq c|g(n)|$). $f(n) \asymp g(n)$ means $f(n)$ and $g(n)$ are orderwise equivalent.

[2]Similar phenomena arise in many other experiments we've conducted (e.g. when the sample size $m$ ranges from $6n$ to $20n$). In fact, this factor seems to improve slightly as $m/n$ increases.

[3]To justify the definition of SNR, note that the signals and noise are captured by $\mu_i = (\boldsymbol{a}_i^\top \boldsymbol{x})^2$ and $y_i - \mu_i$, respectively. The SNR is thus given by $\frac{\sum_{i=1}^m \mu_i^2}{\sum_{i=1}^m \mathbf{Var}[y_i]} = \frac{\sum_{i=1}^m |\boldsymbol{a}_i^\top \boldsymbol{x}|^4}{\sum_{i=1}^m |\boldsymbol{a}_i^\top \boldsymbol{x}|^2} \approx \frac{3m\|\boldsymbol{x}\|^4}{m\|\boldsymbol{x}\|^2} = 3\|\boldsymbol{x}\|^2$.

[4]For complex-valued data, WF converges empirically, as $\min_i |a_i^\top z|$ is much larger than the real case.

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
