[Supplementary Material · supplement_NIPS.pdf]

# Supplemental Materials for: "Solving Random Quadratic Systems of Equations Is Nearly as Easy as Solving Linear Systems"

Yuxin Chen [*]          Emmanuel J. Candès [*†]

## 1 Main Theorems

For convenience of presentation, we repeat the main results as follows. To begin with, the noiseless model, the general additive noise model, and the Poisson noise model are given respectively as follows.

$$\text{(noiseless:)} \qquad y_i = |\langle \boldsymbol{a}_i, \boldsymbol{x} \rangle|^2, \qquad i = 1, \cdots, m, \tag{1}$$

$$\text{(general noise:)} \qquad y_i = |\langle \boldsymbol{a}_i, \boldsymbol{x} \rangle|^2 + \eta_i, \qquad i = 1, \cdots, m, \tag{2}$$

$$\text{(Poisson noise:)} \qquad y_i \overset{\text{ind.}}{\sim} \mathsf{Poisson}\big(|\langle \boldsymbol{a}_i, \boldsymbol{x} \rangle|^2\big), \qquad i = 1, \cdots, m. \tag{3}$$

**Theorem 1 (Exact recovery).** *Consider the noiseless case (1) with an arbitrary signal $\boldsymbol{x} \in \mathbb{R}^n$. Suppose that the step size $\mu_t$ is either taken to be a positive constant $\mu_t \equiv \mu$ or chosen via a backtracking line search. Then there exist some universal constants $0 < \rho, \nu < 1$ and $\mu_0, c_0, c_1, c_2 > 0$ such that with probability exceeding $1 - c_1 \exp(-c_2 m)$, the truncated Wirtinger Flow estimates (Algorithm 1) obey*

$$\text{dist}(\boldsymbol{z}^{(t)}, \boldsymbol{x}) \leq \nu(1-\rho)^t \|\boldsymbol{x}\|, \quad \forall t \in \mathbb{N}, \tag{4}$$

*provided that*

$$m \geq c_0 n \quad and \quad \mu \leq \mu_0.$$

**Theorem 2 (Stability).** *Consider the noisy case (2). Suppose that the step size $\mu_t$ is either taken to be a positive constant $\mu_t \equiv \mu$ or chosen via a backtracking line search. If*

$$m \geq c_0 n, \quad \mu \leq \mu_0, \quad and \quad \|\boldsymbol{\eta}\|_\infty \leq c_1 \|\boldsymbol{x}\|^2, \tag{5}$$

*then with probability at least $1 - c_2 \exp(-c_3 m)$, the truncated Wirtinger Flow estimates (Algorithm 1) satisfy*

$$\text{dist}(\boldsymbol{z}^{(t)}, \boldsymbol{x}) \lesssim \frac{\|\boldsymbol{\eta}\|}{\sqrt{m}\|\boldsymbol{x}\|} + (1-\rho)^t \|\boldsymbol{x}\|, \quad \forall t \in \mathbb{N} \tag{6}$$

*simultaneosuly for all $\boldsymbol{x} \in \mathbb{R}^n$. Here, $0 < \rho < 1$ and $\mu_0, c_0, c_1, c_2, c_3 > 0$ are some universal constants.*

In particular, under the Poisson noise model, there exists an an event of probability at least $1 - c_2 \exp(-c_3 m)$ on which

$$\mathbb{P}\Big\{\text{dist}(\boldsymbol{z}^{(t)}, \boldsymbol{x}) \lesssim 1 + (1-\rho)^t \|\boldsymbol{x}\|, \ \forall t \in \mathbb{N} \ \Big| \ \{\boldsymbol{a}_i\}_{1 \leq i \leq m}\Big\} \to 1. \tag{7}$$

holds for all $\boldsymbol{x} \in \mathbb{R}^n$ satisfying $\|\boldsymbol{x}\| \geq \log^{1.5} m$. In what follows, we prove the above two theorems for a broader range of algorithmic parameters, as summarized in Table 1.

Encouragingly, this is already the best statistical guarantee any algorithm can achieve. We formalize this claim by deriving a fundamental lower bound on the minimax estimation error.

---

[*]Department of Statistics, Stanford University, Stanford, CA 94305, U.S.A.
[†]Department of Mathematics, Stanford University, Stanford, CA 94305, U.S.A.

Table 1: Range of algorithmic parameters

(a) **When a fixed step size $\mu_t \equiv \mu$ is employed:** $(\alpha_z^{\mathrm{lb}}, \alpha_z^{\mathrm{ub}}, \alpha_h, \alpha_y)$ obeys

$$
\begin{cases}
\zeta_1 := \max\left\{ \mathbb{E}\left[\xi^2 \mathbf{1}_{\{|\xi| \le \sqrt{1.01}\alpha_z^{\mathrm{lb}} \text{ or } |\xi| \ge \sqrt{0.99}\alpha_z^{\mathrm{ub}}\}}\right], \mathbb{P}\left(|\xi| \le \sqrt{1.01}\alpha_z^{\mathrm{lb}} \text{ or } |\xi| \ge \sqrt{0.99}\alpha_z^{\mathrm{ub}}\right)\right\} \\
\zeta_2 := \mathbb{E}\left[\xi^2 \mathbf{1}_{\{|\xi| > 0.473\alpha_h\}}\right], \\
2(\zeta_1 + \zeta_2) + \sqrt{8/(9\pi)}\alpha_h^{-1} < 1.99, \\
\alpha_y \ge 3,
\end{cases} \tag{8}
$$

where $\xi \sim \mathcal{N}(0,1)$. By default, $\alpha_z^{\mathrm{lb}} = 0.3$, $\alpha_z^{\mathrm{ub}} = \alpha_h = 5$, and $\alpha_y = 3$.

(b) **When $\mu_t$ is chosen by a backtracking line search:** $(\alpha_z^{\mathrm{lb}}, \alpha_z^{\mathrm{ub}}, \alpha_h, \alpha_y, \alpha_p)$ obeys

$$
0 < \alpha_z^{\mathrm{lb}} \le 0.1, \quad \alpha_z^{\mathrm{ub}} \ge 5, \quad \alpha_h \ge 6, \quad \alpha_y \ge 3, \quad \text{and} \quad \alpha_p \ge 5. \tag{9}
$$

By default, $\alpha_z^{\mathrm{lb}} = 0.1$, $\alpha_z^{\mathrm{ub}} = 5$, $\alpha_h = 6$, $\alpha_y = 3$, and $\alpha_p = 5$.

**Theorem 3** (**Lower bound on the minimax risk**). *Suppose that $\boldsymbol{a}_i \sim \mathcal{N}(\boldsymbol{0}, \boldsymbol{I})$, $m = \kappa n$ for some fixed $\kappa$ independent of $n$, and $n$ is sufficiently large. For any $K \ge \log^{1.5} m$, define[1]*

$$
\Upsilon(K) := \{\boldsymbol{x} \in \mathbb{R}^n \mid \|\boldsymbol{x}\| \in (1 \pm 0.1)K\}.
$$

*With probability approaching one, the minimax risk under the Poisson model (3) obeys*

$$
\inf_{\hat{\boldsymbol{x}}} \sup_{\boldsymbol{x} \in \Upsilon(K)} \mathbb{E}\left[\mathrm{dist}\left(\hat{\boldsymbol{x}}, \boldsymbol{x}\right) \mid \{\boldsymbol{a}_i\}_{1 \le i \le m}\right] \ge \frac{\varepsilon_1}{\sqrt{\kappa}}, \tag{10}
$$

*where the infimum is over all estimator $\hat{\boldsymbol{x}}$. Here, $\varepsilon_1 > 0$ is a numerical constant independent of $n$ and $m$.*

## 2 Exact recovery from noiseless data

This section proves the theoretical guarantees of TWF in the absence of noise (i.e. Theorem 1). We separate the noiseless case mainly out of pedagogical reasons, as most of the steps carry over to the noisy case with slight modification.

The analysis for the truncated spectral method follows similar argument as in [1, Section 7.8], which we defer to Appendix C. In short, for any fixed $\delta > 0$ and $\boldsymbol{x} \in \mathbb{R}^n$, the initial point $\boldsymbol{z}^{(0)}$ returned by the truncated spectral method obeys

$$
\mathrm{dist}(\boldsymbol{z}^{(0)}, \boldsymbol{x}) \le \delta\|\boldsymbol{x}\|
$$

with high probability, provided that $m/n$ exceeds some large constant.

The remaining section then boils down to establishing convergence for the gradient flow stage. To this end, we recall a (local) regularity condition given in [1], which has been shown to be a fundamental criterion that dictates rapid convergence of iterative procedures (including WF and other gradient descent schemes). When specialized to TWF, we say that $-\frac{1}{m}\nabla\ell_{\mathrm{tr}}(\cdot)$ satisfies the *regularity condition*, denoted by $\mathsf{RC}(\mu, \lambda, \epsilon)$, if

$$
\left\langle \boldsymbol{h}, -\frac{1}{m}\nabla\ell_{\mathrm{tr}}(\boldsymbol{z}) \right\rangle \ge \frac{\mu}{2}\left\|\frac{1}{m}\nabla\ell_{\mathrm{tr}}(\boldsymbol{z})\right\|^2 + \frac{\lambda}{2}\|\boldsymbol{h}\|^2 \tag{11}
$$

holds for all $\boldsymbol{z}$ obeying $\|\boldsymbol{z} - \boldsymbol{x}\| \le \epsilon\|\boldsymbol{x}\|$, where $0 < \epsilon < 1$ is some constant. Such an $\epsilon$-ball around $\boldsymbol{x}$ is

sometimes referred to as a *basin of attraction*. Formally, under $\mathsf{RC}\,(\mu, \lambda, \epsilon)$, a little algebra gives

$$
\begin{aligned}
\mathrm{dist}^2\left(\boldsymbol{z} + \frac{\mu}{m}\nabla \ell_{\mathrm{tr}}\left(\boldsymbol{z}\right), \boldsymbol{x}\right) &\leq \left\|\boldsymbol{z} + \frac{\mu}{m}\nabla \ell_{\mathrm{tr}}\left(\boldsymbol{z}\right) - \boldsymbol{x}\right\|^2 \\
&= \|\boldsymbol{h}\|^2 + \left\|\frac{\mu}{m}\nabla \ell_{\mathrm{tr}}\left(\boldsymbol{z}\right)\right\|^2 + 2\mu\left\langle \boldsymbol{h}, \frac{1}{m}\nabla \ell_{\mathrm{tr}}\left(\boldsymbol{z}\right)\right\rangle \\
&\leq \|\boldsymbol{h}\|^2 + \left\|\frac{\mu}{m}\nabla \ell_{\mathrm{tr}}\left(\boldsymbol{z}\right)\right\|^2 - \mu^2\left\|\frac{1}{m}\nabla \ell_{\mathrm{tr}}\left(\boldsymbol{z}\right)\right\|^2 - \mu\lambda\|\boldsymbol{h}\|^2 \\
&= (1 - \mu\lambda)\,\mathrm{dist}^2\left(\boldsymbol{z}, \boldsymbol{x}\right)
\end{aligned}
\tag{12}
$$

for any $\boldsymbol{z}$ with $\|\boldsymbol{z} - \boldsymbol{x}\| \leq \epsilon$. In words, the TWF update rule is locally contractive around the planted solution, provided that $\mathsf{RC}\,(\mu, \lambda, \epsilon)$ holds for some nonzero $\mu$ and $\lambda$. This is stated in the following proposition.

**Proposition 1** (**Local error contraction**). *Consider the noiseless case (1). Under the condition (8), there exist some universal constants $0 < \rho_0 < 1$ and $c_0, c_1, c_2 > 0$ such that with probability exceeding $1 - c_1 \exp\left(-c_2 m\right)$,*

$$
\mathrm{dist}^2\left(\boldsymbol{z} + \frac{\mu}{m}\nabla \ell_{\mathrm{tr}}\left(\boldsymbol{z}\right), \boldsymbol{x}\right) \leq (1 - \rho_0)\,\mathrm{dist}^2\left(\boldsymbol{z}, \boldsymbol{x}\right)
\tag{13}
$$

*holds simultaneously for all $\boldsymbol{x}, \boldsymbol{z} \in \mathbb{R}^n$ obeying*

$$
\frac{\mathrm{dist}\,(\boldsymbol{z}, \boldsymbol{x})}{\|\boldsymbol{z}\|} \leq \min\left\{\frac{1}{11}, \frac{\alpha_z^{\mathrm{lb}}}{3\alpha_h}, \frac{\alpha_z^{\mathrm{lb}}}{6}, \frac{5.7\left(\alpha_z^{\mathrm{lb}}\right)^2}{2\alpha_z^{\mathrm{ub}} + \alpha_z^{\mathrm{lb}}}\right\},
\tag{14}
$$

*provided that $m \geq c_0 n$ and that $\mu$ is some constant obeying $0 < \mu \leq \mu_0 := \frac{0.994 - \zeta_1 - \zeta_2 - \sqrt{2/(9\pi)}\alpha_h^{-1}}{2(1.02 + 0.665/\alpha_h)}$.*

Proposition 1 reveals the monotonicity of the estimation error: once entering a neighborhood around $\boldsymbol{x}$ of a reasonably small size, the iterative updates will remain within this neighborhood all the time and be attracted towards $\boldsymbol{x}$ at a geometric rate.

As shown before, under the hypothesis $\mathsf{RC}\,(\mu, \lambda, \epsilon)$ one can conclude

$$
\mathrm{dist}^2\left(\boldsymbol{z} + \frac{\mu}{m}\nabla \ell_{\mathrm{tr}}(\boldsymbol{z}), \boldsymbol{x}\right) \leq (1 - \mu\lambda)\mathrm{dist}^2(\boldsymbol{z}, \boldsymbol{x}), \quad \forall(\boldsymbol{z}, \boldsymbol{x}) \text{ with } \mathrm{dist}(\boldsymbol{z}, \boldsymbol{x}) \leq \epsilon.
\tag{15}
$$

Thus, everything now boils down to showing $\mathsf{RC}\,(\mu, \lambda, \epsilon)$ for some constants $\mu, \lambda, \epsilon > 0$. This occupies the rest of this section.

## 2.1 Preliminary facts about $\{\mathcal{E}_1^i\}$ and $\{\mathcal{E}_2^i\}$

Before proceeding, we gather a few properties of the events $\mathcal{E}_1^i$ and $\mathcal{E}_2^i$:

$$
\mathcal{E}_1^i(\boldsymbol{z}) := \left\{\alpha_z^{\mathrm{lb}} \leq \frac{|\boldsymbol{a}_i^\top \boldsymbol{z}|}{\|\boldsymbol{z}\|} \leq \alpha_z^{\mathrm{ub}}\right\};
\tag{16}
$$

$$
\mathcal{E}_2^i(\boldsymbol{z}) := \left\{|y_i - |\boldsymbol{a}_i^\top \boldsymbol{z}|^2| \leq \frac{\alpha_h}{m}\left\|\boldsymbol{y} - \mathcal{A}\left(\boldsymbol{z}\boldsymbol{z}^\top\right)\right\|_1 \frac{|\boldsymbol{a}_i^\top \boldsymbol{z}|}{\|\boldsymbol{z}\|}\right\},
\tag{17}
$$

which will prove crucial in establishing $\mathsf{RC}\,(\mu, \lambda, \epsilon)$. To begin with, recall that the truncation level given in $\mathcal{E}_2^i$ depends on $\frac{1}{m}\left\|\mathcal{A}\left(\boldsymbol{x}\boldsymbol{x}^\top - \boldsymbol{z}\boldsymbol{z}^\top\right)\right\|_1$. Instead of working with this random variable directly, we use deterministic quantities that are more amenable to analysis. Specifically, we claim that $\frac{1}{m}\left\|\mathcal{A}\left(\boldsymbol{x}\boldsymbol{x}^\top - \boldsymbol{z}\boldsymbol{z}^\top\right)\right\|_1$ offers a uniform and orderwise tight estimate on $\|\boldsymbol{h}\|\,\|\boldsymbol{z}\|$, which can be seen from the following two facts.

**Lemma 1.** *Fix $\zeta \in (0, 1)$. If $m > c_0 n \zeta^{-2} \log \frac{1}{\zeta}$, then with probability at least $1 - C \exp(-c_1 \zeta^2 m)$,*

$$
0.9\,(1 - \zeta)\,\|\boldsymbol{M}\|_{\mathrm{F}} \leq \frac{1}{m}\left\|\mathcal{A}\left(\boldsymbol{M}\right)\right\|_1 \leq (1 + \zeta)\,\sqrt{2}\,\|\boldsymbol{M}\|_{\mathrm{F}}
\tag{18}
$$

*holds for all symmetric rank-2 matrices $\boldsymbol{M} \in \mathbb{R}^{n \times n}$. Here, $c_0, c_1, C > 0$ are some universal constants.*

Figure 1: $\frac{f(t)}{\sqrt{1+t^2}}$ as a function of $t$.

*Proof.* Since [2, Lemma 3.1] already establishes the upper bound, it suffices to prove the lower tail bound. Consider all symmetric rank-2 matrices $\boldsymbol{M}$ with eigenvalues 1 and $-t$ for some $-1 \leq t \leq 1$. When $t \in [0,1]$, it has been shown in the proof of [2, Lemma 3.2] that with high probability,

$$\frac{1}{m} \|\mathcal{A}(\boldsymbol{M})\|_1 \geq (1-\zeta) f(t), \tag{19}$$

for all such rank-2 matrices $\boldsymbol{M}$, where $f(t) := \frac{2}{\pi}\left\{2\sqrt{t} + (1-t)\left(\pi/2 - 2\arctan(\sqrt{t})\right)\right\}$. The lower bound in this case can then be justified by recognizing that $f(t)/\sqrt{1+t^2} \geq 0.9$ for all $t \in [0,1]$, as illustrated in Fig. 1. The case where $t \in [-1,0]$ is an immediate consequence from [2, Lemma 3.1]. $\qquad\square$

**Lemma 2.** *Consider any* $\boldsymbol{x}, \boldsymbol{z} \in \mathbb{R}^n$ *obeying* $\|\boldsymbol{z} - \boldsymbol{x}\| \leq \delta \|\boldsymbol{z}\|$ *for some* $\delta < \frac{1}{2}$. *Then one has*

$$\sqrt{2-4\delta} \|\boldsymbol{z}-\boldsymbol{x}\| \|\boldsymbol{z}\| \leq \left\|\boldsymbol{x}\boldsymbol{x}^\top - \boldsymbol{z}\boldsymbol{z}^\top\right\|_{\mathrm{F}} \leq (2+\delta) \|\boldsymbol{z}-\boldsymbol{x}\| \|\boldsymbol{z}\|. \tag{20}$$

*Proof.* Take $\boldsymbol{h} = \boldsymbol{z} - \boldsymbol{x}$ and write

$$\begin{aligned}
\left\|\boldsymbol{x}\boldsymbol{x}^\top - \boldsymbol{z}\boldsymbol{z}^\top\right\|_{\mathrm{F}}^2 &= \left\| -\boldsymbol{h}\boldsymbol{z}^\top - \boldsymbol{z}\boldsymbol{h}^\top + \boldsymbol{h}\boldsymbol{h}^\top\right\|_{\mathrm{F}}^2 \\
&= \left\|\boldsymbol{h}\boldsymbol{z}^\top + \boldsymbol{z}\boldsymbol{h}^\top\right\|_{\mathrm{F}}^2 + \|\boldsymbol{h}\|^4 - 2\langle \boldsymbol{h}\boldsymbol{z}^\top + \boldsymbol{z}\boldsymbol{h}^\top, \boldsymbol{h}\boldsymbol{h}^\top\rangle \\
&= 2\|\boldsymbol{z}\|^2 \|\boldsymbol{h}\|^2 + 2|\boldsymbol{h}^\top\boldsymbol{z}|^2 + \|\boldsymbol{h}\|^4 - 2\|\boldsymbol{h}\|^2(\boldsymbol{h}^\top\boldsymbol{z} + \boldsymbol{z}^\top\boldsymbol{h}).
\end{aligned}$$

When $\|\boldsymbol{h}\| < \frac{1}{2}\|\boldsymbol{z}\|$, the Cauchy–Schwartz inequality gives

$$2\|\boldsymbol{z}\|^2 \|\boldsymbol{h}\|^2 - 4\|\boldsymbol{z}\| \|\boldsymbol{h}\|^3 \leq \left\|\boldsymbol{x}\boldsymbol{x}^\top - \boldsymbol{z}\boldsymbol{z}^\top\right\|_{\mathrm{F}}^2 \leq 4\|\boldsymbol{z}\|^2 \|\boldsymbol{h}\|^2 + 4\|\boldsymbol{h}\|^3 \|\boldsymbol{z}\| + \|\boldsymbol{h}\|^4, \tag{21}$$

$$\Rightarrow \quad \sqrt{(2\|\boldsymbol{z}\| - 4\|\boldsymbol{h}\|)\|\boldsymbol{z}\|} \cdot \|\boldsymbol{h}\| \leq \left\|\boldsymbol{x}\boldsymbol{x}^\top - \boldsymbol{z}\boldsymbol{z}^\top\right\|_{\mathrm{F}} \leq (2\|\boldsymbol{z}\| + \|\boldsymbol{h}\|) \cdot \|\boldsymbol{h}\| \tag{22}$$

as claimed. $\qquad\square$

Taken together the above two facts demonstrate that with probability $1 - \exp(-\Omega(m))$,

$$1.15 \|\boldsymbol{z}-\boldsymbol{x}\| \|\boldsymbol{z}\| \leq \frac{1}{m} \left\|\mathcal{A}\left(\boldsymbol{x}\boldsymbol{x}^\top - \boldsymbol{z}\boldsymbol{z}^\top\right)\right\|_1 \leq 3 \|\boldsymbol{z}-\boldsymbol{x}\| \|\boldsymbol{z}\| \tag{23}$$

holds simultaneously for all $\boldsymbol{z}$ and $\boldsymbol{x}$ satisfying $\|\boldsymbol{h}\| \leq \frac{1}{11}\|\boldsymbol{z}\|$. Conditional on (23), the inclusion

$$\mathcal{E}_3^i \subseteq \mathcal{E}_2^i \subseteq \mathcal{E}_4^i \tag{24}$$

holds with respect to the following events

$$\begin{aligned}
\mathcal{E}_3^i &:= \left\{\left||\boldsymbol{a}_i^\top \boldsymbol{x}|^2 - |\boldsymbol{a}_i^\top \boldsymbol{z}|^2\right| \leq 1.15\alpha_h \|\boldsymbol{h}\| \cdot |\boldsymbol{a}_i^\top \boldsymbol{z}|\right\}, \tag{25} \\
\mathcal{E}_4^i &:= \left\{\left||\boldsymbol{a}_i^\top \boldsymbol{x}|^2 - |\boldsymbol{a}_i^\top \boldsymbol{z}|^2\right| \leq 3\alpha_h \|\boldsymbol{h}\| \cdot |\boldsymbol{a}_i^\top \boldsymbol{z}|\right\}. \tag{26}
\end{aligned}$$

The point of introducing these new events is that $\mathcal{E}_3^i$'s (resp. $\mathcal{E}_4^i$'s) are statistically independent for any fixed $\boldsymbol{x}$ and $\boldsymbol{z}$ and are, therefore, easier to work with.

Note that each $\mathcal{E}_3^i$ (resp. $\mathcal{E}_4^i$) is specified by a quadratic inequality. A closer inspection reveals that in order to satisfy these quadratic inequalities, the quantity $\boldsymbol{a}_i^\top \boldsymbol{h}$ must fall within two intervals centered around $0$ and $2\boldsymbol{a}_i^\top \boldsymbol{z}$, respectively. One can thus facilitate analysis by decoupling each quadratic inequality of interest into two simple linear inequalities, as stated in the following lemma.

**Lemma 3.** *For any $\gamma > 0$, define*

$$\mathcal{D}_\gamma^i \quad := \quad \left\{ \left| |\boldsymbol{a}_i^\top \boldsymbol{x}|^2 - |\boldsymbol{a}_i^\top \boldsymbol{z}|^2 \right| \leq \gamma \|\boldsymbol{h}\| \, |\boldsymbol{a}_i^\top \boldsymbol{z}| \right\}, \tag{27}$$

$$\mathcal{D}_\gamma^{i,1} \quad := \quad \left\{ \frac{|\boldsymbol{a}_i^\top \boldsymbol{h}|}{\|\boldsymbol{h}\|} \leq \gamma \right\}, \tag{28}$$

$$and \quad \mathcal{D}_\gamma^{i,2} \quad := \quad \left\{ \left| \frac{\boldsymbol{a}_i^\top \boldsymbol{h}}{\|\boldsymbol{h}\|} - \frac{2\boldsymbol{a}_i^\top \boldsymbol{z}}{\|\boldsymbol{h}\|} \right| \leq \gamma \right\}. \tag{29}$$

*Thus, $\mathcal{D}_\gamma^{i,1}$ and $\mathcal{D}_\gamma^{i,2}$ represent the two intervals on $\boldsymbol{a}_i^\top \boldsymbol{h}$ centered around $0$ and $2\boldsymbol{a}_i^\top \boldsymbol{z}$. If $\frac{\|\boldsymbol{h}\|}{\|\boldsymbol{z}\|} \leq \frac{\alpha_z^{\mathrm{lb}}}{\gamma}$, then the following inclusion holds*

$$\left( \mathcal{D}_{\frac{\gamma}{1+\sqrt{2}}}^{i,1} \cap \mathcal{E}_1^i \right) \cup \left( \mathcal{D}_{\frac{\gamma}{1+\sqrt{2}}}^{i,2} \cap \mathcal{E}_1^i \right) \subseteq \quad \mathcal{D}_\gamma^i \cap \mathcal{E}_1^i \quad \subseteq \quad \left( \mathcal{D}_\gamma^{i,1} \cap \mathcal{E}_1^i \right) \cup \left( \mathcal{D}_\gamma^{i,2} \cap \mathcal{E}_1^i \right). \tag{30}$$

## 2.2  Proof of the regularity condition

By definition, one step towards proving the regularity condition (11) is to control the norm of the truncated gradient. In fact, a crude argument already reveals that $\|\frac{1}{m}\nabla \ell_{\mathrm{tr}}(\boldsymbol{z})\| \lesssim \|\boldsymbol{h}\|$. To see this, introduce $\boldsymbol{v} = [v_i]_{1 \leq i \leq m}$ with $v_i := 2\frac{|\boldsymbol{a}_i^\top \boldsymbol{x}|^2 - |\boldsymbol{a}_i^\top \boldsymbol{z}|^2}{\boldsymbol{a}_i^\top \boldsymbol{z}} \mathbf{1}_{\mathcal{E}_1^i \cap \mathcal{E}_2^i}$. It comes from the truncation rule $\mathcal{E}_1^i$ as well as the inclusion property (24) that

$$\left| \boldsymbol{a}_i^\top \boldsymbol{z} \right| \gtrsim \|\boldsymbol{z}\| \quad \text{and} \quad \left| y_i - \left| \boldsymbol{a}_i^\top \boldsymbol{z} \right|^2 \right| \lesssim \frac{1}{m} \|\mathcal{A}(\boldsymbol{x}\boldsymbol{x}^\top - \boldsymbol{z}\boldsymbol{z}^\top)\|_1 \asymp \|\boldsymbol{h}\| \, \|\boldsymbol{z}\|,$$

implying $|v_i| \lesssim \|\boldsymbol{h}\|$ and hence $\|\boldsymbol{v}\| \lesssim \sqrt{m}\|\boldsymbol{h}\|$. The Marchenko–Pastur law gives $\|\boldsymbol{A}\| \lesssim \sqrt{m}$, whence

$$\frac{1}{m}\|\nabla \ell_{\mathrm{tr}}(\boldsymbol{z})\| = \frac{1}{m}\|\boldsymbol{A}^\top \boldsymbol{v}\| \leq \frac{1}{m}\|\boldsymbol{A}\| \cdot \|\boldsymbol{v}\| \lesssim \|\boldsymbol{h}\|. \tag{31}$$

A more refined estimate will be provided in Lemma 7.

The above argument essentially tells us that to establish RC, it suffices to verify a uniform lower bound of the form

$$-\left\langle \boldsymbol{h}, \frac{1}{m}\nabla \ell_{\mathrm{tr}}(\boldsymbol{z}) \right\rangle \gtrsim \|\boldsymbol{h}\|^2, \tag{32}$$

as formally derived in the following proposition.

**Proposition 2.** *Consider the noise-free measurements $y_i = |\boldsymbol{a}_i^\top \boldsymbol{x}|^2$ and any fixed constant $\epsilon > 0$. Under the condition (8), if $m > c_1 n$, then with probability exceeding $1 - C\exp(-c_0 m)$,*

$$-\left\langle \boldsymbol{h}, \frac{1}{m}\nabla \ell_{\mathrm{tr}}(\boldsymbol{z}) \right\rangle \geq 2\left\{ 1.99 - 2(\zeta_1 + \zeta_2) - \sqrt{8/(9\pi)}\alpha_h^{-1} - \epsilon \right\} \|\boldsymbol{h}\|^2 \tag{33}$$

*holds uniformly over all $\boldsymbol{x}, \boldsymbol{z} \in \mathbb{R}^n$ obeying*

$$\frac{\|\boldsymbol{h}\|}{\|\boldsymbol{z}\|} \leq \min\left\{ \frac{1}{11}, \frac{\alpha_z^{\mathrm{lb}}}{3\alpha_h}, \frac{\alpha_z^{\mathrm{lb}}}{6}, \frac{5.7\left(\alpha_z^{\mathrm{lb}}\right)^2}{2\alpha_z^{\mathrm{ub}} + \alpha_z^{\mathrm{lb}}} \right\}. \tag{34}$$

*Here, $c_0, c_1, C > 0$ are some universal constants, and $\zeta_1$ and $\zeta_2$ are defined in (8).*

The basic starting point is the observation that $(a_i^\top z) - (a_i^\top x)^2 = (a_i^\top h)(2a_i^\top z - a_i^\top h)$ and hence

$$
\begin{aligned}
-\frac{1}{2m}\nabla\ell_{\mathrm{tr}}(z) &= \frac{1}{m}\sum_{i=1}^{m}\frac{(a_i^\top z)^2 - (a_i^\top x)^2}{a_i^\top z}a_i \mathbf{1}_{\mathcal{E}_1^i \cap \mathcal{E}_2^i} \\
&= \frac{1}{m}\sum_{i=1}^{m}2(a_i^\top h)a_i \mathbf{1}_{\mathcal{E}_1^i \cap \mathcal{E}_2^i} - \frac{1}{m}\sum_{i=1}^{m}\frac{(a_i^\top h)^2}{a_i^\top z}a_i \mathbf{1}_{\mathcal{E}_1^i \cap \mathcal{E}_2^i}.
\end{aligned}
\tag{35}
$$

One would expect the contribution of the second term of (35) (which is a second-order quantity) to be small as $\|h\|/\|z\|$ decreases.

To facilitate analysis, we rewrite (35) in terms of the more convenient events $\mathcal{D}_\gamma^{i,1}$ and $\mathcal{D}_\gamma^{i,2}$. Specifically, the inclusion property (24) together with Lemma 3 reveals that

$$
\mathcal{D}_{\gamma_3}^{i,1} \cap \mathcal{E}_1^i \subseteq \mathcal{E}_3^i \cap \mathcal{E}_1^i \subseteq \mathcal{E}_2^i \cap \mathcal{E}_1^i \subseteq \mathcal{E}_4^i \cap \mathcal{E}_1^i \subseteq \left(\mathcal{D}_{\gamma_4}^{i,1} \cup \mathcal{D}_{\gamma_4}^{i,2}\right) \cap \mathcal{E}_1^i,
\tag{36}
$$

where the parameters $\gamma_3, \gamma_4$ are given by

$$
\gamma_3 := 0.476\alpha_h, \quad \text{and} \quad \gamma_4 := 3\alpha_h.
\tag{37}
$$

This taken collectively with the identity (35) leads to a lower estimate

$$
-\left\langle \frac{1}{2m}\nabla\ell_{\mathrm{tr}}(z), h\right\rangle \geq \frac{2}{m}\sum_{i=1}^{m}(a_i^\top h)^2 \mathbf{1}_{\mathcal{E}_1^i \cap \mathcal{D}_{\gamma_3}^{i,1}} - \frac{1}{m}\sum_{i=1}^{m}\frac{|a_i^\top h|^3}{|a_i^\top z|}\mathbf{1}_{\mathcal{E}_1^i \cap \mathcal{D}_{\gamma_4}^{i,1}} - \frac{1}{m}\sum_{i=1}^{m}\frac{|a_i^\top h|^3}{|a_i^\top z|}\mathbf{1}_{\mathcal{E}_1^i \cap \mathcal{D}_{\gamma_4}^{i,2}},
\tag{38}
$$

leaving us with three quantities in the right-hand side to deal with. We pause here to explain and compare the influences of these three terms.

To begin with, as long as the truncation step does not discard too many samples, the first term should be close to $\frac{2}{m}\sum_i |a_i^\top h|^2$, which approximately gives $2\|h\|^2$ from the law of large numbers. This term turns out to be dominant in the right-hand side of (38) as long as $\|h\|/\|z\|$ is reasonably small. To see this, please recognize that the second term in the right-hand side is $\mathcal{O}(\|h\|^3/\|z\|)$, simply because both $a_i^\top h$ and $a_i^\top z$ are absolutely controlled on $\mathcal{D}_{\gamma_4}^{i,1} \cap \mathcal{E}_1^i$. However, $\mathcal{D}_{\gamma_4}^{i,2}$ does not share such a desired feature. By the very definition of $\mathcal{D}_{\gamma_4}^{i,2}$, each nonzero summand of the last term of (38) must obey $|a_i^\top h| \approx 2|a_i^\top z|$ and, therefore, $\frac{|a_i^\top h|^3}{|a_i^\top z|}\mathbf{1}_{\mathcal{E}_1^i \cap \mathcal{D}_{\gamma_4}^{i,2}}$ is roughly of the order of $\|z\|^2$; this could be much larger than our target level $\|h\|^2$. Fortunately, $\mathcal{D}_{\gamma_4}^{i,2}$ is a rare event, thus precluding a noticeable influence upon the descent direction. All of this is made rigorous in Lemma 4 (first term), Lemma 5 (second term) and Lemma 6 (third term) together with subsequent analysis.

**Lemma 4.** *Fix $\gamma > 0$, and let $\mathcal{E}_1^i$ and $\mathcal{D}_\gamma^{i,1}$ be defined in (16) and (28), respectively. Set*

$$
\zeta_1 := 1 - \min\left\{\mathbb{E}\left[\xi^2 \mathbf{1}_{\left\{\sqrt{1.01}\alpha_z^{\mathrm{lb}} \leq |\xi| \leq \sqrt{0.99}\alpha_z^{\mathrm{ub}}\right\}}\right], \mathbb{E}\left[\mathbf{1}_{\left\{\sqrt{1.01}\alpha_z^{\mathrm{lb}} \leq |\xi| \leq \sqrt{0.99}\alpha_z^{\mathrm{ub}}\right\}}\right]\right\}
\tag{39}
$$

$$
\text{and} \quad \zeta_2 := \mathbb{E}\left[\xi^2 \mathbf{1}_{\left\{|\xi| > \sqrt{0.99}\gamma\right\}}\right],
\tag{40}
$$

*where $\xi \sim \mathcal{N}(0,1)$. For any $\epsilon > 0$, if $m > c_1 n\epsilon^{-2}\log\epsilon^{-1}$, then with probability at least $1 - C\exp(-c_0\epsilon^2 m)$,*

$$
\frac{1}{m}\sum_{i=1}^{m}|a_i^\top h|^2 \mathbf{1}_{\mathcal{E}_1^i \cap \mathcal{D}_\gamma^{i,1}} \geq (1 - \zeta_1 - \zeta_2 - \epsilon)\|h\|^2
\tag{41}
$$

*holds for all non-zero vectors $h, z \in \mathbb{R}^n$. Here, $c_0, c_1, C > 0$ are some universal constants.*

We now move on to the second term in the right-hand side of (38). For any fixed $\gamma > 0$, the definition of $\mathcal{E}_1^i$ gives rise to an upper estimate

$$
\frac{1}{m}\sum_{i=1}^{m}\frac{|a_i^\top h|^3}{|a_i^\top z|}\mathbf{1}_{\mathcal{E}_1^i \cap \mathcal{D}_\gamma^{i,1}} \leq \frac{1}{\alpha_z^{\mathrm{lb}}\|z\|}\cdot\frac{1}{m}\sum_{i=1}^{m}|a_i^\top h|^3 \mathbf{1}_{\mathcal{D}_\gamma^{i,1}} \leq \frac{(1+\epsilon)\sqrt{8/\pi}\|h\|^3}{\alpha_z^{\mathrm{lb}}\|z\|},
\tag{42}
$$

where $\sqrt{8/\pi}\,\|\boldsymbol{h}\|^3$ is exactly the untruncated moment $\mathbb{E}[|\boldsymbol{a}_i^\top \boldsymbol{h}|^3]$. The second inequality is a consequence of the lemma below, which arises by observing that the summands $|\boldsymbol{a}_i^\top \boldsymbol{h}|^3 \mathbf{1}_{\mathcal{D}_\gamma^{i,1}}$ are independent sub-Gaussian random variables.

**Lemma 5.** *For any constant $\gamma > 0$, if $m/n \geq c_0 \cdot \epsilon^{-2} \log \epsilon^{-1}$, then*

$$\frac{1}{m}\sum_{i=1}^m \left|\boldsymbol{a}_i^\top \boldsymbol{h}\right|^3 \mathbf{1}_{\mathcal{D}_\gamma^{i,1}} \leq (1+\epsilon)\sqrt{8/\pi}\,\|\boldsymbol{h}\|^3, \quad \forall \boldsymbol{h} \in \mathbb{R}^n \tag{43}$$

*with probability at least $1 - C\exp(-c_1\epsilon^2 m)$ for some universal constants $c_0, c_1, C > 0$.*

It remains to control the last term of (38). As mentioned above, the influence of this term is small since the set of $\boldsymbol{a}_i$'s satisfying $\mathcal{D}_\gamma^{i,2}$ accounts for a small fraction of measurements. Put formally, the number of equations satisfying $|\boldsymbol{a}_i^\top \boldsymbol{h}| \geq \gamma\|\boldsymbol{h}\|$ decays rapidly for large $\gamma$ (at least at a quadratic rate), as stated below.

**Lemma 6.** *For any $0 < \epsilon < 1$, there exist some universal constants $c_0, c_1, C > 0$ such that*

$$\frac{1}{m}\sum_{i=1}^m \mathbf{1}_{\{|\boldsymbol{a}_i^\top \boldsymbol{h}| \geq \gamma\|\boldsymbol{h}\|\}} \leq \frac{1}{0.49\gamma}\exp\left(-0.485\gamma^2\right) + \frac{\epsilon}{\gamma^2}, \quad \forall \boldsymbol{h} \in \mathbb{R}^n\backslash\{\boldsymbol{0}\} \text{ and } \gamma \geq 2 \tag{44}$$

*with probability at least $1 - C\exp\left(-c_0\epsilon^2 m\right)$. This holds with the proviso $m/n \geq c_1 \cdot \epsilon^{-2}\log\epsilon^{-1}$.*

To connect this lemma with the last term of (38), we recognize that when $\gamma \leq \frac{\alpha_z^{\mathrm{lb}}\|\boldsymbol{z}\|}{\|\boldsymbol{h}\|}$, one has

$$\mathbf{1}_{\mathcal{E}_1^i \cap \mathcal{D}_\gamma^{i,2}} \quad \leq \quad \mathbf{1}_{\{|\boldsymbol{a}_i^\top \boldsymbol{h}| \geq \alpha_z^{\mathrm{lb}}\|\boldsymbol{z}\|\}}. \tag{45}$$

The constraint $\left|\frac{\boldsymbol{a}_i^\top \boldsymbol{h}}{\|\boldsymbol{h}\|} - \frac{2\boldsymbol{a}_i^\top \boldsymbol{z}}{\|\boldsymbol{h}\|}\right| \leq \gamma$ of $\mathcal{D}_\gamma^{i,2}$ necessarily requires

$$\frac{|\boldsymbol{a}_i^\top \boldsymbol{h}|}{\|\boldsymbol{h}\|} \geq \frac{2|\boldsymbol{a}_i^\top \boldsymbol{z}|}{\|\boldsymbol{h}\|} - \gamma \geq \frac{2\alpha_z^{\mathrm{lb}}\|\boldsymbol{z}\|}{\|\boldsymbol{h}\|} - \gamma \geq \frac{\alpha_z^{\mathrm{lb}}\|\boldsymbol{z}\|}{\|\boldsymbol{h}\|}, \tag{46}$$

where the last inequality comes from our assumption on $\gamma$. With Lemma 6 in place, (45) immediately gives

$$\begin{aligned}
\sum_{i=1}^m \mathbf{1}_{\mathcal{E}_1^i \cap \mathcal{D}_\gamma^{i,2}} &\leq \frac{\|\boldsymbol{h}\|}{0.49\alpha_z^{\mathrm{lb}}\|\boldsymbol{z}\|}\exp\left(-0.485\left(\frac{\alpha_z^{\mathrm{lb}}\|\boldsymbol{z}\|}{\|\boldsymbol{h}\|}\right)^2\right) + \frac{\epsilon\|\boldsymbol{h}\|^2}{(\alpha_z^{\mathrm{lb}})^2\|\boldsymbol{z}\|^2} \\
&\leq \frac{1}{9800}\left(\frac{\|\boldsymbol{h}\|}{\alpha_z^{\mathrm{lb}}\|\boldsymbol{z}\|}\right)^4 + \frac{\epsilon}{(\alpha_z^{\mathrm{lb}})^2}\left(\frac{\|\boldsymbol{h}\|}{\|\boldsymbol{z}\|}\right)^2
\end{aligned} \tag{47}$$

as long as $\frac{\|\boldsymbol{h}\|}{\|\boldsymbol{z}\|} \leq \frac{\alpha_z^{\mathrm{lb}}}{6}$, where the last inequality uses the majorization $\frac{1}{20000x^4} \geq \frac{1}{x}\exp\left(-0.485x^2\right)$ holding for any $x \geq 6$.

In addition, on $\mathcal{E}_1^i \cap \mathcal{D}_\gamma^{i,2}$, the amplitude of each summand can be bounded in such a way that

$$\frac{|\boldsymbol{a}_i^\top \boldsymbol{h}|^3}{|\boldsymbol{a}_i^\top \boldsymbol{z}|} \quad \leq \quad \frac{|2\boldsymbol{a}_i^\top \boldsymbol{z}| + \gamma\|\boldsymbol{h}\|}{|\boldsymbol{a}_i^\top \boldsymbol{z}|}\left(2\alpha_z^{\mathrm{ub}}\|\boldsymbol{z}\| + \gamma\|\boldsymbol{h}\|\right)^2 \tag{48}$$

$$\leq \quad \left(2 + \frac{\gamma}{\alpha_z^{\mathrm{lb}}}\frac{\|\boldsymbol{h}\|}{\|\boldsymbol{z}\|}\right)\left(2\alpha_z^{\mathrm{ub}} + \gamma\frac{\|\boldsymbol{h}\|}{\|\boldsymbol{z}\|}\right)^2\|\boldsymbol{z}\|^2, \tag{49}$$

where both inequalities are immediate consequences from the definitions of $\mathcal{D}_\gamma^{i,2}$ and $\mathcal{E}_1^i$ (see (29) and (16)). Taking this together with the cardinality bound (47) and picking $\epsilon$ appropriately, we get

$$\frac{1}{m}\sum_{i=1}^m \frac{|\boldsymbol{a}_i^\top \boldsymbol{h}|^3}{|\boldsymbol{a}_i^\top \boldsymbol{z}|}\mathbf{1}_{\mathcal{E}_1^i \cap \mathcal{D}_\gamma^{i,2}} \quad \leq \quad \left\{\underbrace{\frac{\left(2 + \frac{\gamma}{\alpha_z^{\mathrm{lb}}}\frac{\|\boldsymbol{h}\|}{\|\boldsymbol{z}\|}\right)\left(2\alpha_z^{\mathrm{ub}} + \gamma\frac{\|\boldsymbol{h}\|}{\|\boldsymbol{z}\|}\right)^2}{9800\,(\alpha_z^{\mathrm{lb}})^4}\frac{\|\boldsymbol{h}\|^2}{\|\boldsymbol{z}\|^2}}_{\vartheta_1} + \epsilon\right\}\|\boldsymbol{h}\|^2. \tag{50}$$

Furthermore, under the condition that

$$\gamma \le \alpha_z^{\mathrm{lb}} \frac{\|\boldsymbol{z}\|}{\|\boldsymbol{h}\|} \quad \text{and} \quad \frac{\|\boldsymbol{h}\|}{\|\boldsymbol{z}\|} \le \frac{\sqrt{98}\,(\alpha_z^{\mathrm{lb}})^2}{\sqrt{3}\,(2\alpha_z^{\mathrm{ub}} + \alpha_z^{\mathrm{lb}})},$$

one can simplify (50) by observing that $\vartheta_1 \le \frac{1}{100}$, which results in

$$\frac{1}{m} \sum_{i=1}^m \frac{\left|\boldsymbol{a}_i^\top \boldsymbol{h}\right|^3}{\left|\boldsymbol{a}_i^\top \boldsymbol{z}\right|} \mathbf{1}_{\mathcal{E}_1^i \cap \mathcal{D}_\gamma^{i,2}} \quad \le \quad \left(\frac{1}{100} + \epsilon\right) \|\boldsymbol{h}\|^2 . \tag{51}$$

Putting all preceding results in this subsection together reveals that with probability exceeding $1 - \exp\left(-\Omega\left(m\right)\right)$,

$$
\begin{aligned}
-\left\langle \boldsymbol{h}, \frac{1}{2m} \nabla \ell_{\mathrm{tr}}\left(\boldsymbol{z}\right) \right\rangle &\ge \left\{ 1.99 - 2\left(\zeta_1 + \zeta_2\right) - \sqrt{8/\pi} \frac{\|\boldsymbol{h}\|}{\alpha_z^{\mathrm{lb}} \|\boldsymbol{z}\|} - 3\epsilon \right\} \|\boldsymbol{h}\|^2 \\
&\ge \left\{ 1.99 - 2\left(\zeta_1 + \zeta_2\right) - \sqrt{8/\pi}(3\alpha_h)^{-1} - 3\epsilon \right\} \|\boldsymbol{h}\|^2
\end{aligned} \tag{52}
$$

holds simultaneously over all $\boldsymbol{x}$ and $\boldsymbol{z}$ satisfying

$$\frac{\|\boldsymbol{h}\|}{\|\boldsymbol{z}\|} \le \min\left\{ \frac{\alpha_z^{\mathrm{lb}}}{3\alpha_h}, \frac{\alpha_z^{\mathrm{lb}}}{6}, \frac{\sqrt{98/3}\,(\alpha_z^{\mathrm{lb}})^2}{2\alpha_z^{\mathrm{ub}} + \alpha_z^{\mathrm{lb}}}, \frac{1}{11} \right\} \tag{53}$$

as claimed in Proposition 2.

To conclude this section, we provide a tighter estimate about the norm of the truncated gradient.

**Lemma 7.** *Fix $\delta > 0$, and assume that $y_i = (\boldsymbol{a}_i^\top \boldsymbol{x})^2$. Suppose that $m \ge c_0 n$ for some large constant $c_0 > 0$. There exist some universal constants $c, C > 0$ such that with probability at least $1 - C \exp\left(-cm\right)$,*

$$\frac{1}{m} \left\| \nabla \ell_{\mathrm{tr}}\left(\boldsymbol{z}\right) \right\| \le (1 + \delta) \cdot 4\sqrt{1.02 + 0.665/\alpha_h} \, \|\boldsymbol{h}\| \tag{54}$$

*holds simultaneously for all $\boldsymbol{x}, \boldsymbol{z} \in \mathbb{R}^n$ satisfying $\frac{\|\boldsymbol{h}\|}{\|\boldsymbol{z}\|} \le \min\left\{ \frac{\alpha_z^{\mathrm{lb}}}{3\alpha_h}, \frac{\alpha_z^{\mathrm{lb}}}{6}, \frac{\sqrt{98/3}(\alpha_z^{\mathrm{lb}})^2}{2\alpha_z^{\mathrm{ub}} + \alpha_z^{\mathrm{lb}}}, \frac{1}{11} \right\}$.*

Lemma 7 complements the preceding arguments by allowing us to identify a concrete plausible range for the step size. Specifically, putting Lemma 7 and Proposition 2 together suggests that

$$-\left\langle \boldsymbol{h}, \frac{1}{m} \nabla \ell_{\mathrm{tr}}\left(\boldsymbol{z}\right) \right\rangle \ge \frac{2\left\{1.99 - 2\left(\zeta_1 + \zeta_2\right) - \sqrt{8/(9\pi)}\alpha_h^{-1} - \epsilon\right\}}{(1+\delta)^2 \cdot 16\left(1.02 + 0.665/\alpha_h\right)} \left\| \frac{1}{m} \nabla \ell_{\mathrm{tr}}\left(\boldsymbol{z}\right) \right\|^2 . \tag{55}$$

Taking $\epsilon$ and $\delta$ to be sufficiently small we arrive at a feasible range (cf. Definition (11))

$$\mu \le \frac{0.994 - \zeta_1 - \zeta_2 - \sqrt{2/(9\pi)}\alpha_h^{-1}}{2\left(1.02 + 0.665/\alpha_h\right)} := \mu_0. \tag{56}$$

This establishes Proposition 1 and in turn Theorem 1 when $\mu_t$ is taken to be a fixed constant.

To justify the contraction under backtracking line search, it suffices to prove that the resulting step size falls within this range (56), which we defer to Appendix D.

# 3   Stability

This section goes in the direction of establishing stability guarantees of TWF. We concentrate on the iterative gradient stage, and defer the analysis for the initialization stage to Appendix C.

Before continuing, we collect two bounds that we shall use several times. The first is the observation that

$$\frac{1}{m}\|\boldsymbol{y} - \mathcal{A}(\boldsymbol{z}\boldsymbol{z}^\top)\|_1 \le \frac{1}{m}\|\mathcal{A}(\boldsymbol{x}\boldsymbol{x}^\top - \boldsymbol{z}\boldsymbol{z}^\top)\|_1 + \frac{1}{m}\|\boldsymbol{\eta}\|_1 \lesssim \|\boldsymbol{h}\|\|\boldsymbol{z}\| + \frac{1}{m}\|\boldsymbol{\eta}\|_1 \lesssim \|\boldsymbol{h}\|\|\boldsymbol{z}\| + \frac{1}{\sqrt{m}}\|\boldsymbol{\eta}\|, \tag{57}$$

where the last inequality follows from Cauchy-Schwarz. Setting

$$v_i := 2 \frac{y_i - |\boldsymbol{a}_i^\top \boldsymbol{z}|^2}{\boldsymbol{a}_i^\top \boldsymbol{z}} \mathbf{1}_{\mathcal{E}_1^i \cap \mathcal{E}_2^i}$$

as usual, this inequality together with the truncation rules $\mathcal{E}_1^i$ and $\mathcal{E}_1^2$ give

$$
\begin{aligned}
& |v_i| \lesssim \|\boldsymbol{h}\| + \frac{\|\boldsymbol{\eta}\|}{\sqrt{m}\|\boldsymbol{z}\|} \\
\implies \quad \left\| \tfrac{1}{m} \nabla \ell_{\mathrm{tr}}(\boldsymbol{z}) \right\| \;=\; \tfrac{1}{m} \|\boldsymbol{A}^\top \boldsymbol{v}\| \;\le\; \left\| \tfrac{1}{\sqrt{m}} \boldsymbol{A} \right\| \tfrac{1}{\sqrt{m}} \|\boldsymbol{v}\| \;\overset{\text{(i)}}{\lesssim}\; \tfrac{1}{\sqrt{m}} \|\boldsymbol{v}\| \;\lesssim\; \|\boldsymbol{h}\| + \frac{\|\boldsymbol{\eta}\|}{\sqrt{m}\|\boldsymbol{z}\|},
\end{aligned}
\tag{58}
$$

where (i) arises from [3, Corollary 5.35].

As discussed before, the estimation error is contractive if $-\frac{1}{m}\nabla\ell_{\mathrm{tr}}(\boldsymbol{z})$ satisfies the regularity condition. With (58) in place, RC reduces to

$$-\frac{1}{m}\left\langle \nabla\ell_{\mathrm{tr}}(\boldsymbol{z}), \boldsymbol{h}\right\rangle \;\gtrsim\; \|\boldsymbol{h}\|^2. \tag{59}$$

Unfortunately, (59) does not hold for all $\boldsymbol{z}$ within the neighborhood of $\boldsymbol{x}$ due to the existence of noise. Instead we establish the following:

- The condition (59) holds for all $\boldsymbol{h}$ obeying

$$c_3 \frac{\|\boldsymbol{\eta}\|/\sqrt{m}}{\|\boldsymbol{z}\|} \le \|\boldsymbol{h}\| \le c_4\|\boldsymbol{x}\| \tag{60}$$

  for some constants $c_3, c_4 > 0$ (we shall call it *Regime 1*); this will be proved later. In this regime, the reasoning before gives

$$\mathrm{dist}\left(\boldsymbol{z} + \frac{\mu}{m}\nabla\ell_{\mathrm{tr}}(\boldsymbol{z}), \ \boldsymbol{x}\right) \le (1 - \rho)\mathrm{dist}(\boldsymbol{z}, \boldsymbol{x}) \tag{61}$$

  for some appropriate constants $\mu, \rho > 0$ and, hence, error contraction occurs as in the noiseless setting.

- However, once the iterate enters *Regime 2* where

$$\|\boldsymbol{h}\| \le \frac{c_3\|\boldsymbol{\eta}\|}{\sqrt{m}\|\boldsymbol{z}\|} \tag{62}$$

  the estimation error might no longer be contractive. Fortunately, in this regime each move by $\frac{\mu}{m}\nabla\ell_{\mathrm{tr}}(\boldsymbol{z})$ is of size at most $\mathcal{O}(\frac{\|\boldsymbol{\eta}\|}{\sqrt{m}\|\boldsymbol{z}\|})$, compare (58). As a result, at each iteration the estimation error cannot increase by more than a numerical constant times $\frac{\|\boldsymbol{\eta}\|}{\sqrt{m}\|\boldsymbol{z}\|}$ before possibly jumping out (of this regime). Therefore,

$$\mathrm{dist}\left(\boldsymbol{z} + \frac{\mu}{m}\nabla\ell_{\mathrm{tr}}(\boldsymbol{z}), \ \boldsymbol{x}\right) \le c_5 \frac{\|\boldsymbol{\eta}\|}{\sqrt{m}\|\boldsymbol{x}\|} \tag{63}$$

  for some constant $c_5 > 0$. Moreover, as long as $\|\boldsymbol{\eta}\|_\infty/\|\boldsymbol{x}\|^2$ is sufficiently small, one can guarantee that $c_5 \frac{\|\boldsymbol{\eta}\|}{\sqrt{m}\|\boldsymbol{x}\|} \le c_5 \frac{\|\boldsymbol{\eta}\|_\infty}{\|\boldsymbol{x}\|} \le c_4\|\boldsymbol{x}\|$. In other words, if the iterate jumps out of Regime 2, it will still fall within Regime 1.

To summarize, suppose the initial guess $\boldsymbol{z}^{(0)}$ obeys $\mathrm{dist}(\boldsymbol{z}^{(0)}, \boldsymbol{x}) \le c_4\|\boldsymbol{x}\|$. Then the estimation error will shrink at a geometric rate $1 - \rho$ before it enters Regime 2. Afterwards, $\boldsymbol{z}^{(t)}$ will either stay within Regime 2 or jump back and forth between Regimes 1 and 2. Because of the bounds (63) and (61), the estimation errors will never exceed the order of $\frac{\|\boldsymbol{\eta}\|}{\sqrt{m}\|\boldsymbol{x}\|}$ from then on. Putting these together establishes (6), namely, the first part of the theorem.

Below we justify the condition (59) for Regime 1, for which we start by gathering additional properties of the truncation rules. By Cauchy-Schwartz, $\frac{1}{m}\|\boldsymbol{\eta}\|_1 \le \frac{1}{\sqrt{m}}\|\boldsymbol{\eta}\| \le \frac{1}{c_3}\|\boldsymbol{h}\|\|\boldsymbol{z}\|$. When $c_3$ is sufficiently large, applying Lemmas 1 and 2 gives

$$
\begin{aligned}
\tfrac{1}{m}\sum_{l=1}^m \left| y_l - |\boldsymbol{a}_l^\top\boldsymbol{z}|^2 \right| &\le \tfrac{1}{m}\left\|\mathcal{A}\left(\boldsymbol{x}\boldsymbol{x}^\top - \boldsymbol{z}\boldsymbol{z}^\top\right)\right\|_1 + \tfrac{1}{m}\|\boldsymbol{\eta}\|_1 \le 2.98\|\boldsymbol{h}\|\|\boldsymbol{z}\|; \\
\tfrac{1}{m}\sum_{l=1}^m \left| y_l - |\boldsymbol{a}_l^\top\boldsymbol{z}|^2 \right| &\ge \tfrac{1}{m}\left\|\mathcal{A}\left(\boldsymbol{x}\boldsymbol{x}^\top - \boldsymbol{z}\boldsymbol{z}^\top\right)\right\|_1 - \tfrac{1}{m}\|\boldsymbol{\eta}\|_1 \ge 1.151\|\boldsymbol{h}\|\|\boldsymbol{z}\|.
\end{aligned}
\tag{64}
$$

From now on, we shall denote $\tilde{\mathcal{E}}_2^i := \left\{ \left| |\boldsymbol{a}_i^\top \boldsymbol{x}|^2 - |\boldsymbol{a}_i^\top \boldsymbol{z}|^2 \right| \leq \frac{\alpha_h}{m} \left\| \boldsymbol{y} - \mathcal{A} \left( \boldsymbol{z}\boldsymbol{z}^\top \right) \right\|_1 \frac{|\boldsymbol{a}_i^\top \boldsymbol{z}|}{\|\boldsymbol{z}\|} \right\}$ to differentiate from $\mathcal{E}_2^i$. For any small constant $\epsilon > 0$, we introduce the index set $\mathcal{G} := \{ i : |\eta_i| \leq C_\epsilon \|\boldsymbol{\eta}\| / \sqrt{m} \}$ that satisfies $|\mathcal{G}| = (1-\epsilon)m$. Note that $C_\epsilon$ must be bounded as $n$ scales, since

$$\|\boldsymbol{\eta}\|^2 \;\geq\; \sum_{i \notin \mathcal{G}} \eta_i^2 \;\geq\; (m - |\mathcal{G}|) \cdot C_\epsilon^2 \|\boldsymbol{\eta}\|^2 / m \;\geq\; \epsilon C_\epsilon^2 \|\boldsymbol{\eta}\|^2 \quad \Rightarrow \quad C_\epsilon \leq 1/\sqrt{\epsilon}. \tag{65}$$

We are now ready to analyze the truncated gradient, which we separate into several components as follows

$$\nabla_{\mathrm{tr}}\ell(\boldsymbol{z}) \;=\; \underbrace{2\sum_{i \in \mathcal{G}} \frac{|\boldsymbol{a}_i^\top \boldsymbol{x}|^2 - |\boldsymbol{a}_i^\top \boldsymbol{z}|^2}{\boldsymbol{a}_i^\top \boldsymbol{z}} \boldsymbol{a}_i \mathbf{1}_{\mathcal{E}_1^i \cap \mathcal{E}_2^i} + 2\sum_{i \notin \mathcal{G}} \frac{|\boldsymbol{a}_i^\top \boldsymbol{x}|^2 - |\boldsymbol{a}_i^\top \boldsymbol{z}|^2}{\boldsymbol{a}_i^\top \boldsymbol{z}} \boldsymbol{a}_i \mathbf{1}_{\mathcal{E}_1^i \cap \tilde{\mathcal{E}}_2^i}}_{:= \nabla_{\mathrm{tr}}^{\mathrm{clean}}\ell(\boldsymbol{z})}$$

$$+ \underbrace{2\sum_{i \in \mathcal{G}} \frac{\eta_i}{\boldsymbol{a}_i^\top \boldsymbol{z}} \boldsymbol{a}_i \mathbf{1}_{\mathcal{E}_1^i \cap \mathcal{E}_2^i}}_{:= \nabla_{\mathrm{tr}}^{\mathrm{noise}}\ell(\boldsymbol{z})} + \underbrace{2\sum_{i \notin \mathcal{G}} \left( \frac{y_i - |\boldsymbol{a}_i^\top \boldsymbol{z}|^2}{\boldsymbol{a}_i^\top \boldsymbol{z}} \mathbf{1}_{\mathcal{E}_1^i \cap \mathcal{E}_2^i} - \frac{|\boldsymbol{a}_i^\top \boldsymbol{x}|^2 - |\boldsymbol{a}_i^\top \boldsymbol{z}|^2}{\boldsymbol{a}_i^\top \boldsymbol{z}} \mathbf{1}_{\mathcal{E}_1^i \cap \tilde{\mathcal{E}}_2^i} \right) \boldsymbol{a}_i}_{:= \nabla_{\mathrm{tr}}^{\mathrm{extra}}\ell(\boldsymbol{z})}. \tag{66}$$

- For each index $i \in \mathcal{G}$, the inclusion property (24) (i.e. $\mathcal{E}_3^i \subseteq \mathcal{E}_2^i \subseteq \mathcal{E}_4^i$) holds. To see this, observe that

$$\left| y_i - |\boldsymbol{a}_i^\top \boldsymbol{z}|^2 \right| \in \left| |\boldsymbol{a}_i^\top \boldsymbol{x}|^2 - |\boldsymbol{a}_i^\top \boldsymbol{z}|^2 \right| \pm |\eta_i|.$$

Since $|\eta_i| \leq C_\epsilon \|\boldsymbol{\eta}\| / \sqrt{m} \ll \|\boldsymbol{h}\|\|\boldsymbol{z}\|$ when $c_3$ is sufficiently large, one can derive the inclusion (24) immediately from (64). As a result, all the proof arguments for Proposition 2 carry over to $\nabla_{\mathrm{tr}}^{\mathrm{clean}}\ell(\boldsymbol{z})$, suggesting that

$$- \left\langle \boldsymbol{h}, \frac{1}{m} \nabla_{\mathrm{tr}}^{\mathrm{clean}}\ell(\boldsymbol{z}) \right\rangle \;\geq\; 2\left\{ 1.99 - 2(\zeta_1 + \zeta_2) - \sqrt{8/(9\pi)}\alpha_h^{-1} - \epsilon \right\} \|\boldsymbol{h}\|^2. \tag{67}$$

- Next, letting $w_i = \frac{2\eta_i}{\boldsymbol{a}_i^\top \boldsymbol{z}} \mathbf{1}_{\mathcal{E}_1^i \cap \mathcal{E}_2^i} \mathbf{1}_{\{i \in \mathcal{G}\}}$, we see that for any constant $\delta > 0$, the noise component obeys

$$\left\| \frac{1}{m} \nabla_{\mathrm{tr}}^{\mathrm{noise}}\ell(\boldsymbol{z}) \right\| \;=\; \left\| \frac{1}{m} \boldsymbol{A}^\top \boldsymbol{w} \right\| \;\leq\; \left\| \frac{1}{\sqrt{m}} \boldsymbol{A} \right\| \left\| \frac{1}{\sqrt{m}} \boldsymbol{w} \right\| \;\overset{(ii)}{\leq}\; \frac{1+\delta}{\sqrt{m}} \|\boldsymbol{w}\| \;\leq\; (1+\delta)\frac{2\|\boldsymbol{\eta}\|/\sqrt{m}}{\alpha_z^{\mathrm{lb}} \|\boldsymbol{z}\|}, \tag{68}$$

when $m/n$ is sufficiently large. Here, (ii) arises from [3, Corollary 5.35], and the last inequality is a consequence of the upper estimate

$$\|\boldsymbol{w}\|^2 \;\leq\; 4\sum_{i=1}^m \frac{|\eta_i|^2}{(\boldsymbol{a}_i^\top \boldsymbol{z})^2} \mathbf{1}_{\mathcal{E}_1^i \cap \mathcal{E}_2^i} \;\leq\; 4\sum_{i=1}^m \frac{|\eta_i|^2}{(\alpha_z^{\mathrm{lb}} \|\boldsymbol{z}\|)^2} \;=\; \frac{4\|\boldsymbol{\eta}\|^2}{(\alpha_z^{\mathrm{lb}} \|\boldsymbol{z}\|)^2}. \tag{69}$$

In turn, this immediately gives

$$\left| \left\langle \boldsymbol{h}, \frac{1}{m} \nabla_{\mathrm{tr}}^{\mathrm{noise}}\ell(\boldsymbol{z}) \right\rangle \right| \;\leq\; \|\boldsymbol{h}\| \left\| \frac{1}{m} \nabla_{\mathrm{tr}}^{\mathrm{noise}}\ell(\boldsymbol{z}) \right\| \;\leq\; \frac{2(1+\delta)}{\alpha_z^{\mathrm{lb}}} \frac{\|\boldsymbol{\eta}\|}{\sqrt{m}\|\boldsymbol{z}\|} \|\boldsymbol{h}\|. \tag{70}$$

- We now turn to the last term $\nabla_{\mathrm{tr}}^{\mathrm{extra}}\ell(\boldsymbol{z})$. According to the definition of $\mathcal{E}_2^i$ and $\tilde{\mathcal{E}}_2^i$ as well as the property (64), the weight $q_i := 2\left( \frac{y_i - |\boldsymbol{a}_i^\top \boldsymbol{z}|^2}{\boldsymbol{a}_i^\top \boldsymbol{z}} \mathbf{1}_{\mathcal{E}_1^i \cap \mathcal{E}_2^i} - \frac{|\boldsymbol{a}_i^\top \boldsymbol{x}|^2 - |\boldsymbol{a}_i^\top \boldsymbol{z}|^2}{\boldsymbol{a}_i^\top \boldsymbol{z}} \mathbf{1}_{\mathcal{E}_1^i \cap \tilde{\mathcal{E}}_2^i} \right) \mathbf{1}_{\{i \notin \mathcal{G}\}}$ is bounded in magnitude by $6\|\boldsymbol{h}\|$. This gives

$$\|\boldsymbol{q}\| \leq \sqrt{m - |\mathcal{G}|} \cdot 6\|\boldsymbol{h}\| \leq 6\sqrt{\epsilon m}\|\boldsymbol{h}\|,$$

$$\Rightarrow \quad \left| \left\langle \frac{1}{m} \nabla_{\mathrm{tr}}^{\mathrm{extra}}\ell(\boldsymbol{z}), \boldsymbol{h} \right\rangle \right| \leq \|\boldsymbol{h}\| \cdot \left\| \frac{1}{m} \nabla_{\mathrm{tr}}^{\mathrm{extra}}\ell(\boldsymbol{z}) \right\| = \frac{1}{m} \|\boldsymbol{h}\| \cdot \left\| \boldsymbol{A}^\top \boldsymbol{q} \right\| \leq 6(1+\delta)\sqrt{\epsilon}\|\boldsymbol{h}\|^2. \tag{71}$$

Taking the above bounds together yields

$$-\frac{1}{m}\left\langle\nabla\ell_{\mathrm{tr}}\left(\boldsymbol{z}\right),\boldsymbol{h}\right\rangle \;\geq\; 2\left\{1.99-2\left(\zeta_1+\zeta_2\right)-\sqrt{\frac{8}{9\pi}}\frac{1}{\alpha_h}-6(1+\delta)\sqrt{\epsilon}-\epsilon\right\}\|\boldsymbol{h}\|^2-\frac{2\left(1+\delta\right)}{\alpha_z^{\mathrm{lb}}}\frac{\|\boldsymbol{\eta}\|}{\sqrt{m}\,\|\boldsymbol{z}\|}\|\boldsymbol{h}\|.$$

Since $\|\boldsymbol{h}\|\geq c_3\frac{\|\boldsymbol{\eta}\|}{\sqrt{m}\|\boldsymbol{z}\|}$ for some large constant $c_3>0$, setting $\epsilon$ to be small one obtains

$$-\frac{1}{m}\left\langle\nabla\ell_{\mathrm{tr}}\left(\boldsymbol{z}\right),\boldsymbol{h}\right\rangle \;\geq\; 2\left\{1.95-2\left(\zeta_1+\zeta_2\right)-\sqrt{8/(9\pi)}\alpha_h^{-1}\right\}\|\boldsymbol{h}\|^2 \tag{72}$$

for all $\boldsymbol{h}$ obeying

$$\frac{c_3\|\boldsymbol{\eta}\|/\sqrt{m}}{\|\boldsymbol{z}\|} \;\leq\; \|\boldsymbol{h}\| \;\leq\; \min\left\{\frac{1}{11},\frac{\alpha_z^{\mathrm{lb}}}{3\alpha_h},\frac{\alpha_z^{\mathrm{lb}}}{6},\frac{\sqrt{98/3}\left(\alpha_z^{\mathrm{lb}}\right)^2}{2\alpha_z^{\mathrm{ub}}+\alpha_z^{\mathrm{lb}}}\right\}\|\boldsymbol{z}\|,$$

which finishes the proof of Theorem 2 for general $\boldsymbol{\eta}$.

Up until now, we have established the theorem for general $\boldsymbol{\eta}$, and it remains to specialize it to the Poisson model. Standard concentration results, which we omit, give

$$\frac{1}{m}\|\boldsymbol{\eta}\|^2 \approx \frac{1}{m}\sum_{i=1}^m \mathbb{E}\left[\eta_i^2\right] = \frac{1}{m}\sum_{i=1}^m \left(\boldsymbol{a}_i^\top\boldsymbol{x}\right)^2 \approx \|\boldsymbol{x}\|^2. \tag{73}$$

Substitution into (6) completes the proof.

# 4    Minimax lower bound

The goal of this section is to establish the minimax lower bound given in Theorem 3. For notational simplicity, we denote by $\mathbb{P}\left(\boldsymbol{y}\mid\boldsymbol{w}\right)$ the likelihood of $y_i\overset{\mathrm{ind.}}{\sim}\mathsf{Poisson}(|\boldsymbol{a}_i^\top\boldsymbol{w}|^2)$, $1\leq i\leq m$ conditional on $\{\boldsymbol{a}_i\}$. For any two probability measures $P$ and $Q$, we denote by $\mathsf{KL}\left(P\|Q\right)$ the Kullback–Leibler (KL) divergence between them:

$$\mathsf{KL}\left(P\|Q\right) := \int\log\left(\frac{\mathrm{d}P}{\mathrm{d}Q}\right)\mathrm{d}P, \tag{74}$$

The basic idea is to adopt the general reduction scheme discussed in [4, Section 2.2], which amounts to finding a finite collection of hypotheses that are minimally separated. Below we gather one result useful for constructing and analyzing such hypotheses.

**Lemma 8.** *Suppose that $\boldsymbol{a}_i\sim\mathcal{N}\left(\boldsymbol{0},\boldsymbol{I}_n\right)$, $n$ is sufficiently large, and $m=\kappa n$ for some sufficiently large constant $\kappa>0$. Consider any $\boldsymbol{x}\in\mathbb{R}^n\backslash\{\boldsymbol{0}\}$. On an event $\mathcal{B}$ of probability approaching one, there exists a collection $\mathcal{M}$ of $M=\exp\left(n/30\right)$ distinct vectors obeying the following properties:*

*(i) $\boldsymbol{x}\in\mathcal{M}$;*

*(ii) for all $\boldsymbol{w}^{(l)},\boldsymbol{w}^{(j)}\in\mathcal{M}$,*

$$1/\sqrt{8}-(2n)^{-1/2}\leq\left\|\boldsymbol{w}^{(l)}-\boldsymbol{w}^{(j)}\right\|\leq 3/2+n^{-1/2}; \tag{75}$$

*(iii) for all $\boldsymbol{w}\in\mathcal{M}$,*

$$\frac{|\boldsymbol{a}_i^\top\left(\boldsymbol{w}-\boldsymbol{x}\right)|^2}{|\boldsymbol{a}_i^\top\boldsymbol{x}|^2} \leq \frac{\|\boldsymbol{w}-\boldsymbol{x}\|^2}{\|\boldsymbol{x}\|^2}\{2+17\log^3 m\}, \quad 1\leq i\leq m. \tag{76}$$

In words, Lemma 8 constructs a set $\mathcal{M}$ of exponentially many vectors/hypotheses scattered around $\boldsymbol{x}$ and yet well separated. From (ii) we see that each pair of hypotheses in $\mathcal{M}$ is separated by a distance roughly on the order of 1, and all hypotheses reside within a spherical ball centered at $\boldsymbol{x}$ of radius $3/2+o(1)$. When $\|\boldsymbol{x}\|\geq\log^{1.5}m$, every hypothesis $\boldsymbol{w}\in\mathcal{M}$ satisfies $\|\boldsymbol{w}\|\approx\|\boldsymbol{x}\|\gg 1$. In addition, (iii) says that the

quantities $|\boldsymbol{a}_i^\top (\boldsymbol{w} - \boldsymbol{x})| / |\boldsymbol{a}_i^\top \boldsymbol{x}|$ are all very well controlled (modulo some logarithmic factor). In particular, when $\|\boldsymbol{x}\| \geq \log^{1.5} m$, one must have

$$\frac{|\boldsymbol{a}_i^\top (\boldsymbol{w} - \boldsymbol{x})|^2}{|\boldsymbol{a}_i^\top \boldsymbol{x}|^2} \lesssim \frac{\|\boldsymbol{w} - \boldsymbol{x}\|^2}{\|\boldsymbol{x}\|^2} \log^3 m \lesssim \frac{1}{\log^3 m} \log^3 m \lesssim 1. \tag{77}$$

In the Poisson model, such a quantity turns out to be crucial in controlling the information divergence between two hypotheses, as demonstrated in the following lemma.

**Lemma 9.** *Fix a family of design vectors* $\{\boldsymbol{a}_i\}$. *Then for any* $\boldsymbol{w}$ *and* $\boldsymbol{r} \in \mathbb{R}^n$,

$$\mathsf{KL}\big(\,\mathbb{P}\,(\boldsymbol{y} \mid \boldsymbol{w} + \boldsymbol{r}) \parallel \mathbb{P}\,(\boldsymbol{y} \mid \boldsymbol{w})\,\big) \;\leq\; \sum_{i=1}^m |\boldsymbol{a}_i^\top \boldsymbol{r}|^2 \Big( 8 + \frac{2|\boldsymbol{a}_i^\top \boldsymbol{r}|^2}{|\boldsymbol{a}_i^\top \boldsymbol{w}|^2} \Big). \tag{78}$$

Lemma 9 and (77) taken collectively suggest that on the event $\mathcal{B} \cap \mathcal{C}$ ($\mathcal{B}$ is in Lemma 8 and $\mathcal{C} := \{\|\boldsymbol{A}\| \leq \sqrt{2m}\}$), the conditional KL divergence (we condition on the $\boldsymbol{a}_i$'s) obeys

$$\mathsf{KL}\big(\,\mathbb{P}\,(\boldsymbol{y} \mid \boldsymbol{w}) \parallel \mathbb{P}\,(\boldsymbol{y} \mid \boldsymbol{x})\,\big) \leq c_3 \sum_{i=1}^m \big|\boldsymbol{a}_i^\top (\boldsymbol{w} - \boldsymbol{x})\big|^2 \leq 2c_3 m \|\boldsymbol{w} - \boldsymbol{x}\|^2, \quad \forall \boldsymbol{w} \in \mathcal{M}; \tag{79}$$

here, the inequality holds for some constant $c_3 > 0$ provided that $\|\boldsymbol{x}\| \geq \log^{1.5} m$, and the last inequality is a result of $\mathcal{C}$ (which occurs with high probability). We now use hypotheses as in Lemma 8 but rescaled in such a way that

$$\|\boldsymbol{w} - \boldsymbol{x}\| \asymp \delta, \quad \text{and} \quad \|\boldsymbol{w} - \tilde{\boldsymbol{w}}\| \asymp \delta, \quad \forall \boldsymbol{w}, \tilde{\boldsymbol{w}} \in \mathcal{M} \text{ with } \boldsymbol{w} \neq \tilde{\boldsymbol{w}}. \tag{80}$$

for some $0 < \delta < 1$. This is achieved via the substitution $\boldsymbol{w} \longleftarrow \boldsymbol{x} + \delta(\boldsymbol{w} - \boldsymbol{x})$; with a slight abuse of notation, $\mathcal{M}$ denotes the new set.

The hardness of a minimax estimation problem is known to be dictated by information divergence inequalities such as (79). Indeed, suppose that

$$\frac{1}{M-1} \sum_{\boldsymbol{w} \in \mathcal{M} \backslash \{\boldsymbol{x}\}} \mathsf{KL}\big(\,\mathbb{P}\,(\boldsymbol{y} \mid \boldsymbol{w}) \parallel \mathbb{P}\,(\boldsymbol{y} \mid \boldsymbol{x})\,\big) \leq \frac{1}{10} \log (M - 1) \tag{81}$$

holds, then the Fano-type minimax lower bound [4, Theorem 2.7] asserts that

$$\inf_{\hat{\boldsymbol{x}}} \sup_{\boldsymbol{x} \in \mathcal{M}} \mathbb{E}\big[\|\hat{\boldsymbol{x}} - \boldsymbol{x}\| \mid \{\boldsymbol{a}_i\}\big] \gtrsim \min_{\boldsymbol{w}, \tilde{\boldsymbol{w}} \in \mathcal{M}, \boldsymbol{w} \neq \tilde{\boldsymbol{w}}} \|\boldsymbol{w} - \tilde{\boldsymbol{w}}\|. \tag{82}$$

Since $M = \exp(n/30)$, (81) would follow from

$$2c_3 \|\boldsymbol{w} - \boldsymbol{x}\|^2 \leq n/(300m). \quad \boldsymbol{w} \in \mathcal{M}. \tag{83}$$

Hence, we just need to select $\delta$ to be a small multiple of $\sqrt{n/m}$. This in turn gives

$$\inf_{\hat{\boldsymbol{x}}} \sup_{\boldsymbol{x} \in \mathcal{M}} \mathbb{E}\big[\|\hat{\boldsymbol{x}} - \boldsymbol{x}\| \mid \{\boldsymbol{a}_i\}\big] \gtrsim \min_{\boldsymbol{w}, \tilde{\boldsymbol{w}} \in \mathcal{M}, \boldsymbol{w} \neq \tilde{\boldsymbol{w}}} \|\boldsymbol{w} - \tilde{\boldsymbol{w}}\| \gtrsim \sqrt{n/m}. \tag{84}$$

Finally, it remains to connect $\|\hat{\boldsymbol{x}} - \boldsymbol{x}\|$ with $\mathrm{dist}\,(\hat{\boldsymbol{x}}, \boldsymbol{x})$. Since all the $\boldsymbol{w} \in \mathcal{M}$ are clustered around $\boldsymbol{x}$ and are at a mutual distance about $\delta$ that is much smaller than $\|\boldsymbol{x}\|$, we can see that for any reasonable estimator, $\mathrm{dist}(\hat{\boldsymbol{x}}, \boldsymbol{x}) = \|\hat{\boldsymbol{x}} - \boldsymbol{x}\|$. This finishes the proof.

# A  Proofs for Section 2

## A.1  Proof of Lemma 3

First, we make the observation that $(\boldsymbol{a}_i^\top \boldsymbol{z})^2 - (\boldsymbol{a}_i^\top \boldsymbol{x})^2 = \big(2\boldsymbol{a}_i^\top \boldsymbol{z} - \boldsymbol{a}_i^\top \boldsymbol{h}\big) \boldsymbol{a}_i^\top \boldsymbol{h}$ is a quadratic function in $\boldsymbol{a}_i^\top \boldsymbol{h}$. If we assume $\gamma \leq \frac{\alpha_z^{\mathrm{lb}} \|\boldsymbol{z}\|}{\|\boldsymbol{h}\|}$, then on the event $\mathcal{E}_1^i$ one has

$$(\boldsymbol{a}_i^\top \boldsymbol{z})^2 \;\geq\; \alpha_z^{\mathrm{lb}} \|\boldsymbol{z}\| \cdot |\boldsymbol{a}_i^\top \boldsymbol{z}| \;\geq\; \gamma \|\boldsymbol{h}\| \, |\boldsymbol{a}_i^\top \boldsymbol{z}|. \tag{85}$$

Solving the quadratic inequality that specifies $\mathcal{D}_\gamma^i$ gives

$$
\boldsymbol{a}_i^\top \boldsymbol{h} \;\in\; \left[ \boldsymbol{a}_i^\top \boldsymbol{z} - \sqrt{\left(\boldsymbol{a}_i^\top \boldsymbol{z}\right)^2 + \gamma \left\| \boldsymbol{h} \right\| \left| \boldsymbol{a}_i^\top \boldsymbol{z} \right|}, \; \boldsymbol{a}_i^\top \boldsymbol{z} - \sqrt{\left(\boldsymbol{a}_i^\top \boldsymbol{z}\right)^2 - \gamma \left\| \boldsymbol{h} \right\| \left| \boldsymbol{a}_i^\top \boldsymbol{z} \right|} \right],
$$

$$
\text{or} \quad \boldsymbol{a}_i^\top \boldsymbol{h} \;\in\; \left[ \boldsymbol{a}_i^\top \boldsymbol{z} + \sqrt{\left(\boldsymbol{a}_i^\top \boldsymbol{z}\right)^2 - \gamma \left\| \boldsymbol{h} \right\| \left| \boldsymbol{a}_i^\top \boldsymbol{z} \right|}, \; \boldsymbol{a}_i^\top \boldsymbol{z} + \sqrt{\left(\boldsymbol{a}_i^\top \boldsymbol{z}\right)^2 + \gamma \left\| \boldsymbol{h} \right\| \left| \boldsymbol{a}_i^\top \boldsymbol{z} \right|} \right],
$$

which we will simplify in the sequel.

Suppose for the moment that $\boldsymbol{a}_i^\top \boldsymbol{z} \geq 0$, then the preceding two intervals are respectively equivalent to

$$
\boldsymbol{a}_i^\top \boldsymbol{h} \;\in\; \left[ \frac{-\gamma \left\| \boldsymbol{h} \right\| \left| \boldsymbol{a}_i^\top \boldsymbol{z} \right|}{\boldsymbol{a}_i^\top \boldsymbol{z} + \sqrt{\left(\boldsymbol{a}_i^\top \boldsymbol{z}\right)^2 + \gamma \left\| \boldsymbol{h} \right\| \left| \boldsymbol{a}_i^\top \boldsymbol{z} \right|}}, \; \frac{\gamma \left\| \boldsymbol{h} \right\| \left| \boldsymbol{a}_i^\top \boldsymbol{z} \right|}{\boldsymbol{a}_i^\top \boldsymbol{z} + \sqrt{\left(\boldsymbol{a}_i^\top \boldsymbol{z}\right)^2 - \gamma \left\| \boldsymbol{h} \right\| \left| \boldsymbol{a}_i^\top \boldsymbol{z} \right|}} \right] := I_1;
$$

$$
\boldsymbol{a}_i^\top \boldsymbol{h} - 2\boldsymbol{a}_i^\top \boldsymbol{z} \;\in\; \left[ \frac{-\gamma \left\| \boldsymbol{h} \right\| \left| \boldsymbol{a}_i^\top \boldsymbol{z} \right|}{\boldsymbol{a}_i^\top \boldsymbol{z} + \sqrt{\left(\boldsymbol{a}_i^\top \boldsymbol{z}\right)^2 - \gamma \left\| \boldsymbol{h} \right\| \left| \boldsymbol{a}_i^\top \boldsymbol{z} \right|}}, \; \frac{\gamma \left\| \boldsymbol{h} \right\| \left| \boldsymbol{a}_i^\top \boldsymbol{z} \right|}{\boldsymbol{a}_i^\top \boldsymbol{z} + \sqrt{\left(\boldsymbol{a}_i^\top \boldsymbol{z}\right)^2 + \gamma \left\| \boldsymbol{h} \right\| \left| \boldsymbol{a}_i^\top \boldsymbol{z} \right|}} \right] := I_2.
$$

Assuming (85) and making use of the observations

$$
\frac{\gamma \left\| \boldsymbol{h} \right\| \left| \boldsymbol{a}_i^\top \boldsymbol{z} \right|}{\boldsymbol{a}_i^\top \boldsymbol{z} + \sqrt{\left(\boldsymbol{a}_i^\top \boldsymbol{z}\right)^2 - \gamma \left\| \boldsymbol{h} \right\| \left| \boldsymbol{a}_i^\top \boldsymbol{z} \right|}} \;\leq\; \frac{\gamma \left\| \boldsymbol{h} \right\| \left| \boldsymbol{a}_i^\top \boldsymbol{z} \right|}{\boldsymbol{a}_i^\top \boldsymbol{z}} = \gamma \left\| \boldsymbol{h} \right\|
$$

$$
\text{and} \quad \frac{\gamma \left\| \boldsymbol{h} \right\| \left| \boldsymbol{a}_i^\top \boldsymbol{z} \right|}{\boldsymbol{a}_i^\top \boldsymbol{z} + \sqrt{\left(\boldsymbol{a}_i^\top \boldsymbol{z}\right)^2 + \gamma \left\| \boldsymbol{h} \right\| \left| \boldsymbol{a}_i^\top \boldsymbol{z} \right|}} \;\geq\; \frac{\gamma \left\| \boldsymbol{h} \right\| \left| \boldsymbol{a}_i^\top \boldsymbol{z} \right|}{\left(1 + \sqrt{2}\right) \left| \boldsymbol{a}_i^\top \boldsymbol{z} \right|} = \frac{\gamma}{1 + \sqrt{2}} \left\| \boldsymbol{h} \right\|,
$$

we obtain the inner and outer bounds

$$
\left[ \pm \left(1 + \sqrt{2}\right)^{-1} \gamma \left\| \boldsymbol{h} \right\| \right] \;\subseteq\; I_1, I_2 \;\subseteq\; \left[ \pm \gamma \left\| \boldsymbol{h} \right\| \right].
$$

Setting $\gamma_1 := \frac{\gamma}{1+\sqrt{2}}$ gives

$$
\left( \mathcal{D}_{\gamma_1}^{i,1} \cap \mathcal{E}_{i,1} \right) \cup \left( \mathcal{D}_{\gamma_1}^{i,2} \cap \mathcal{E}_{i,1} \right) \;\subseteq\; \mathcal{D}_\gamma \cap \mathcal{E}_{i,1} \;\subseteq\; \left( \mathcal{D}_\gamma^{i,1} \cap \mathcal{E}_{i,1} \right) \cup \left( \mathcal{D}_\gamma^{i,2} \cap \mathcal{E}_{i,1} \right).
$$

Proceeding with the same argument, we can derive exactly the same inner and outer bounds in the regime where $\boldsymbol{a}_i^\top \boldsymbol{z} < 0$, concluding the proof.

## A.2 Proof of Lemma 4

By homogeneity, it suffices to establish the claim for the case where both $\boldsymbol{h}$ and $\boldsymbol{z}$ are *unit vectors*.

Suppose for the moment that $\boldsymbol{h}$ and $\boldsymbol{z}$ are *statistically independent* from $\{\boldsymbol{a}_i\}$. We introduce two auxiliary Lipschitz functions approximating indicator functions:

$$
\chi_z(\tau) \;:=\; \begin{cases} 1, & \text{if } |\tau| \in \left[ \sqrt{1.01}\alpha_z^{\mathrm{lb}}, \sqrt{0.99}\alpha_z^{\mathrm{ub}} \right]; \\ -100 \left(\alpha_z^{\mathrm{ub}}\right)^{-2} \tau^2 + 100, & \text{if } |\tau| \in \left[ \sqrt{0.99}\alpha_z^{\mathrm{ub}}, \alpha_z^{\mathrm{ub}} \right]; \\ 100 \left(\alpha_z^{\mathrm{lb}}\right)^{-2} \tau^2 - 100, & \text{if } |\tau| \in \left[ \alpha_z^{\mathrm{lb}}, \sqrt{1.01}\alpha_z^{\mathrm{lb}} \right]; \\ 0, & \text{else.} \end{cases} \tag{86}
$$

$$
\chi_h(\tau) \;:=\; \begin{cases} 1, & \text{if } |\tau| \in \left[ 0, \sqrt{0.99}\gamma \right]; \\ -\frac{100}{\gamma^2}\tau^2 + 100, & \text{if } |\tau| \in \left[ \sqrt{0.99}\gamma, \gamma \right]; \\ 0, & \text{else.} \end{cases} \tag{87}
$$

Since $\boldsymbol{h}$ and $\boldsymbol{z}$ are assumed to be unit vectors, these two functions obey

$$
0 \leq \chi_z\left( \boldsymbol{a}_i^\top \boldsymbol{z} \right) \leq \mathbf{1}_{\mathcal{E}_1^i}, \quad \text{and} \quad 0 \leq \chi_h\left( \boldsymbol{a}_i^\top \boldsymbol{h} \right) \leq \mathbf{1}_{\mathcal{D}_\gamma^{i,1}} \tag{88}
$$

and thus,

$$\frac{1}{m}\sum_{i=1}^{m}\left(\boldsymbol{a}_i^\top\boldsymbol{h}\right)^2\mathbf{1}_{\mathcal{E}_1^i\cap\mathcal{D}_\gamma^{i,1}} \geq \frac{1}{m}\sum_{i=1}^{m}(\boldsymbol{a}_i^\top\boldsymbol{h})^2\chi_z(\boldsymbol{a}_i^\top\boldsymbol{z})\chi_h(\boldsymbol{a}_i^\top\boldsymbol{h}). \tag{89}$$

We proceed to lower bound $\frac{1}{m}\sum_{i=1}^{m}\left(\boldsymbol{a}_i^\top\boldsymbol{h}\right)^2\chi_z\left(\boldsymbol{a}_i^\top\boldsymbol{z}\right)\chi_h\left(\boldsymbol{a}_i^\top\boldsymbol{h}\right)$.

Firstly, to compute the mean of $(\boldsymbol{a}_i^\top\boldsymbol{h})^2\chi_z(\boldsymbol{a}_i^\top\boldsymbol{z})\chi_h(\boldsymbol{a}_i^\top\boldsymbol{h})$, we introduce an auxiliary orthonormal matrix

$$\boldsymbol{U}_{\boldsymbol{z}} = \begin{bmatrix} \boldsymbol{z}^\top/\|\boldsymbol{z}\| \\ \vdots \end{bmatrix} \tag{90}$$

whose first row is along the direction of $\boldsymbol{z}$, and set

$$\tilde{\boldsymbol{h}} := \boldsymbol{U}_{\boldsymbol{z}}\boldsymbol{h}, \quad \text{and} \quad \tilde{\boldsymbol{a}}_i := \boldsymbol{U}_{\boldsymbol{z}}\boldsymbol{a}_i. \tag{91}$$

Also, denote by $\tilde{a}_{i,1}$ (resp. $\tilde{h}_1$) the first entry of $\tilde{\boldsymbol{a}}_i$ (resp. $\tilde{\boldsymbol{h}}$), and $\tilde{\boldsymbol{a}}_{i,\backslash 1}$ (resp. $\tilde{\boldsymbol{h}}_{\backslash 1}$) the remaining entries of $\tilde{\boldsymbol{a}}_i$ (resp. $\tilde{\boldsymbol{h}}$), and let $\xi\sim\mathcal{N}(0,1)$. We have

$$\begin{aligned}
\mathbb{E}\left[\left(\boldsymbol{a}_i^\top\boldsymbol{h}\right)^2\chi_z\left(\boldsymbol{a}_i^\top\boldsymbol{z}\right)\chi_h\left(\boldsymbol{a}_i^\top\boldsymbol{h}\right)\right] &\geq \mathbb{E}\left[(\boldsymbol{a}_i^\top\boldsymbol{h})^2\chi_z\left(\boldsymbol{a}_i^\top\boldsymbol{z}\right)\right] - \mathbb{E}\left[\left(\boldsymbol{a}_i^\top\boldsymbol{h}\right)^2\left(1-\chi_h\left(\boldsymbol{a}_i^\top\boldsymbol{h}\right)\right)\right] \\
&\geq \mathbb{E}\left[\left(\tilde{a}_{i,1}\tilde{h}_1\right)^2\chi_z\left(\boldsymbol{a}_i^\top\boldsymbol{z}\right)\right] + \mathbb{E}\left[\left(\tilde{\boldsymbol{a}}_{i,\backslash 1}^\top\tilde{\boldsymbol{h}}_{\backslash 1}\right)^2\right]\mathbb{E}\left[\chi_z\left(\boldsymbol{a}_i^\top\boldsymbol{z}\right)\right] - \|\boldsymbol{h}\|^2\mathbb{E}\left[\xi^2\mathbf{1}_{\{|\xi|>\sqrt{0.99}\gamma\}}\right] \\
&\geq |\tilde{h}_1|^2(1-\zeta_1) + \|\tilde{\boldsymbol{h}}_{\backslash 1}\|^2(1-\zeta_1) - \zeta_2\|\boldsymbol{h}\|^2 \\
&\geq (1-\zeta_1-\zeta_2)\|\boldsymbol{h}\|^2,
\end{aligned} \tag{92}$$

where the identity (92) arises from (39) and (40). Since $\left(\boldsymbol{a}_i^\top\boldsymbol{h}\right)^2\chi_z\left(\boldsymbol{a}_i^\top\boldsymbol{z}\right)\chi_h\left(\boldsymbol{a}_i^\top\boldsymbol{h}\right)$ is bounded in magnitude by $\gamma^2\|\boldsymbol{h}\|^2$, it is a sub-Gaussian random variable with sub-Gaussian norm $\mathcal{O}(\gamma^2\|\boldsymbol{h}\|^2)$. Apply the Hoeffding-type inequality [3, Proposition 5.10] to deduce that for any $\epsilon>0$,

$$\begin{aligned}
\frac{1}{m}\sum_{i=1}^{m}\left(\boldsymbol{a}_i^\top\boldsymbol{h}\right)^2\chi_z\left(\boldsymbol{a}_i^\top\boldsymbol{z}\right)\chi_h\left(\boldsymbol{a}_i^\top\boldsymbol{h}\right) &\geq \mathbb{E}\left[\left(\boldsymbol{a}_i^\top\boldsymbol{h}\right)^2\chi_z\left(\boldsymbol{a}_i^\top\boldsymbol{z}\right)\chi_h\left(\boldsymbol{a}_i^\top\boldsymbol{h}\right)\right] - \epsilon\|\boldsymbol{h}\|^2 \tag{93} \\
&\geq (1-\zeta_1-\zeta_2-\epsilon)\|\boldsymbol{h}\|^2 \tag{94}
\end{aligned}$$

with probability at least $1-\exp(-\Omega(\epsilon^2 m))$.

The next step is to obtain uniform control over all *unit vectors*, for which we adopt a basic version of an $\epsilon$-net argument. Specifically, we construct an $\epsilon$-net $\mathcal{N}_\epsilon$ with cardinality $|\mathcal{N}_\epsilon|\leq(1+2/\epsilon)^{2n}$ (cf. [3]) such that for any $(\boldsymbol{h},\boldsymbol{z})$ with $\|\boldsymbol{h}\|=\|\boldsymbol{z}\|=1$, there exists a pair $\boldsymbol{h}_0,\boldsymbol{z}_0\in\mathcal{N}_\epsilon$ satisfying $\|\boldsymbol{h}-\boldsymbol{h}_0\|\leq\epsilon$ and $\|\boldsymbol{z}-\boldsymbol{z}_0\|\leq\epsilon$. Now that we have discretized the unit spheres using a finite set, taking the union bound gives

$$\frac{1}{m}\sum_{i=1}^{m}\left(\boldsymbol{a}_i^\top\boldsymbol{h}_0\right)^2\chi_z\left(\boldsymbol{a}_i^\top\boldsymbol{z}_0\right)\chi_h\left(\boldsymbol{a}_i^\top\boldsymbol{h}_0\right)\geq(1-\zeta_1-\zeta_2-\epsilon)\|\boldsymbol{h}_0\|^2, \quad \forall\boldsymbol{h}_0,\boldsymbol{z}_0\in\mathcal{N}_\epsilon \tag{95}$$

with probability at least $1-(1+2/\epsilon)^{2n}\exp(-\Omega(\epsilon^2 m))$.

Define $f_1(\cdot)$ and $f_2(\cdot)$ such that $f_1(\tau):=\tau\chi_h(\sqrt{\tau})$ and $f_2(\tau):=\chi_z(\sqrt{\tau})$, which are both bounded functions with Lipschitz constant $\mathcal{O}(1)$. This guarantees that for each *unit* vector pair $\boldsymbol{h}$ and $\boldsymbol{z}$,

$$\begin{aligned}
&\left|\left(\boldsymbol{a}_i^\top\boldsymbol{h}\right)^2\chi_z\left(\boldsymbol{a}_i^\top\boldsymbol{z}\right)\chi_h\left(\boldsymbol{a}_i^\top\boldsymbol{h}\right) - \left(\boldsymbol{a}_i^\top\boldsymbol{h}_0\right)^2\chi_z\left(\boldsymbol{a}_i^\top\boldsymbol{z}_0\right)\chi_h\left(\boldsymbol{a}_i^\top\boldsymbol{h}_0\right)\right| \\
&\quad\leq \left|\chi_h\left(\boldsymbol{a}_i^\top\boldsymbol{z}\right)\right|\cdot\left|\left(\boldsymbol{a}_i^\top\boldsymbol{h}\right)^2\chi_h\left(\boldsymbol{a}_i^\top\boldsymbol{h}\right) - \left(\boldsymbol{a}_i^\top\boldsymbol{h}_0\right)^2\chi_h\left(\boldsymbol{a}_i^\top\boldsymbol{h}_0\right)\right| + \left|(\boldsymbol{a}_i^\top\boldsymbol{h}_0)^2\chi_h\left(\boldsymbol{a}_i^\top\boldsymbol{h}_0\right)\right|\cdot\left|\chi_z\left(\boldsymbol{a}_i^\top\boldsymbol{z}\right) - \chi_z\left(\boldsymbol{a}_i^\top\boldsymbol{z}_0\right)\right| \\
&\quad\leq \left|\chi_h\left(\boldsymbol{a}_i^\top\boldsymbol{z}\right)\right|\cdot\left|f_1\left(|\boldsymbol{a}_i^\top\boldsymbol{h}|^2\right) - f_1\left(|\boldsymbol{a}_i^\top\boldsymbol{h}_0|^2\right)\right| + \left|(\boldsymbol{a}_i^\top\boldsymbol{h}_0)^2\chi_h\left(\boldsymbol{a}_i^\top\boldsymbol{h}_0\right)\right|\cdot\left|f_2\left(|\boldsymbol{a}_i^\top\boldsymbol{z}|^2\right) - f_2\left(|\boldsymbol{a}_i^\top\boldsymbol{z}_0|^2\right)\right| \\
&\quad\lesssim \left|(\boldsymbol{a}_i^\top\boldsymbol{h})^2 - (\boldsymbol{a}_i^\top\boldsymbol{h}_0)^2\right| + (\boldsymbol{a}_i^\top\boldsymbol{z})^2 - (\boldsymbol{a}_i^\top\boldsymbol{z}_0)^2|.
\end{aligned}$$

Consequently, there exists some universal constant $c_3 > 0$ such that

$$\left| \frac{1}{m} \sum_{i=1}^m \left( \boldsymbol{a}_i^\top \boldsymbol{h} \right)^2 \chi_z \left( \boldsymbol{a}_i^\top \boldsymbol{z} \right) \chi_h \left( \boldsymbol{a}_i^\top \boldsymbol{h} \right) - \frac{1}{m} \sum_{i=1}^m \left( \boldsymbol{a}_i^\top \boldsymbol{h}_0 \right)^2 \chi_z \left( \boldsymbol{a}_i^\top \boldsymbol{z}_0 \right) \chi_h \left( \boldsymbol{a}_i^\top \boldsymbol{h}_0 \right) \right|$$

$$\lesssim \frac{1}{m} \left\| \mathcal{A}(\boldsymbol{h}\boldsymbol{h}^\top - \boldsymbol{h}_0\boldsymbol{h}_0^\top) \right\|_1 + \frac{1}{m} \left\| \mathcal{A}(\boldsymbol{z}\boldsymbol{z}^\top - \boldsymbol{z}_0\boldsymbol{z}_0^\top) \right\|_1$$

$$\overset{(i)}{\leq} c_3 \left\{ \left\| \boldsymbol{h}\boldsymbol{h}^\top - \boldsymbol{h}_0\boldsymbol{h}_0^\top \right\|_{\mathrm{F}} + \left\| \boldsymbol{z}\boldsymbol{z}^\top - \boldsymbol{z}_0\boldsymbol{z}_0^\top \right\|_{\mathrm{F}} \right\}$$

$$\overset{(ii)}{\leq} 2.5c_3 \left\{ \| \boldsymbol{h} - \boldsymbol{h}_0 \| \cdot \| \boldsymbol{h} \| + \| \boldsymbol{z} - \boldsymbol{z}_0 \| \cdot \| \boldsymbol{z} \| \right\} \leq 5c_3 \epsilon,$$

where (i) results from Lemma 1, and (ii) arises from Lemma 2 whenever $\epsilon < 1/2$.

With the assertion (95) in place, we see that with high probability,

$$\frac{1}{m} \sum_{i=1}^m \left( \boldsymbol{a}_i^\top \boldsymbol{h} \right)^2 \chi_z \left( \boldsymbol{a}_i^\top \boldsymbol{z} \right) \chi_h \left( \boldsymbol{a}_i^\top \boldsymbol{h} \right) \geq (1 - \zeta_1 - \zeta_2 - (5c_3 + 1)\,\epsilon) \| \boldsymbol{h} \|^2$$

for all unit vectors $\boldsymbol{h}$ and $\boldsymbol{z}$. Since $\epsilon$ can be arbitrary, putting this and (89) together completes the proof.

## A.3   Proof of Lemma 5

The proof makes use of standard concentration of measure and covering arguments, and it suffices to restrict our attention to *unit vectors* $\boldsymbol{h}$. We find it convenient to work with an auxiliary function

$$\chi_2(\tau) = \begin{cases} |\tau|^{\frac{3}{2}}, & \text{if } |\tau| \leq \gamma^2, \\ -\gamma \left( |\tau| - \gamma^2 \right) + \gamma^3, & \text{if } \gamma^2 < |\tau| \leq 2\gamma^2, \\ 0, & \text{else.} \end{cases}$$

Apparently, $\chi_2(\tau)$ is a Lipschitz function of $\tau$ with Lipschitz norm $\mathcal{O}(\gamma)$. Recalling the definition of $\mathcal{D}_\gamma^{i,1}$, we see that each summand is bounded above by

$$|\boldsymbol{a}_i^\top \boldsymbol{h}|^3 \, \mathbf{1}_{\mathcal{D}_\gamma^{i,1}} \leq \chi_2\big(|\boldsymbol{a}_i^\top \boldsymbol{h}|^2\big).$$

For each fixed $\boldsymbol{h}$ and $\epsilon > 0$, applying the Bernstein inequality [3, Proposition 5.16] gives

$$\frac{1}{m} \sum_{i=1}^m |\boldsymbol{a}_i^\top \boldsymbol{h}|^3 \mathbf{1}_{\mathcal{D}_\gamma^{i,1}} \leq \frac{1}{m} \sum_{i=1}^m \chi_2 \left( |\boldsymbol{a}_i^\top \boldsymbol{h}|^2 \right) \leq \mathbb{E}\left[ \chi_2 \left( |\boldsymbol{a}_i^\top \boldsymbol{h}|^2 \right) \right] + \epsilon$$

$$\leq \mathbb{E}\big[ |\boldsymbol{a}_i^\top \boldsymbol{h}|^3 \big] + \epsilon = \sqrt{8/\pi} + \epsilon$$

with probability exceeding $1 - \exp\left( -\Omega\left( \epsilon^2 m \right) \right)$.

From [3, Lemma 5.2], there exists an $\epsilon$-net $\mathcal{N}_\epsilon$ of the unit sphere with cardinality $|\mathcal{N}_\epsilon| \leq \left( 1 + \frac{2}{\epsilon} \right)^n$. For each $\boldsymbol{h}$, suppose that $\| \boldsymbol{h}_0 - \boldsymbol{h} \| \leq \epsilon$ for some $\boldsymbol{h}_0 \in \mathcal{N}_\epsilon$. The Lipschitz property of $\chi_2$ implies

$$\frac{1}{m} \sum_{i=1}^m \left\{ \chi_2 \left( |\boldsymbol{a}_i^\top \boldsymbol{h}|^2 \right) - \chi_2 \left( |\boldsymbol{a}_i^\top \boldsymbol{h}_0|^2 \right) \right\} \lesssim \frac{1}{m} \sum_{i=1}^m \left| |\boldsymbol{a}_i^\top \boldsymbol{h}|^2 - |\boldsymbol{a}_i^\top \boldsymbol{h}_0|^2 \right| \overset{(i)}{\asymp} \| \boldsymbol{h} - \boldsymbol{h}_0 \| \| \boldsymbol{h} \| \asymp \epsilon,$$

where (i) arises by combining Lemmas 1 and 2. This demonstrates that with high probability,

$$\frac{1}{m} \sum_{i=1}^m |\boldsymbol{a}_i^\top \boldsymbol{h}|^3 \mathbf{1}_{\mathcal{D}_\gamma^{i,1}} \leq \frac{1}{m} \sum_{i=1}^m \chi_2 \left( |\boldsymbol{a}_i^\top \boldsymbol{h}|^2 \right) \leq \sqrt{8/\pi} + \mathcal{O}(\epsilon)$$

for all unit vectors $\boldsymbol{h}$, as claimed.

## A.4 Proof of Lemma 6

Without loss of generality, the proof focuses on the case where $\|\boldsymbol{h}\| = 1$. Fix an arbitrary small constant $\delta > 0$. One can eliminate the difficulty of handling the discontinuous indicator functions by working with the following auxiliary function

$$\chi_3(\tau, \gamma) := \begin{cases} 1, & \text{if } \sqrt{\tau} \geq \psi_{\text{lb}}(\gamma); \\ \frac{100\tau}{\psi_{\text{lb}}^2(\gamma)} - 99, & \text{if } \sqrt{\tau} \in \left[\sqrt{0.99}\psi_{\text{lb}}(\gamma), \psi_{\text{lb}}(\gamma)\right]; \\ 0, & \text{else.} \end{cases} \tag{96}$$

Here, $\psi_{\text{lb}}(\cdot)$ is a piecewise constant function defined as

$$\psi_{\text{lb}}(\gamma) := (1+\delta)^{\left\lfloor \frac{\log \gamma}{\log(1+\delta)} \right\rfloor},$$

which clearly satisfy $\frac{\gamma}{1+\delta} \leq \psi_{\text{lb}}(\gamma) \leq \gamma$. Such a function is useful for our purpose since for any $0 < \delta \leq 0.005$,

$$\mathbf{1}_{\{|\boldsymbol{a}_i^\top \boldsymbol{h}| \geq \gamma\}} \leq \chi_3\left(\left|\boldsymbol{a}_i^\top \boldsymbol{h}\right|^2, \gamma\right) \leq \mathbf{1}_{\{|\boldsymbol{a}_i^\top \boldsymbol{h}| \geq \sqrt{0.99}\psi_{\text{lb}}(\gamma)\}} \leq \mathbf{1}_{\{|\boldsymbol{a}_i^\top \boldsymbol{h}| \geq 0.99\gamma\}}. \tag{97}$$

For any fixed unit vector $\boldsymbol{h}$, the above argument leads to an upper tail estimate: for any $0 < t \leq 1$,

$$\begin{aligned} \mathbb{P}\left\{\chi_3\left(\left|\boldsymbol{a}_i^\top \boldsymbol{h}\right|^2, \gamma\right) \geq t\right\} &\leq \mathbb{P}\left\{\mathbf{1}_{\{|\boldsymbol{a}_i^\top \boldsymbol{h}| \geq 0.99\gamma\}} \geq t\right\} = \mathbb{P}\left\{\mathbf{1}_{\{|\boldsymbol{a}_i^\top \boldsymbol{h}| \geq 0.99\gamma\}} = 1\right\} \\ &= 2\int_{0.99\gamma}^\infty \phi(x)\,\mathrm{d}x \leq \frac{2}{0.99\gamma}\phi(0.99\gamma), \end{aligned} \tag{98}$$

where $\phi(x)$ is the density of a standard normal, and (98) follows from the tail bound $\int_x^\infty \phi(x)\mathrm{d}x \leq \frac{1}{x}\phi(x)$ for all $x > 0$. This implies that when $\gamma \geq 2$, both $\chi_3(|\boldsymbol{a}_i^\top \boldsymbol{h}|^2, \gamma)$ and $\mathbf{1}_{\{|\boldsymbol{a}_i^\top \boldsymbol{h}| \geq 0.99\gamma\}}$ are sub-exponential with sub-exponential norm $\mathcal{O}(\gamma^{-2})$ (cf. [3, Definition 5.13]). We apply the Bernstein-type inequality for the sum of sub-exponential random variables [3, Corollary 5.17], which indicates that for any fixed $\boldsymbol{h}$ and $\gamma$ as well as any sufficiently small $\epsilon \in (0, 1)$,

$$\begin{aligned} \frac{1}{m}\sum_{i=1}^m \chi_3\left(\left|\boldsymbol{a}_i^\top \boldsymbol{h}\right|^2, \gamma\right) &\leq \frac{1}{m}\sum_{i=1}^m \mathbf{1}_{\{|\boldsymbol{a}_i^\top \boldsymbol{h}| \geq 0.99\gamma\}} \leq \mathbb{E}\left[\mathbf{1}_{\{|\boldsymbol{a}_i^\top \boldsymbol{h}| \geq 0.99\gamma\}}\right] + \epsilon\frac{1}{\gamma^2} \\ &\leq \frac{2}{0.99\gamma}\exp\left(-0.49\gamma^2\right) + \epsilon\frac{1}{\gamma^2} \end{aligned}$$

holds with probability exceeding $1 - \exp\left(-\Omega(\epsilon^2 m)\right)$.

We now proceed to obtain uniform control over all $\boldsymbol{h}$ and $2 \leq \gamma \leq 2^n$. To begin with, we consider all $2 \leq \gamma \leq m$ and construct an $\epsilon$-net $\mathcal{N}_\epsilon$ over the unit sphere such that: (i) $|\mathcal{N}_\epsilon| \leq \left(1 + \frac{2}{\epsilon}\right)^n$; (ii) for any $\boldsymbol{h}$ with $\|\boldsymbol{h}\| = 1$, there exists a unit vector $\boldsymbol{h}_0 \in \mathcal{N}_\epsilon$ obeying $\|\boldsymbol{h} - \boldsymbol{h}_0\| \leq \epsilon$. Taking the union bound gives the following: with probability at least $1 - \frac{\log m}{\log(1+\delta)}\left(1 + \frac{2}{\epsilon}\right)^n \exp(-\Omega(\epsilon^2 m))$,

$$\frac{1}{m}\sum_{i=1}^m \chi_3\left(\left|\boldsymbol{a}_i^\top \boldsymbol{h}_0\right|^2, \gamma_0\right) \leq (0.495\gamma_0)^{-1}\exp\left(-0.49\gamma_0^2\right) + \epsilon\gamma_0^{-2}$$

holds simultaneously for all $\boldsymbol{h}_0 \in \mathcal{N}_\epsilon$ and $\gamma_0 \in \left\{(1+\delta)^k \mid 1 \leq k \leq \frac{\log m}{\log(1+\delta)}\right\}$.

Note that $\chi_3(\tau, \gamma_0)$ is a Lipschitz function in $\tau$ with the Lipschitz constant bounded above by $\frac{100}{\psi_{\text{lb}}^2(\gamma_0)}$. With this in mind, for any $(\boldsymbol{h}, \gamma)$ with $\|\boldsymbol{h}\| = 1$ and $\gamma_0 := (1+\delta)^k \leq \gamma < (1+\delta)^{k+1}$, one has

$$\begin{aligned} \left|\chi_3\left(\left|\boldsymbol{a}_i^\top \boldsymbol{h}_0\right|^2, \gamma_0\right) - \chi_3\left(\left|\boldsymbol{a}_i^\top \boldsymbol{h}\right|^2, \gamma\right)\right| &= \left|\chi_3\left(\left|\boldsymbol{a}_i^\top \boldsymbol{h}_0\right|^2, \gamma_0\right) - \chi_3\left(\left|\boldsymbol{a}_i^\top \boldsymbol{h}\right|^2, \gamma_0\right)\right| \\ &\leq \frac{100}{\psi_{\text{lb}}^2(\gamma_0)}\left|\left|\boldsymbol{a}_i^\top \boldsymbol{h}\right|^2 - \left|\boldsymbol{a}_i^\top \boldsymbol{h}_0\right|^2\right|. \end{aligned}$$

It then follows from Lemmas 1-2 that

$$\frac{1}{m}\left|\sum_{i=1}^{m}\chi_3\left(\left|\boldsymbol{a}_i^\top \boldsymbol{h}_0\right|^2,\ \gamma_0\right)-\sum_{i=1}^{m}\chi_3\left(\left|\boldsymbol{a}_i^\top \boldsymbol{h}\right|^2,\ \gamma\right)\right| \leq \frac{100}{\psi_{\mathrm{lb}}^2\left(\gamma_0\right)}\frac{1}{m}\left\|\mathcal{A}\left(\boldsymbol{h}\boldsymbol{h}^\top-\boldsymbol{h}_0\boldsymbol{h}_0^\top\right)\right\|_1$$

$$\leq \frac{250\left(1+\delta\right)^2}{\gamma^2}\|\boldsymbol{h}-\boldsymbol{h}_0\|\|\boldsymbol{h}\| \leq \frac{250(1+\delta)^2\epsilon}{\gamma^2}.$$

Putting the above results together gives that for all $2 \leq \gamma \leq (1+\delta)^{\frac{\log m}{\log(1+\delta)}} = m$,

$$\frac{1}{m}\sum_{i=1}^{m}\chi_3\left(\left|\boldsymbol{a}_i^\top\boldsymbol{h}\right|^2,\ \gamma\right) \leq \frac{1}{m}\sum_{i=1}^{m}\chi_3\left(\left|\boldsymbol{a}_i^\top\boldsymbol{h}_0\right|^2,\ \gamma_0\right)+\frac{250\left(1+\delta\right)^2}{\gamma^2}\epsilon$$

$$\leq \frac{1}{0.495\gamma_0}\exp\left(-0.49\gamma_0^2\right)+251\left(1+\delta\right)^2\frac{\epsilon}{\gamma^2}$$

$$\leq \frac{1}{0.49\gamma}\exp\left(-0.485\gamma^2\right)+251\left(1+\delta\right)^2\frac{\epsilon}{\gamma^2}$$

with probability exceeding $1-\frac{\log m}{\log(1+\delta)}\left(1+\frac{2}{\epsilon}\right)^n\exp\left(-c\epsilon^2 m\right)$. This establishes (44) for all $2\leq\gamma\leq m$.

It remains to deal with the case where $\gamma > m$. To this end, we rely on the following observation:

$$\frac{1}{m}\sum_{i=1}^{m}\mathbf{1}_{\left\{|\boldsymbol{a}_i^\top\boldsymbol{h}|\geq m\right\}} \leq \frac{1}{m}\sum_{i=1}^{m}\frac{\left|\boldsymbol{a}_i^\top\boldsymbol{h}\right|^2}{m^2} \overset{\text{(i)}}{\leq} \frac{1+\delta}{m^2}\|\boldsymbol{h}\|^2 \ll \frac{1}{m}, \quad \forall\boldsymbol{h}\ \text{with}\ \|\boldsymbol{h}\|=1,$$

where (i) comes from [2, Lemmas 3.1]. This basically tells us that with high probability, none of the indicator variables can be equal to 1. Consequently, $\frac{1}{m}\sum_{i=1}^{m}\mathbf{1}_{\left\{|\boldsymbol{a}_i^\top\boldsymbol{h}|\geq m\right\}} = 0$, which proves the claim.

## A.5  Proof of Lemma 7

Fix $\delta > 0$. Recalling the notation $v_i := 2\left\{2\boldsymbol{a}_i^\top\boldsymbol{h}-\frac{|\boldsymbol{a}_i^\top\boldsymbol{h}|^2}{\boldsymbol{a}_i^\top\boldsymbol{z}}\right\}\mathbf{1}_{\mathcal{E}_1^i\cap\mathcal{E}_2^i}$, we see from the expansion (35) that

$$\left\|\frac{1}{m}\nabla_{\mathrm{tr}}\ell(\boldsymbol{z})\right\|=\left\|\frac{1}{m}\boldsymbol{A}^\top\boldsymbol{v}\right\| \leq \frac{1}{m}\|\boldsymbol{A}\|\cdot\|\boldsymbol{v}\| \leq (1+\delta)\frac{\|\boldsymbol{v}\|}{\sqrt{m}} \tag{99}$$

as soon as $m \geq c_1 n$ for some sufficiently large $c_1 > 0$. Here, the norm estimate $\|\boldsymbol{A}\| \leq \sqrt{m}\left(1+\delta\right)$ arises from standard random matrix results [3, Corollary 5.35].

Everything then comes down to controlling $\|\boldsymbol{v}\|$. To this end, making use of the inclusion (36) yields

$$\frac{1}{4m}\|\boldsymbol{v}\|^2 = \frac{1}{m}\sum_{i=1}^{m}\left(2\boldsymbol{a}_i^\top\boldsymbol{h}-\frac{|\boldsymbol{a}_i^\top\boldsymbol{h}|^2}{\boldsymbol{a}_i^\top\boldsymbol{z}}\right)^2\mathbf{1}_{\mathcal{E}_1^i\cap\mathcal{E}_2^i} \leq \frac{1}{m}\sum_{i=1}^{m}\left(2\left|\boldsymbol{a}_i^\top\boldsymbol{h}\right|+\frac{|\boldsymbol{a}_i^\top\boldsymbol{h}|^2}{|\boldsymbol{a}_i^\top\boldsymbol{z}|}\right)^2\mathbf{1}_{\mathcal{E}_1^i\cap\left(\mathcal{D}_{\gamma_4}^{i,1}\cup\mathcal{D}_{\gamma_4}^{i,2}\right)}$$

$$\leq \frac{1}{m}\sum_{i=1}^{m}\left\{4(\boldsymbol{a}_i^\top\boldsymbol{h})^2+\left(\frac{4|\boldsymbol{a}_i^\top\boldsymbol{h}|^3}{|\boldsymbol{a}_i^\top\boldsymbol{z}|}+\frac{|\boldsymbol{a}_i^\top\boldsymbol{h}|^4}{|\boldsymbol{a}_i^\top\boldsymbol{z}|^2}\right)\mathbf{1}_{\mathcal{E}_1^i\cap\left(\mathcal{D}_{\gamma_4}^{i,1}\cup\mathcal{D}_{\gamma_4}^{i,2}\right)}\right\}$$

$$= \frac{1}{m}\sum_{i=1}^{m}\left\{4\left(\boldsymbol{a}_i^\top\boldsymbol{h}\right)^2+\left(4+\frac{|\boldsymbol{a}_i^\top\boldsymbol{h}|}{|\boldsymbol{a}_i^\top\boldsymbol{z}|}\right)\frac{|\boldsymbol{a}_i^\top\boldsymbol{h}|^3}{|\boldsymbol{a}_i^\top\boldsymbol{z}|}\left(\mathbf{1}_{\mathcal{E}_1^i\cap\mathcal{D}_{\gamma_4}^{i,1}}+\mathbf{1}_{\mathcal{E}_1^i\cap\mathcal{D}_{\gamma_4}^{i,2}}\right)\right\}.$$

The first term is controlled by [2, Lemma 3.1] in such a way that with probability $1-\exp(-\Omega(m))$,

$$\frac{1}{m}\sum_{i=1}^{m}4\left(\boldsymbol{a}_i^\top\boldsymbol{h}\right)^2 \leq 4\left(1+\delta\right)\|\boldsymbol{h}\|^2.$$

Turning to the remaining terms, we see from the definition of $\mathcal{D}_\gamma^{i,1}$ and $\mathcal{D}_\gamma^{i,2}$ that

$$\frac{|\boldsymbol{a}_i^\top\boldsymbol{h}|}{|\boldsymbol{a}_i^\top\boldsymbol{z}|} \leq \begin{cases}\frac{\gamma\|\boldsymbol{h}\|}{\alpha_z^{\mathrm{lb}}\|\boldsymbol{z}\|}, & \text{on }\mathcal{E}_1^i\cap\mathcal{D}_\gamma^{i,1} \\ 2+\frac{\gamma\|\boldsymbol{h}\|}{\alpha_z^{\mathrm{lb}}\|\boldsymbol{z}\|}, & \text{on }\mathcal{E}_1^i\cap\mathcal{D}_\gamma^{i,2}\end{cases} \leq \begin{cases}1, & \text{on }\mathcal{E}_1^i\cap\mathcal{D}_\gamma^{i,1} \\ 3, & \text{on }\mathcal{E}_1^i\cap\mathcal{D}_\gamma^{i,2}\end{cases}$$

as long as $\gamma \leq \frac{\alpha_z^{\mathrm{lb}}\|z\|}{\|h\|}$. Consequently, one can bound

$$\frac{1}{m}\sum_{i=1}^{m}\left(4+\frac{|\boldsymbol{a}_i^\top \boldsymbol{h}|}{|\boldsymbol{a}_i^\top \boldsymbol{z}|}\right)\frac{|\boldsymbol{a}_i^\top \boldsymbol{h}|^3}{|\boldsymbol{a}_i^\top \boldsymbol{z}|}\left(\mathbf{1}_{\mathcal{E}_1^i\cap\mathcal{D}_\gamma^{i,1}}+\mathbf{1}_{\mathcal{E}_1^i\cap\mathcal{D}_\gamma^{i,2}}\right)$$

$$\leq \frac{5}{m}\sum_{i=1}^{m}\frac{|\boldsymbol{a}_i^\top \boldsymbol{h}|^3}{|\boldsymbol{a}_i^\top \boldsymbol{z}|}\mathbf{1}_{\mathcal{E}_1^i\cap\mathcal{D}_\gamma^{i,1}}+\frac{7}{m}\sum_{i=1}^{m}\frac{|\boldsymbol{a}_i^\top \boldsymbol{h}|^3}{|\boldsymbol{a}_i^\top \boldsymbol{z}|}\mathbf{1}_{\mathcal{E}_1^i\cap\mathcal{D}_\gamma^{i,2}}$$

$$\leq \frac{5\left(1+\delta\right)\sqrt{8/\pi}\|\boldsymbol{h}\|^3}{\alpha_z^{\mathrm{lb}}\|\boldsymbol{z}\|}+\frac{7}{100}\left(1+\delta\right)\|\boldsymbol{h}\|^2,$$

where the last inequality follows from (42) and (51).

Recall that $\gamma_4 = 3\alpha_h$. Taken together all these bounds lead to the upper bound

$$\frac{1}{4m}\|\boldsymbol{v}\|^2 \;\leq\; (1+\delta)\left\{4+\frac{5\sqrt{8/\pi}\,\|\boldsymbol{h}\|}{\alpha_z^{\mathrm{lb}}\,\|\boldsymbol{z}\|}+\frac{7}{100}\right\}\|\boldsymbol{h}\|^2 \;\leq\; (1+\delta)\left\{4+\frac{5\sqrt{8/\pi}}{3\alpha_h}+\frac{7}{100}\right\}\|\boldsymbol{h}\|^2$$

whenever $\frac{\|\boldsymbol{h}\|}{\|\boldsymbol{z}\|}\leq \min\left\{\frac{\alpha_z^{\mathrm{lb}}}{3\alpha_h},\frac{\alpha_z^{\mathrm{lb}}}{6},\frac{\sqrt{98/3}(\alpha_z^{\mathrm{lb}})^2}{2\alpha_z^{\mathrm{ub}}+\alpha_z^{\mathrm{lb}}},\frac{1}{11}\right\}$. Substituting this into (99) completes the proof.

# B  Proofs for Section 4

## B.1  Proof of Lemma 8

Firstly, we collect a few results on the magnitudes of $\boldsymbol{a}_i^\top \boldsymbol{x}$ ($1\leq i\leq m$) that will be useful in constructing the hypotheses. Observe that for any given $\boldsymbol{x}$ and any sufficiently large $m$,

$$\mathbb{P}\left\{\min_{1\leq i\leq m}|\boldsymbol{a}_i^\top \boldsymbol{x}|\geq \frac{1}{m\log m}\|\boldsymbol{x}\|\right\}=\left(\mathbb{P}\left\{|\boldsymbol{a}_i^\top \boldsymbol{x}|\geq \frac{1}{m\log m}\|\boldsymbol{x}\|\right\}\right)^m\geq \left(1-\frac{2}{\sqrt{2\pi}}\frac{1}{m\log m}\right)^m\geq 1-o(1).$$

Besides, since $\mathbb{E}\left[\mathbf{1}_{\left\{|\boldsymbol{a}_i^\top \boldsymbol{x}|\leq \frac{\|\boldsymbol{x}\|}{5\log m}\right\}}\right]\leq \frac{1}{\sqrt{2\pi}}\frac{2}{5\log m}\leq \frac{1}{5\log m}$, applying Hoeffding's inequality yields

$$\mathbb{P}\left\{\sum_{i=1}^{m}\mathbf{1}_{\left\{|\boldsymbol{a}_i^\top \boldsymbol{x}|\leq \frac{\|\boldsymbol{x}\|}{5\log m}\right\}}>\frac{m}{4\log m}\right\}$$
$$=\mathbb{P}\left\{\frac{1}{m}\sum_{i=1}^{m}\left(\mathbf{1}_{\left\{|\boldsymbol{a}_i^\top \boldsymbol{x}|\leq \frac{\|\boldsymbol{x}\|}{5\log m}\right\}}-\mathbb{E}\left[\mathbf{1}_{\left\{|\boldsymbol{a}_i^\top \boldsymbol{x}|\leq \frac{\|\boldsymbol{x}\|}{5\log m}\right\}}\right]\right)>\frac{1}{20\log m}\right\}\leq \exp\left(-\Omega\left(\frac{m}{\log^2 m}\right)\right).$$

To summarize, with probability $1-o(1)$, one has

$$\min_{1\leq i\leq m}|\boldsymbol{a}_i^\top \boldsymbol{x}| \;\geq\; \frac{1}{m\log m}\|\boldsymbol{x}\|; \tag{100}$$

$$\sum_{i=1}^{m}\mathbf{1}_{\left\{|\boldsymbol{a}_i^\top \boldsymbol{x}|\leq \frac{\|\boldsymbol{x}\|}{\log m}\right\}} \;\leq\; \frac{m}{4\log m}:=k. \tag{101}$$

In the sequel, we will first produce a set $\mathcal{M}_1$ of exponentially many vectors surrounding $\boldsymbol{x}$ in such a way that every pair is separated by about the same distance, and then verify that a non-trivial fraction of $\mathcal{M}_1$ obeys (76). Without loss of generality, we assume that $\boldsymbol{x}$ takes the form $\boldsymbol{x}=[b,0,\cdots,0]^\top$ for some $b>0$.

The construction of $\mathcal{M}_1$ follows a standard random packing argument. Let $\boldsymbol{w}=[w_1,\cdots,w_n]^\top$ be a random vector with

$$w_i = x_i + \frac{1}{\sqrt{2n}}z_i,\quad 1\leq i\leq n,$$

where $z_i \overset{\mathrm{ind.}}{\sim}\mathcal{N}(0,1)$. The collection $\mathcal{M}_1$ is then obtained by generating $M_1=\exp\left(\frac{n}{20}\right)$ independent copies $\boldsymbol{w}^{(l)}$ ($1\leq l<M_1$) of $\boldsymbol{w}$. For any $\boldsymbol{w}^{(l)},\boldsymbol{w}^{(j)}\in\mathcal{M}_1$, the concentration inequality [3, Corollary 5.35] gives

$$\mathbb{P}\left\{0.5\sqrt{n}-1\leq \sqrt{n}\|\boldsymbol{w}^{(l)}-\boldsymbol{w}^{(j)}\|\leq 1.5\sqrt{n}+1\right\}\geq 1-2\exp\left(-n/8\right);$$
$$\mathbb{P}\left\{0.5\sqrt{n}-1\leq \sqrt{2n}\|\boldsymbol{w}^{(l)}-\boldsymbol{x}\|\leq 1.5\sqrt{n}+1\right\}\;\geq 1-2\exp\left(-n/8\right).$$

Taking the union bound over all $\binom{M_1}{2}$ pairs we obtain

$$
\begin{aligned}
0.5 - n^{-1/2} \leq \; & \left\| \boldsymbol{w}^{(l)} - \boldsymbol{w}^{(j)} \right\| && \leq 1.5 + n^{-1/2}, \quad \forall l \neq j \\
1/\sqrt{8} - (2n)^{-1/2} \leq \; & \left\| \boldsymbol{w}^{(l)} - \boldsymbol{x} \right\| && \leq \sqrt{9/8} + (2n)^{-1/2}, \quad 1 \leq l \leq M_1
\end{aligned}
\tag{102}
$$

with probability exceeding $1 - 2M_1^2 \exp\left(-\frac{n}{8}\right) \geq 1 - 2\exp\left(-\frac{n}{40}\right)$.

The next step is to show that many vectors in $\mathcal{M}_1$ satisfy (76). For any given $\boldsymbol{w}$ with $\boldsymbol{r} := \boldsymbol{w} - \boldsymbol{x}$, by letting $\boldsymbol{a}_{i,\perp} := [a_{i,2}, \cdots, a_{i,n}]^{\top}$, $r_{\parallel} := r_1$, and $\boldsymbol{r}_{\perp} := [r_2, \cdots, r_n]^{\top}$, we derive

$$
\frac{|\boldsymbol{a}_i^{\top} \boldsymbol{r}|^2}{|\boldsymbol{a}_i^{\top} \boldsymbol{x}|^2} \leq \frac{2|a_{i,1} r_{\parallel}|^2 + 2|\boldsymbol{a}_{i,\perp}^{\top} \boldsymbol{r}_{\perp}|^2}{|a_{i,1}|^2 \|\boldsymbol{x}\|^2} \leq \frac{2|r_{\parallel}|^2}{\|\boldsymbol{x}\|^2} + \frac{2|\boldsymbol{a}_{i,\perp}^{\top} \boldsymbol{r}_{\perp}|^2}{|a_{i,1}|^2 \|\boldsymbol{x}\|^2} \leq \frac{2\|\boldsymbol{r}\|^2}{\|\boldsymbol{x}\|^2} + \frac{2|\boldsymbol{a}_{i,\perp}^{\top} \boldsymbol{r}_{\perp}|^2}{|a_{i,1}|^2 \|\boldsymbol{x}\|^2}.
\tag{103}
$$

It then boils down to developing an upper bound on $\frac{|\boldsymbol{a}_{i,\perp}^{\top} \boldsymbol{r}_{\perp}|^2}{|a_{i,1}|^2}$. This ratio is convenient to work with since the numerator and denominator are stochastically independent. To simplify presentation, we reorder $\{\boldsymbol{a}_i\}$ in a way that

$$
(m \log m)^{-1} \|\boldsymbol{x}\| \leq \left| \boldsymbol{a}_1^{\top} \boldsymbol{x} \right| \leq \left| \boldsymbol{a}_2^{\top} \boldsymbol{x} \right| \leq \cdots \leq \left| \boldsymbol{a}_m^{\top} \boldsymbol{x} \right|;
$$

this will not affect our subsequent analysis concerning $\boldsymbol{a}_{i,\perp}^{\top} \boldsymbol{r}_{\perp}$ since it is independent of $\boldsymbol{a}_i^{\top} \boldsymbol{x}$.

To proceed, we let $\boldsymbol{r}_{\perp}^{(l)}$ consist of all but the first entry of $\boldsymbol{w}^{(l)} - \boldsymbol{x}$, and introduce the indicator variables

$$
\xi_i^l := \begin{cases} \mathbf{1}_{\left\{ \left| \boldsymbol{a}_{i,\perp}^{\top} \boldsymbol{r}_{\perp}^{(l)} \right| \leq \frac{1}{m}\sqrt{\frac{n-1}{2n}} \right\}}, & 1 \leq i \leq k, \\[2mm] \mathbf{1}_{\left\{ \left| \boldsymbol{a}_{i,\perp}^{\top} \boldsymbol{r}_{\perp}^{(l)} \right| \leq \sqrt{\frac{2(n-1)\log n}{n}} \right\}}, & i > k, \end{cases}
\tag{104}
$$

where $k = \frac{m}{4 \log m}$ as before. In words, we divide $\boldsymbol{a}_{i,\perp}^{\top} \boldsymbol{r}_{\perp}^{(l)}$, $1 \leq i \leq m$ into two groups, with the first group enforcing far more stringent control than the second group. These indicator variables are useful since any $\boldsymbol{w}^{(l)}$ obeying $\prod_{i=1}^{m} \xi_i^l = 1$ will satisfy (76) when $n$ is sufficiently large. To see this, note that for the first group of indices, $\xi_i^l = 1$ requires

$$
\left| \boldsymbol{a}_{i,\perp}^{\top} \boldsymbol{r}_{\perp}^{(l)} \right| \leq \frac{1}{m}\sqrt{\frac{n-1}{2n}} \leq \frac{2}{m}\frac{\sqrt{n-1}}{\sqrt{n}-2}\|\boldsymbol{r}^{(l)}\| \leq \frac{3}{m}\|\boldsymbol{r}^{(l)}\|, \quad 1 \leq i \leq k,
\tag{105}
$$

where the second inequality follows from (102). This taken collectively with (100) and (103) yields

$$
\frac{|\boldsymbol{a}_i^{\top} \boldsymbol{r}^{(l)}|^2}{|\boldsymbol{a}_i^{\top} \boldsymbol{x}|^2} \leq \frac{2\|\boldsymbol{r}^{(l)}\|^2}{\|\boldsymbol{x}\|^2} + \frac{\frac{9}{m^2}\|\boldsymbol{r}^{(l)}\|^2}{\frac{1}{m^2 \log^2 m}\|\boldsymbol{x}\|^2} \leq \frac{(2 + 9\log^2 m)\|\boldsymbol{r}^{(l)}\|^2}{\|\boldsymbol{x}\|^2}, \quad 1 \leq i \leq k.
$$

Regarding the second group of indices, $\xi_i^l = 1$ gives

$$
\left| \boldsymbol{a}_{i,\perp}^{\top} \boldsymbol{r}_{\perp}^{(l)} \right| \leq \sqrt{\frac{2(n-1)\log n}{n}} \leq \sqrt{17 \log n}\|\boldsymbol{r}^{(l)}\|, \quad i = k+1, \cdots, m,
\tag{106}
$$

where the last inequality again follows from (102). Plugging (106) and (101) into (103) gives

$$
\frac{|\boldsymbol{a}_i^{\top} \boldsymbol{r}^{(l)}|^2}{|\boldsymbol{a}_i^{\top} \boldsymbol{x}|^2} \leq \frac{2\|\boldsymbol{r}^{(l)}\|^2}{\|\boldsymbol{x}\|^2} + \frac{17\|\boldsymbol{r}^{(l)}\|^2 \log n}{\|\boldsymbol{x}\|^2 / \log^2 m} \leq \frac{(2 + 17\log^3 m)\|\boldsymbol{r}^{(l)}\|^2}{\|\boldsymbol{x}\|^2}, \quad i \geq k+1.
$$

Consequently, (76) is satified for all $1 \leq i \leq m$. It then suffices to guarantee the existence of exponentially many vectors obeying $\prod_{i=1}^{m} \xi_i^l = 1$.

Note that the first group of indicator variables are quite stringent, namely, for each $i$ only a fraction $\mathcal{O}(1/m)$ of the equations could satisfy $\xi_i^l = 1$. Fortunately, $M_1$ is exponentially large, and hence even $M_1/m^k$ is exponentially large. Put formally, we claim that the first group satisfies

$$
\sum_{l=1}^{M_1} \prod_{i=1}^{k} \xi_i^l \geq \frac{1}{2}\frac{M_1}{(2\pi)^{k/2}(1 + 4\sqrt{k/n})^{k/2}}\left(\frac{1}{\sqrt{2\pi}m}\right)^k := \widetilde{M}_1
\tag{107}
$$

with probability exceeding $1 - \exp\left(-\Omega\left(k\right)\right) - \exp(-\widetilde{M_1}/4)$. With this claim in place (which will be proved later), one has

$$\sum_{l=1}^{M_1} \prod_{i=1}^{k} \xi_i^l \geq \frac{1}{2} M_1 \frac{1}{\left(e^2 m\right)^k} = \frac{1}{2} \exp\left(\left(\frac{1}{20} - \frac{k\left(2 + \log m\right)}{n}\right) n\right) \geq \frac{1}{2} \exp\left(\frac{1}{25} n\right)$$

when $n$ and $m/n$ are sufficiently large. In light of this, we will let $\mathcal{M}_2$ be a collection comprising all $\boldsymbol{w}^{(l)}$ obeying $\prod_{i=1}^{k} \xi_i^l = 1$, which has size $M_2 \geq \frac{1}{2} \exp\left(\frac{1}{25} n\right)$ based on the preceding argument. For notational simplicity, it will be assumed that the vectors in $\mathcal{M}_2$ are exactly $\boldsymbol{w}^{(j)}$ $(1 \leq j \leq M_2)$.

We now move on to the second group by examining how many vectors $\boldsymbol{w}^{(j)}$ in $\mathcal{M}_2$ further satisfy $\prod_{i=k+1}^{m} \xi_i^j = 1$. Notably, the above construction of $\mathcal{M}_2$ relies only on $\{\boldsymbol{a}_i\}_{1 \leq i \leq k}$ and is independent of the remaining vectors $\{\boldsymbol{a}_i\}_{i>k}$. In what follows the argument proceeds conditional on $\mathcal{M}_2$ and $\{\boldsymbol{a}_i\}_{1 \leq i \leq k}$. Applying the union bound gives

$$\mathbb{E}\left[\sum_{j=1}^{M_2}\left(1 - \prod_{i=k+1}^{m} \xi_i^j\right)\right] = \sum_{j=1}^{M_2} \mathbb{P}\left\{\exists i \, (k < i \leq m): \, \left|\boldsymbol{a}_{i,\perp}^\top \boldsymbol{r}_\perp^{(l)}\right| > \sqrt{\frac{2\left(n-1\right)\log n}{n}}\right\}$$

$$\leq \sum_{j=1}^{M_2} \sum_{i=k+1}^{m} \mathbb{P}\left\{\left|\boldsymbol{a}_{i,\perp}^\top \boldsymbol{r}_\perp^{(l)}\right| > \sqrt{\frac{2\left(n-1\right)\log n}{n}}\right\} \leq M_2 m \frac{1}{n^2}.$$

This combined with Markov's inequality gives

$$\sum_{j=1}^{M_2}\left(1 - \prod_{i=k+1}^{m} \xi_i^j\right) \leq \frac{m \log m}{n^2} \cdot M_2$$

with probability $1 - o(1)$. Putting the above inequalities together suggests that with probability $1 - o(1)$, there exist at least

$$\left(1 - \frac{m \log m}{n^2}\right) M_2 \geq \frac{1}{2}\left(1 - \frac{m \log m}{n^2}\right) \exp\left(\frac{1}{25} n\right) \geq \exp\left(\frac{n}{30}\right)$$

vectors in $\mathcal{M}_2$ satisfying $\prod_{l=k+1}^{m} \xi_i^l = 1$. We then choose $\mathcal{M}$ to be the set consisting of all these vectors, which forms a valid collection satisfying the properties of Lemma 8.

Finally, the only remaining step is to establish the claim (107). To start with, consider an $n \times k$ matrix $\boldsymbol{B} := [\boldsymbol{b}_1, \cdots, \boldsymbol{b}_k]$ of i.i.d. standard normal entries, and let $\boldsymbol{u} \sim \mathcal{N}\left(\boldsymbol{0}, \frac{1}{n} \boldsymbol{I}_n\right)$. Conditional on the $\{\boldsymbol{b}_i$'s,

$$\boldsymbol{b}_{\boldsymbol{u}} = \begin{bmatrix} b_{1,\boldsymbol{u}} \\ \vdots \\ b_{k,\boldsymbol{u}} \end{bmatrix} := \begin{bmatrix} \boldsymbol{b}_1^\top \boldsymbol{u} \\ \vdots \\ \boldsymbol{b}_k^\top \boldsymbol{u} \end{bmatrix} \sim \mathcal{N}\left(\boldsymbol{0}, \frac{1}{n} \boldsymbol{B}^\top \boldsymbol{B}\right).$$

For sufficiently large $m$, one has $k = \frac{m}{4 \log m} \leq \frac{1}{4} n$. Using [3, Corollary 5.35] we get

$$\left\|\frac{1}{n} \boldsymbol{B}^\top \boldsymbol{B} - \boldsymbol{I}\right\| \leq 4\sqrt{k/n} \tag{108}$$

with probability $1 - \exp\left(-\Omega(k)\right)$. Thus, for any constant $0 < \epsilon < \frac{1}{2}$, conditional on $\{\boldsymbol{b}_i\}$ and (108) we obtain

$$\mathbb{P}\left\{\bigcap_{i=1}^{k}\left\{|\boldsymbol{b}_i^\top \boldsymbol{u}| \leq \frac{1}{m}\right\}\right\} \geq (2\pi)^{-\frac{k}{2}} \det^{-\frac{1}{2}}\left(\frac{1}{n} \boldsymbol{B}^\top \boldsymbol{B}\right) \int_{\boldsymbol{b}_{\boldsymbol{u}} \in \Upsilon} \exp\left(-\frac{1}{2} \boldsymbol{b}_{\boldsymbol{u}}^\top \left(\frac{1}{n} \boldsymbol{B}^\top \boldsymbol{B}\right)^{-1} \boldsymbol{b}_{\boldsymbol{u}}\right) \mathrm{d}\boldsymbol{b}_{\boldsymbol{u}}$$

$$\geq (2\pi)^{-\frac{k}{2}}\left(1 + 4\sqrt{k/n}\right)^{-\frac{k}{2}} \int_{\boldsymbol{b}_{\boldsymbol{u}} \in \Upsilon} \exp\left(-\frac{1}{2}\left(1 - 4\sqrt{k/n}\right)^{-1} \sum_{i=1}^{k} b_{i,u}^2\right) \mathrm{d}\boldsymbol{b}_{\boldsymbol{u}} \tag{109}$$

$$\geq (2\pi)^{-\frac{k}{2}}\left(1 + 4\sqrt{k/n}\right)^{-\frac{k}{2}}\left(\sqrt{2\pi} m\right)^{-k}, \tag{110}$$

where $\Upsilon := \{\tilde{\boldsymbol{b}} \mid |\tilde{b}_i| \leq m^{-1}, 1 \leq i \leq k\}$ and (109) is a direct consequence from (108).

When it comes to our quantity of interest, the above lower bound (110) indicates that on an event (defined via $\{\boldsymbol{a}_i\}$) of probability approaching 1, we have

$$\mathbb{E}\Big[\sum_{l=1}^{M_1}\prod_{i=1}^{k}\xi_i^l\Big] \;\geq\; M_1\,(2\pi)^{-\frac{k}{2}}\left(1+4\sqrt{k/n}\right)^{-\frac{k}{2}}\left(\sqrt{2\pi}m\right)^{-k}. \tag{111}$$

Since conditional on $\{\boldsymbol{a}_i\}$, $\prod_{i=1}^{k}\xi_i^l$ are independent across $l$, applying the Chernoff-type bound [5, Theorem 4.5] gives

$$\sum_{l=1}^{M_1}\prod_{i=1}^{k}\xi_i^l \geq \frac{M_1}{2}\,(2\pi)^{-\frac{k}{2}}\left(1+4\sqrt{k/n}\right)^{-\frac{k}{2}}\left(\sqrt{2\pi}m\right)^{-k}$$

with probability exceeding $1-\exp\left(-\frac{1}{8}\frac{M_1}{(2\pi)^{k/2}(1+4\sqrt{k/n})^{k/2}}\left(\frac{1}{\sqrt{2\pi}m}\right)^k\right)$. This concludes the proof.

## B.2   Proof of Lemma 9

Before proceeding, we introduce the $\chi^2$-divergence between two probability measures $P$ and $Q$ as

$$\chi^2\left(P\|Q\right) := \int\left(\frac{\mathrm{d}P}{\mathrm{d}Q}\right)^2\mathrm{d}Q - 1. \tag{112}$$

It is well known (e.g. [4, Lemma 2.7]) that

$$\mathsf{KL}\left(P\|Q\right) \leq \log(1+\chi^2\left(P\|Q\right)), \tag{113}$$

and hence it suffices to develop an upper bound on the $\chi^2$ divergence.

Under independence, for any $\boldsymbol{w}_0, \boldsymbol{w}_1 \in \mathbb{R}^n$, the decoupling identity of the $\chi^2$ divergence [4, Page 96] gives

$$\begin{aligned}
\chi^2\left(\mathbb{P}\left(\boldsymbol{y}\mid\boldsymbol{w}_1\right)\,\|\,\mathbb{P}\left(\boldsymbol{y}\mid\boldsymbol{w}_0\right)\right) &= \prod_{i=1}^{m}\left(1+\chi^2\left(\mathbb{P}\left(y_i\mid\boldsymbol{w}_1\right)\,\|\,\mathbb{P}\left(y_i\mid\boldsymbol{w}_0\right)\right)\right) - 1 \\
&= \exp\left(\sum_{i=1}^{m}\frac{\left(|\boldsymbol{a}_i^\top\boldsymbol{w}_1|^2 - |\boldsymbol{a}_i^\top\boldsymbol{w}_0|^2\right)^2}{|\boldsymbol{a}_i^\top\boldsymbol{w}_0|^2}\right) - 1.
\end{aligned} \tag{114}$$

The preceding identity (114) arises from the following computation: by definition of $\chi^2(\cdot\|\cdot)$,

$$\chi^2\left(\mathsf{Poisson}\left(\lambda_1\right)\,\|\,\mathsf{Poisson}\left(\lambda_0\right)\right) = \left\{\sum_{k=0}^{\infty}\frac{\left(\lambda_1^k\exp\left(-\lambda_1\right)\right)^2}{\lambda_0^k\exp\left(-\lambda_0\right)k!}\right\} - 1$$

$$= \exp\left(\lambda_0 - 2\lambda_1 + \frac{\lambda_1^2}{\lambda_0}\right)\left\{\sum_{k=0}^{\infty}\frac{\left(\lambda_1^2/\lambda_0\right)^k}{k!}\exp\left(-\frac{\lambda_1^2}{\lambda_0}\right)\right\} - 1 \;=\; \exp\left(\frac{(\lambda_1-\lambda_0)^2}{\lambda_0}\right) - 1.$$

Set $\boldsymbol{r} := \boldsymbol{w}_1 - \boldsymbol{w}_0$. To summarize,

$$\begin{aligned}
\mathsf{KL}\left(\mathbb{P}\left(\boldsymbol{y}\mid\boldsymbol{w}_1\right)\,\|\,\mathbb{P}\left(\boldsymbol{y}\mid\boldsymbol{w}_0\right)\right) &\leq \sum_{i=1}^{m}\frac{\left(|\boldsymbol{a}_i^\top\boldsymbol{w}_1|^2 - |\boldsymbol{a}_i^\top\boldsymbol{w}_0|^2\right)^2}{|\boldsymbol{a}_i^\top\boldsymbol{w}_0|^2} \tag{115} \\
&\leq \sum_{i=1}^{m}\frac{|\boldsymbol{a}_i^\top\boldsymbol{r}|^2\left(2\left|\boldsymbol{a}_i^\top\boldsymbol{w}_0\right| + \left|\boldsymbol{a}_i^\top\boldsymbol{r}\right|\right)^2}{|\boldsymbol{a}_i^\top\boldsymbol{w}_0|^2} \\
&= \sum_{i=1}^{m}|\boldsymbol{a}_i^\top\boldsymbol{r}|^2\left(\frac{8|\boldsymbol{a}_i^\top\boldsymbol{w}_0|^2 + 2|\boldsymbol{a}_i^\top\boldsymbol{r}|^2}{|\boldsymbol{a}_i^\top\boldsymbol{w}_0|^2}\right). \tag{116}
\end{aligned}$$

# C   Initialization via truncated spectral Method

This section demonstrates that the truncated spectral method works when $m \asymp n$, as stated in the proposition below.

**Proposition 3.** *Fix $\delta > 0$ and $\boldsymbol{x} \in \mathbb{R}^n$. Consider the model where $y_i = \boldsymbol{a}_i^\top \boldsymbol{x} + \eta_i$ and $\boldsymbol{a}_i \stackrel{\text{ind.}}{\sim} \mathcal{N}(\boldsymbol{0}, \boldsymbol{I})$. Suppose that $\|\boldsymbol{\eta}\|_\infty \leq \varepsilon \|\boldsymbol{x}\|^2$ for some sufficiently small constant $\varepsilon > 0$. With probability exceeding $1 - \exp(-\Omega(m))$, the solution $\boldsymbol{z}^{(0)}$ returned by the truncated spectral method obeys*

$$\operatorname{dist}(\boldsymbol{z}^{(0)}, \boldsymbol{x}) \leq \delta \|\boldsymbol{x}\|, \tag{117}$$

*provided that $m > c_0 n$ for some constant $c_0 > 0$.*

*Proof.* By homogeneity, it suffices to consider the case where $\|\boldsymbol{x}\| = 1$. Recall from [2, Lemma 3.1] that $\frac{1}{m} \sum_{i=1}^m (\boldsymbol{a}_i^\top \boldsymbol{x})^2 \in [1 \pm \varepsilon] \|\boldsymbol{x}\|^2$. Under the hypothesis $\|\boldsymbol{\eta}\|_\infty \leq \varepsilon \|\boldsymbol{x}\|^2$, one has $\frac{1}{m} \|\boldsymbol{\eta}\|_1 \leq \varepsilon \|\boldsymbol{x}\|^2$, which yields

$$\frac{1}{m} \sum_{l=1}^m y_l = \frac{1}{m} \sum_{l=1}^m \left(\boldsymbol{a}_l^\top \boldsymbol{x}\right)^2 + \frac{1}{m} \sum_{l=1}^m \eta_l \in [1 \pm 2\varepsilon] \|\boldsymbol{x}\|^2$$

with probability $1 - \exp(-\Omega(m))$. This in turn implies that

$$\mathbf{1}_{\left\{|(\boldsymbol{a}_i^\top \boldsymbol{x})^2 + \eta_i| \leq \alpha_y^2 (\frac{1}{m} \sum_l y_l)\right\}} \leq \mathbf{1}_{\left\{|\boldsymbol{a}_i^\top \boldsymbol{x}|^2 \leq \alpha_y^2 (\frac{1}{m} \sum_l y_l) + |\eta_i|\right\}} \leq \mathbf{1}_{\left\{|\boldsymbol{a}_i^\top \boldsymbol{x}|^2 \leq (1 + 2\varepsilon)\alpha_y^2 + \varepsilon\right\}}$$
$$\mathbf{1}_{\left\{|(\boldsymbol{a}_i^\top \boldsymbol{x})^2 + \eta_i| \leq \alpha_y^2 (\frac{1}{m} \sum_l y_l)\right\}} \geq \mathbf{1}_{\left\{|\boldsymbol{a}_i^\top \boldsymbol{x}|^2 \leq \alpha_y^2 (\frac{1}{m} \sum_l y_l) - |\eta_i|\right\}} \geq \mathbf{1}_{\left\{|\boldsymbol{a}_i^\top \boldsymbol{x}|^2 \leq (1 - 2\varepsilon)\alpha_y^2 - \varepsilon\right\}}$$

and, hence,

$$\underbrace{\frac{1}{m} \sum_{i=1}^m \boldsymbol{a}_i \boldsymbol{a}_i^\top \left(\boldsymbol{a}_i^\top \boldsymbol{x}\right)^2 \mathbf{1}_{\left\{|\boldsymbol{a}_i^\top \boldsymbol{x}| \leq \sqrt{(1-2\varepsilon)\alpha_y^2 - \varepsilon}\right\}}}_{:=\boldsymbol{Y}_2} \preceq \boldsymbol{Y} \preceq \underbrace{\frac{1}{m} \sum_{i=1}^m \boldsymbol{a}_i \boldsymbol{a}_i^\top \left(\boldsymbol{a}_i^\top \boldsymbol{x}\right)^2 \mathbf{1}_{\left\{|\boldsymbol{a}_i^\top \boldsymbol{x}| \leq \sqrt{(1+2\varepsilon)\alpha_y^2 + \varepsilon}\right\}}}_{:=\boldsymbol{Y}_1}. \tag{118}$$

Letting $\xi \sim \mathcal{N}(0, 1)$, one can compute

$$\mathbb{E}[\boldsymbol{Y}_1] = \beta_1 \boldsymbol{x} \boldsymbol{x}^\top + \beta_2 \boldsymbol{I}, \quad \text{and} \quad \mathbb{E}[\boldsymbol{Y}_2] = \beta_3 \boldsymbol{x} \boldsymbol{x}^\top + \beta_4 \boldsymbol{I}, \tag{119}$$

where $\beta_1 := \mathbb{E}\big[\xi^4 \mathbf{1}_{\{|\xi| \leq \sqrt{(1+2\varepsilon)\alpha_y^2 + \varepsilon}\}}\big] - \mathbb{E}\big[\xi^2 \mathbf{1}_{\{|\xi| \leq \sqrt{(1+2\varepsilon)\alpha_y^2 + \varepsilon}\}}\big]$, $\beta_2 := \mathbb{E}\big[\xi^2 \mathbf{1}_{\{|\xi| \leq \sqrt{(1+2\varepsilon)\alpha_y^2 + \varepsilon}\}}\big]$, $\beta_3 := \mathbb{E}\big[\xi^4 \mathbf{1}_{\{|\xi| \leq \sqrt{(1-2\varepsilon)\alpha_y^2 - \varepsilon}\}}\big] - \mathbb{E}\big[\xi^2 \mathbf{1}_{\{|\xi| \leq \sqrt{(1-2\varepsilon)\alpha_y^2 - \varepsilon}\}}\big]$ and $\beta_4 := \mathbb{E}\big[\xi^2 \mathbf{1}_{\{|\xi| \leq \sqrt{(1-2\varepsilon)\alpha_y^2 - \varepsilon}\}}\big]$. Recognizing that $\boldsymbol{a}_i \boldsymbol{a}_i^\top \left(\boldsymbol{a}_i^\top \boldsymbol{x}\right)^2 \mathbf{1}_{\{|\boldsymbol{a}_i^\top \boldsymbol{x})| \leq c\}}$ can be rewritten as $\boldsymbol{b}_i \boldsymbol{b}_i^\top$ for some sub-Gaussian vector $\boldsymbol{b}_i := \boldsymbol{a}_i \left(\boldsymbol{a}_i^\top \boldsymbol{x}\right) \mathbf{1}_{\{|\boldsymbol{a}_i^\top \boldsymbol{x}) | \leq c\}}$, we apply standard results on random matrices with non-isotropic sub-Gaussian rows [3, Equation (5.26)] to deduce

$$\|\boldsymbol{Y}_1 - \mathbb{E}[\boldsymbol{Y}_1]\| \leq \delta, \quad \|\boldsymbol{Y}_2 - \mathbb{E}[\boldsymbol{Y}_2]\| \leq \delta \tag{120}$$

with probability $1 - \exp(-\Omega(m))$, provided that $m/n$ exceeds some large constant. Besides, when $\varepsilon$ is sufficiently small, one further has $\|\mathbb{E}[\boldsymbol{Y}_1] - \mathbb{E}[\boldsymbol{Y}_2]\| \leq \delta$. These taken together with (118) give

$$\|\boldsymbol{Y} - \beta_1 \boldsymbol{x} \boldsymbol{x}^\top - \beta_2 \boldsymbol{I}\| \leq 3\delta. \tag{121}$$

Fix $\tilde{\delta} > 0$. With (121) in place, repeating the same proof arguments as in [1, Section 7.8] (which we omit in the current paper) and taking $\delta, \varepsilon$ to be sufficiently small, we obtain

$$\operatorname{dist}(\boldsymbol{z}^{(0)}, \boldsymbol{x}) \leq \tilde{\delta} \tag{122}$$

as long as $m/n$ is sufficiently large, as claimed.

$\square$

We now justify that the Poisson model (3) satisfies the condition $\|\boldsymbol{\eta}\| \leq \varepsilon \|\boldsymbol{x}\|^2$ whenever $\|\boldsymbol{x}\| \geq \log^{1.5} m$. Suppose that $\mu_i = (\boldsymbol{a}_i^\top \boldsymbol{x})^2$ and hence $y_i \sim \mathsf{Poisson}(\mu_i)$. It follows from the Chernoff bound that

$$\mathbb{P}(y_i - \mu_i \geq \tau) \leq \frac{\mathbb{E}[e^{ty_i}]}{\exp(t(\mu_i + \tau))} = \frac{\exp(\mu_i(e^t - 1))}{\exp(t(\mu_i + \tau))} = \exp\left(\mu_i(e^t - t - 1) - t\tau\right), \quad \forall t \geq 0.$$

Taking $\tau = 2\tilde{\varepsilon}\mu_i$ and $t = \tilde{\varepsilon}$ for any $0 \leq \tilde{\varepsilon} \leq 1$ gives

$$\mathbb{P}\left(y_i - \mu_i \geq 2\tilde{\varepsilon}\mu_i\right) \quad \leq \quad \exp\left(\mu_i\left(e^t - t - 1 - 2\tilde{\varepsilon}t\right)\right) \overset{(i)}{\leq} \exp\left(\mu_i\left(t^2 - 2\tilde{\varepsilon}t\right)\right) = \exp\left(-\mu_i\tilde{\varepsilon}^2\right),$$

where (i) follows since $e^t \leq 1 + t + t^2$ ($0 \leq t \leq 1$). Letting $\kappa_i = \mu_i/\|\boldsymbol{x}\|^2$ and setting $\tilde{\varepsilon} = \varepsilon/2\kappa_i$, we obtain

$$\mathbb{P}\left(y_i - \mu_i \geq \varepsilon\|\boldsymbol{x}\|^2\right) = \mathbb{P}\left(y_i - \mu_i \geq 2\tilde{\varepsilon}\mu_i\right) \quad \leq \quad \exp\left(-\kappa_i\|\boldsymbol{x}\|^2\tilde{\varepsilon}^2\right) = \exp\left(-\frac{\varepsilon^2\|\boldsymbol{x}\|^2}{4\kappa_i}\right).$$

In addition, standard results on Gaussian measures indicate that $\max_{1 \leq i \leq m} \kappa_i \lesssim \log n$. As a consequence, if $\|\boldsymbol{x}\|^2 \gtrsim \log^3 m$, then $\frac{\|\boldsymbol{x}\|^2}{\kappa_i} \gtrsim \log^2 m$ ($1 \leq i \leq m$), which further gives

$$\mathbb{P}\left(\forall i: \eta_i \geq \varepsilon\|\boldsymbol{x}\|^2\right) = \mathbb{P}\left(\forall i: y_i - \mu_i \geq \varepsilon\|\boldsymbol{x}\|^2\right) \quad \leq \quad m\exp\left(-\Omega\left(\varepsilon^2\log^2 m\right)\right)$$

from the union bound. Similarly, applying the same argument on $-y_i$ we get $\eta_i \geq -\varepsilon\|\boldsymbol{x}\|^2$ for all $i$, which together with (123) establish that

$$\|\boldsymbol{\eta}\|_\infty \leq \varepsilon\|\boldsymbol{x}\|^2 \tag{123}$$

with high probability. In conclusion, the claim (117) applies to the Poisson model.

# D    Local error contraction with backtracking line search

In this paper, we also consider a backtracking line search with truncated objective to determine the learning rate. This strategy performs a line search along the descent direction

$$\boldsymbol{p}_t := \frac{1}{m}\nabla\ell_{\mathrm{tr}}(\boldsymbol{z}_t)$$

and determines an appropriate step size that guarantees a sufficient improvement. In contrast to the conventional search strategy that determines the sufficient progress with respect to the true objective function, we propose to evaluate instead a truncated version of the objective function. Specifically, put

$$\widehat{\ell}(\boldsymbol{z}) := \sum_{i \in \widehat{\mathcal{T}}(\boldsymbol{z})} \left\{y_i \log(|\boldsymbol{a}_i^\top \boldsymbol{z}|^2) - |\boldsymbol{a}_i^\top \boldsymbol{z}|^2\right\}, \tag{124}$$

where

$$\widehat{\mathcal{T}}(\boldsymbol{z}) := \left\{i \mid \left|\boldsymbol{a}_i^\top \boldsymbol{z}\right| \geq \alpha_z^{\mathrm{lb}}\|\boldsymbol{z}\| \text{ and } \left|\boldsymbol{a}_i^\top \boldsymbol{p}\right| \leq \alpha_p\|\boldsymbol{p}\|\right\}.$$

Then the backtracking line search proceeds as

1. Start with $\tau = 1$;

2. Repeat $\tau \leftarrow \beta\tau$ until

$$\frac{1}{m}\widehat{\ell}\left(\boldsymbol{z}^{(t)} + \tau\boldsymbol{p}^{(t)}\right) \geq \frac{1}{m}\widehat{\ell}\left(\boldsymbol{z}^{(t)}\right) + \frac{1}{2}\tau\left\|\boldsymbol{p}^{(t)}\right\|^2, \tag{125}$$

   where $\beta \in (0,1)$ is some pre-determined constant;

3. Set $\mu_t = \tau$.

By definition (124), evaluating $\widehat{\ell}(\boldsymbol{z}^{(t)} + \tau\boldsymbol{p}^{(t)})$ mainly consists in calculating the matrix-vector product $\boldsymbol{A}(\boldsymbol{z}^{(t)} + \tau\boldsymbol{p}^{(t)})$. In total, we are going to evaluate $\widehat{\ell}(\boldsymbol{z}^{(t)} + \tau\boldsymbol{p}^{(t)})$ for $\mathcal{O}\left(\log 1/\beta\right)$ different $\tau$'s, and hence the total cost amounts to computing $\boldsymbol{A}\boldsymbol{z}^{(t)}$, $\boldsymbol{A}\boldsymbol{p}^{(t)}$ as well as $\mathcal{O}(m\log 1/\beta)$ additional flops. Note that the matrix-vector products $\boldsymbol{A}\boldsymbol{z}^{(t)}$ and $\boldsymbol{A}\boldsymbol{p}^{(t)}$ need to be computed even when one adopts a pre-determined step size. Hence, the extra cost incurred by a backtracking line search, which is $\mathcal{O}(m\log 1/\beta)$ flops, is negligible compared to that of computing the gradient even once.

In this section, we verify the effectiveness of a backtracking line search strategy by showing local error contraction. To keep it concise, we only sketch the proof for the noiseless case, but the proof extends to the noisy case without much difficulty. Also we do not strive to obtain an optimized constant. For concreteness, we prove the following proposition.

**Proposition 4.** *The claim in Proposition 1 continues to hold if $\alpha_h \geq 6$, $\alpha_z^{\mathrm{ub}} \geq 5$, $\alpha_z^{\mathrm{lb}} \leq 0.1$, $\alpha_p \geq 5$, and*

$$\|\boldsymbol{h}\|/\|\boldsymbol{z}\| \leq \epsilon_{\mathrm{tr}} \tag{126}$$

*for some constant $\epsilon_{\mathrm{tr}} > 0$ independent of $n$ and $m$.*

Note that if $\alpha_h \geq 6$, $\alpha_z^{\mathrm{ub}} \geq 5$ and $\alpha_z^{\mathrm{lb}} \leq 0.1$, then the boundary step size $\mu_0$ given in Proposition 1 satisfies

$$\frac{0.994 - \zeta_1 - \zeta_2 - \sqrt{2/(9\pi)}\alpha_h^{-1}}{2\left(1.02 + 0.665\alpha_h^{-1}\right)} \geq 0.384.$$

Thus, it suffices to show that the step size obtained by a backtracking line search lies within (0,0.384). For notational convenience, we will set

$$\boldsymbol{p} := m^{-1}\nabla\ell_{\mathrm{tr}}\left(\boldsymbol{z}\right) \quad \text{and} \quad \mathcal{E}_3^i := \left\{\left|\boldsymbol{a}_i^\top \boldsymbol{z}\right| \geq \alpha_z^{\mathrm{lb}}\|\boldsymbol{z}\| \text{ and } \left|\boldsymbol{a}_i^\top \boldsymbol{p}\right| \leq \alpha_p\|\boldsymbol{p}\|\right\}$$

throughout the rest of the proof. We also impose the assumption

$$\|\boldsymbol{p}\|/\|\boldsymbol{z}\| \leq \epsilon \tag{127}$$

for some sufficiently small constant $\epsilon > 0$, so that $\left|\boldsymbol{a}_i^\top \boldsymbol{p}\right|/\left|\boldsymbol{a}_i^\top \boldsymbol{z}\right|$ is small for all non-truncated terms. It is self-evident from (52) that in the regime under study, one has

$$\|\boldsymbol{p}\| \geq 2\left\{1.99 - 2\left(\zeta_1 + \zeta_2\right) - \sqrt{8/\pi}(3\alpha_h)^{-1} - o\left(1\right)\right\}\|\boldsymbol{h}\| \geq 3.64\|\boldsymbol{h}\|. \tag{128}$$

To start with, consider three scalars $h$, $b$, and $\delta$. Setting $b_\delta := \frac{(b+\delta)^2 - b^2}{b^2}$, we get

$$(b+h)^2 \log\frac{(b+\delta)^2}{b^2} - (b+\delta)^2 + b^2 = (b+h)^2 \log\left(1 + b_\delta\right) - b^2 b_\delta$$

$$\overset{\text{(i)}}{\leq} (b+h)^2\left\{b_\delta - 0.4875b_\delta^2\right\} - b^2 b_\delta = \left((b+h)^2 - b^2\right)b_\delta - 0.4875\left(b+h\right)^2 b_\delta^2$$

$$= h\delta\left(2 + h/b\right)\left(2 + \delta/b\right) - 0.4875\left(1 + h/b\right)^2|\delta\left(2 + \delta/b\right)|^2$$

$$= 4h\delta + \frac{2h^2\delta}{b} + \frac{2h\delta^2}{b} + \frac{h^2\delta^2}{b^2} - 0.4875\delta^2\left(1 + \frac{h}{b}\right)^2\left(2 + \frac{\delta}{b}\right)^2, \tag{129}$$

where (i) follows from the inequality $\log\left(1 + x\right) \leq x - 0.4875x^2$ for sufficiently small $x$. To further simplify the bound, observe that

$$\delta^2\left(1 + \frac{h}{b}\right)^2\left(2 + \frac{\delta}{b}\right)^2 \geq 4\delta^2\left(1 + \frac{h}{b}\right)^2 + \delta^2\left(1 + \frac{h}{b}\right)^2\frac{4\delta}{b} \quad \text{and} \quad \frac{2h\delta^2}{b} + \frac{h^2\delta^2}{b^2} = \left(\left(1 + \frac{h}{b}\right)^2 - 1\right)\delta^2.$$

Plugging these two identities into (129) yields

$$(129) \quad \leq \quad 4h\delta + \frac{2h^2\delta}{b} - \left(0.95\left(1 + \frac{h}{b}\right)^2 + 1\right)\delta^2 - 0.4875\delta^2\left(1 + \frac{h}{b}\right)^2\frac{4\delta}{b}$$

$$\leq \quad 4h\delta - 1.95\delta^2 + \frac{2h^2|\delta|}{|b|} + \frac{1.9|h|}{|b|}\delta^2 + \frac{1.95|\delta^3|}{|b|}\left(1 + \frac{h}{b}\right)^2.$$

Replacing respectively $b$, $\delta$, and $h$ with $\boldsymbol{a}_i^\top \boldsymbol{z}$, $\tau \boldsymbol{a}_i^\top \boldsymbol{p}$, and $-\boldsymbol{a}_i^\top \boldsymbol{h}$, one sees that the log-likelihood $\ell_i(\boldsymbol{z}) = y_i \log(|\boldsymbol{a}_i^\top \boldsymbol{z}|^2) - |\boldsymbol{a}_i^\top \boldsymbol{z}|^2$ obeys

$$\ell_i(\boldsymbol{z} + \tau \boldsymbol{p}) - \ell_i(\boldsymbol{z}) = y_i \log \frac{\left|\boldsymbol{a}_i^\top (\boldsymbol{z} + \tau \boldsymbol{p})\right|^2}{\left|\boldsymbol{a}_i^\top \boldsymbol{z}\right|^2} - \left|\boldsymbol{a}_i^\top (\boldsymbol{z} + \tau \boldsymbol{p})\right|^2 + \left|\boldsymbol{a}_i^\top \boldsymbol{z}\right|^2$$

$$\leq \underbrace{-4\tau \left(\boldsymbol{a}_i^\top \boldsymbol{h}\right)\left(\boldsymbol{a}_i^\top \boldsymbol{p}\right)}_{:=I_{1,i}} - \underbrace{1.95\tau^2 \left(\boldsymbol{a}_i^\top \boldsymbol{p}\right)^2}_{:=I_{2,i}} + \underbrace{\frac{2\tau \left(\boldsymbol{a}_i^\top \boldsymbol{h}\right)^2 \left|\boldsymbol{a}_i^\top \boldsymbol{p}\right|}{\left|\boldsymbol{a}_i^\top \boldsymbol{z}\right|}}_{:=I_{3,i}} + \underbrace{\frac{1.9\tau^2 \left|\boldsymbol{a}_i^\top \boldsymbol{h}\right|}{\left|\boldsymbol{a}_i^\top \boldsymbol{z}\right|} \left(\boldsymbol{a}_i^\top \boldsymbol{p}\right)^2}_{:=I_{4,i}}$$

$$+ \underbrace{\frac{1.95\tau^3 \left|\boldsymbol{a}_i^\top \boldsymbol{p}\right|^3}{\left|\boldsymbol{a}_i^\top \boldsymbol{z}\right|} \left(1 - \frac{\boldsymbol{a}_i^\top \boldsymbol{h}}{\boldsymbol{a}_i^\top \boldsymbol{z}}\right)^2}_{:=I_{5,i}}.$$

The next step is then to bound each of these terms separately. Most of the following bounds are straightforward consequences from [2, Lemma 3.1] combined with the truncation rule. For the first term, applying the AM-GM inequality we get

$$\frac{1}{m} \sum_{i=1}^m I_{1,i} \mathbf{1}_{\mathcal{E}_3^i} \leq \frac{4\tau}{3.64m} \sum_{i=1}^m \left\{ \frac{3.64^2}{2} \left(\boldsymbol{a}_i^\top \boldsymbol{h}\right)^2 + \frac{1}{2} \left(\boldsymbol{a}_i^\top \boldsymbol{p}\right)^2 \right\} \leq \frac{4\tau (1 + \delta)}{3.64} \left\{ \frac{3.64^2}{2} \|\boldsymbol{h}\|^2 + \frac{1}{2} \|\boldsymbol{p}\|^2 \right\}.$$

Secondly, it follows from Lemma 4 that

$$\frac{1}{m} \sum_{i=1}^m I_{2,i} \mathbf{1}_{\mathcal{E}_3^i} = -1.95\tau^2 \frac{1}{m} \sum_{i=1}^m \left(\boldsymbol{a}_i^\top \boldsymbol{p}\right)^2 \mathbf{1}_{\mathcal{E}_3^i} \leq -1.95 \left(1 - \tilde{\zeta}_1 - \tilde{\zeta}_2\right) \tau^2 \|\boldsymbol{p}\|^2,$$

where $\tilde{\zeta}_1 := \max\{\mathbb{E}[\xi^2 \mathbf{1}_{\{|\xi| \leq \sqrt{1.01}\alpha_z^{\text{lb}}\}}], \mathbb{E}[\mathbf{1}_{\{|\xi| \leq \sqrt{1.01}\alpha_z^{\text{lb}}\}}]\}$ and $\tilde{\zeta}_2 := \mathbb{E}[\xi^2 \mathbf{1}_{\{|\xi| > \sqrt{0.99}\alpha_h\}}]$. The third term is controlled by

$$\frac{1}{m} \sum_{i=1}^m I_{3,i} \mathbf{1}_{\mathcal{E}_3^i} \leq 2\tau \frac{\alpha_p \|\boldsymbol{p}\|}{\alpha_z^{\text{lb}} \|\boldsymbol{z}\|} \left\{ \frac{1}{m} \sum_{i=1}^m \left(\boldsymbol{a}_i^\top \boldsymbol{h}\right)^2 \right\} \lesssim \tau\epsilon \|\boldsymbol{h}\|^2.$$

Fourthly, it arises from the AM-GM inequality that

$$\frac{1}{m} \sum_{i=1}^m I_{4,i} \mathbf{1}_{\mathcal{E}_3^i} \leq \frac{1.9\tau^2 \alpha_p \|\boldsymbol{p}\|}{\alpha_z^{\text{lb}} \|\boldsymbol{z}\|} \frac{1}{m} \sum_{i=1}^m \left|\boldsymbol{a}_i^\top \boldsymbol{h}\right| \left|\boldsymbol{a}_i^\top \boldsymbol{p}\right| \lesssim \epsilon\tau^2 \frac{1}{m} \sum_{i=1}^m \left\{ 2\left|\boldsymbol{a}_i^\top \boldsymbol{h}\right|^2 + \frac{1}{8} \left|\boldsymbol{a}_i^\top \boldsymbol{p}\right|^2 \right\} \lesssim \epsilon\tau^2 \|\boldsymbol{p}\|^2.$$

Finally, the last term is bounded by

$$\frac{1}{m} \sum_{i=1}^m I_{5,i} \mathbf{1}_{\mathcal{E}_3^i} \leq \frac{1}{m} \sum_{i=1}^m \frac{1.95\tau^3 \left|\boldsymbol{a}_i^\top \boldsymbol{p}\right|^3}{\left|\boldsymbol{a}_i^\top \boldsymbol{z}\right|} \left(\frac{\boldsymbol{a}_i^\top \boldsymbol{x}}{\boldsymbol{a}_i^\top \boldsymbol{z}}\right)^2 \leq \frac{1.95\tau^3 \alpha_p^3 \|\boldsymbol{p}\|^3}{(\alpha_z^{\text{lb}})^3 \|\boldsymbol{z}\|^3} \frac{1}{m} \sum_{i=1}^m \left(\boldsymbol{a}_i^\top \boldsymbol{x}\right)^2 \lesssim \tau^3 \epsilon \frac{\|\boldsymbol{x}\|^2}{\|\boldsymbol{z}\|^2} \|\boldsymbol{p}\|^2.$$

Under the hypothesis (128), we can further derive $\frac{1}{m} \sum_{i=1}^m I_{1,i} \mathbf{1}_{\mathcal{E}_3^i} \leq \tau (1.1 + \delta) \|\boldsymbol{p}\|^2$. Putting all the above bounds together yields that the truncated objective function is majorized by

$$\frac{1}{m} \sum_{i=1}^m \left\{ \ell_i(\boldsymbol{z} + \tau \boldsymbol{p}) - \ell_i(\boldsymbol{z}) \right\} \mathbf{1}_{\mathcal{E}_3^i} \leq \frac{1}{m} \sum_{i=1}^m \left(I_{1,i} + I_{2,i} + I_{3,i} + I_{4,i} + I_{5,i}\right) \mathbf{1}_{\mathcal{E}_3^i}$$

$$\leq \tau (1.1 + \delta) \|\boldsymbol{p}\|^2 - 1.95 \left(1 - \tilde{\zeta}_1 - \tilde{\zeta}_2\right) \tau^2 \|\boldsymbol{p}\|^2 + \tau\tilde{\epsilon} \|\boldsymbol{p}\|^2$$

$$= \left\{ \tau (1.1 + \delta) - 1.95 \left(1 - \tilde{\zeta}_1 - \tilde{\zeta}_2\right) \tau^2 + \tau\tilde{\epsilon} \right\} \|\boldsymbol{p}\|^2 \tag{130}$$

for some constant $\tilde{\epsilon} > 0$ that is linear in $\epsilon$.

Note that the backtracking line search seeks a point satisfying $\frac{1}{m} \sum_{i=1}^m \left\{ \ell_i(\boldsymbol{z} + \tau \boldsymbol{p}) - \ell_i(\boldsymbol{z}) \right\} \mathbf{1}_{\mathcal{E}_3^i} \geq \frac{1}{2}\tau \|\boldsymbol{p}\|^2$. Given the above majorization (130), this search criterion is satisfied only if

$$\tau/2 \leq \tau (1.1 + \delta) - 1.95(1 - \tilde{\zeta}_1 - \tilde{\zeta}_2)\tau^2 + \tau\tilde{\epsilon}$$

or, equivalently,

$$\tau \leq \frac{0.6 + \delta + \tilde{\epsilon}}{1.95(1 - \tilde{\zeta}_1 - \tilde{\zeta}_2)} := \tau_{\text{ub}}.$$

Taking $\delta$ and $\tilde{\epsilon}$ to be sufficiently small, we see that $\tau \leq \tau_{\text{ub}} \leq 0.384$, provided that $\alpha_z^{\text{lb}} \leq 0.1$, $\alpha_z^{\text{ub}} \geq 5$, $\alpha_h \geq 6$, and $\alpha_p \geq 5$.

Using very similar arguments, one can also show that $\frac{1}{m} \sum_{i=1}^{m} \{\ell_i(\boldsymbol{z} + \tau\boldsymbol{p}) - \ell_i(\boldsymbol{z})\} \mathbf{1}_{\mathcal{E}_3^i}$ is minorized by a similar quadratic function, which combined with the stopping criterion $\frac{1}{m} \sum_{i=1}^{m} \{\ell_i(\boldsymbol{z} + \tau\boldsymbol{p}) - \ell_i(\boldsymbol{z})\} \mathbf{1}_{\mathcal{E}_3^i} \geq \frac{1}{2}\tau \|\boldsymbol{p}\|^2$ suggests that $\tau$ is bounded away from 0. We omit this part for conciseness.

## Footnotes

[1]Here, 0.1 can be replaced by any positive constant within $(0, 1/2)$.