[Reviews · NeurIPS 2015]

Submitted by Assigned_Reviewer_1

I was asked to provide a "light review" for this paper.

The paper considers the problem of recovering an n-dimensional signal from random quadratic measurements (i.e., the phase retrieval problem). This problem is known to be solvable in principle from about n measurements; SDP relaxations efficiently solve the problem with O(n) measurements but are not scalable to very large problems. One way of obtaining more efficient algorithms is to consider nonconvex optimization. Interesting results have been obtained using gradient methods (Wirtinger flow) and alternating directions methods, but these guarantees require at least O( n log n ) measurements.

The submission gives a nonconvex method which requires only O( n ) measurements. It also converges more rapidly than the Wirtinger flow, and is stable under small noise. The main idea is a truncation trick which prevents a few measurements from overly influencing the direction of the gradients. This is important, since the algorithm is derived from a Poisson noise model and the gradients are not integrable. Truncation is also deployed to improve the spectral initialization step of the algorithm.

The truncation idea is intuitive, but its analysis is nontrivial. It could be useful for improving the analyses of other nonconvex problems. The paper exhibits this possibility with an application to subspace segmentation / mixed linear regression. Since truncation discards samples with undue influence on the local structure of the objective function, it would be interesting to know if it can convey robustness to some fraction of gross measurement errors.

Overall, the paper makes a nice contribution to the literature on phase retrieval and on provable nonconvex methods in general.
Summary: The submission gives a nonconvex method for phase retrieval which requires only O( n ) measurements. It also converges more rapidly than existing methods, and is stable under small noise. The main idea is a truncation trick which prevents a few measurements from overly influencing the direction of the gradients. This trick may be useful in the analysis of other nonconvex methods, especially for functions whose gradient is heavy-tailed.

Submitted by Assigned_Reviewer_2

The paper is proposing a fast iterative algorithm for solving a set of quadratic equations of the form y_i = |(a_i,x)|^2 where the a_i's are chosen as i.i.d. Gaussian vectors. The problem arises in application such as the phase retrieval problem in imaging.

The method relies and improve on a recent paper which applies an iterative Wirtinger flow algorithm. The main novelty of the current paper is a truncation scheme which ignores examples with large magnitude of the gradient and stabilize the variance of the resulting stochastic gradient descent steps.

This, together with

careful analysis, enable the authors to prove convergence guarantees on the method, for both the noiseless and noisy case, and using a number of samples which is optimal up to a constant, thus improving upon the un-truncated version. The paper is also quite clearly written and understandable.

The numerical experiments show good estimation error and running time - but the authors do not compare their method to several other methods (refs. [2,13,15,16]) in terms of accuracy and speed (except an short analysis of the sample complexity of [15]) - therefore, although the author's method is superior in terms of provable guarantees, it is not clear to me that it is superior in practice for the simulations shown.

The author's observation of a factor-4 scaling between the linear and quadratic case is interesting. However, it is not clear how universal it is - does it arise in a specific simulation or is it more general and robust to choices of parameters such as dimension, sample size, noise level etc.? there seem to be no explanation of this phenomena, and it does not carry over to the rigorous results in section 1.4 which uses general constants as far as I understand.

Summary: The paper presents a provable non-convex iterative solver for a random system of quadratic equations. Numerical results show only a slight (constant) increase in computation time and decrease in accuracy compared to the linear case.

Submitted by Assigned_Reviewer_3

Usually non-convex problems do not guarantee convergence to a global optimum. However, based on a recent result by Candes

et. al) /(2015l. ensure that The Wirtinger flow (WF) permits to reach efficient results under random features. This is possible by initialising the iteration using a spectral method, and then applying a recurrence formula that successively

refines the parameter vector $z$. This equation depends on the log likelihood of the gradient of the likelihood function, $\log(z,y_i), where $y_i$ are the elements of the observation vector.

The core of this paper is the introduction of the Truncated Wirtinger Flow, which leads to a linear time algorithm that delivers near optimal solutions. A careful selection of a subset of the observations that do not have strong influence on the spectral estimates is used to initialise z; then an iteration on the gradient descent using at each step different data subsets such that the gradient of the likelihood function is well controlled is performed, until convergence or until a stopping number of iterations. The consequences of this careful choice of the data subsets is that the initialisation is tighter, and that descent directions become more robust than in the classic Wirtinger flow.

Numerical examples are presented, where TWF is compared to a conjugate gradient method to solve the least squares problem. The result is that TWF is only about 4 times slower. Then, this example is backed up by theoretical results. Finally, in Section 2, the TWF is explained in depth, both for initialisation stage and the truncated gradient stage.

To sum up, the paper presents an original and significant contribution, and is well organised and well written. Proofs provided in the

supplementary material are extensive and clear.

Summary: The authors present a technique to solve quadratic systems of equations by random sampling the observations sequentially, and prove convergence to the right solution. The advantage is that under this setting the problem, which in its classic deterministic approach can be casted as a QCQP problem, is combinatorial and in general NP-complete.

The methodology proposed by the authors seeks to recover the exact solution in the noiseless case, and an near optimal one when noise is present) by minimising the non-convex objective function. This is achieved by finding the MLE starting from a regularised form of the standard spectral initialization. The idea is to discard terms bearing too much influence on the initial estimate and search directions of the gradient descent. The authors demonstrate that their approach runs in linear time with the number of variables, and therefore the approach becomes nearly as easy as solving linear systems.

Submitted by Assigned_Reviewer_4

A truncated Wirtinger flow method is proposed for phase retrieval. It is shown that phase retrieval can be done as efficient as solving linear systems using TWF. Exact recovery and stability results of TWF are proved. Numerical results confirm that TWF is very effective for phase retrieval. A technically solid paper, and should be accepted.
Summary: A truncated Wirtinger flow method is proposed for phase retrieval. Exact recovery and stability properties were established.

Submitted by Assigned_Reviewer_5

Summary: The paper is concerned with solving a quadratic system of equations. Such problem, appears in a variety of applications, ranging from physics (phase retrieval), to mixed regression (which is of certain interest to the NIPS audience). In fact, the authors propose a quite general way of minimizing a non-convex objective function relying on truncated versions of its gradient (truncated Wirtinger flow), and this may have other implications as in matrix completion or dictionary learning.

Quality: The paper is a very nice piece of work.

Clarity: The paper is well-written, though the technicalities could not be included for obvious space limitations, but can be found in the supplemntary material (and the arxiv version).

Originality: Phase retrieval via Wirtinger flow (WT) was already propose in [14]. In this paper, the authors propose key algorithmic changes based on a truncated version of WF (TWF) yielding enhanced performance compared WF. TWF operates only upon a subset of data enabling better descent directions, and moves in a more aggressive way in these directions. Under Gaussian design, the approach is shown to provably optimal (in sample complexity, reaching the information theoretic limit) in the noiseless case, stable to noise, and runs in linear-time complexity.

Comments:

1) Figure 5 (a) and (b): in the text, the corresponding experiments are in the noiseless setting. So speaking of Poisson and Gaussian likelihood in the legend is meaningless. Please correct.

2) Figure 5: (b) should be complex Gaussian design.

3) Figure 5 and page 8, line 410-413: the slope should rather be -1/2 because of the square-root on SNR.

4) I found that a clear discussion of the theoretical value of the universal constant c_0 is missing, and should be included and compared to the empirical bounds 5 (real) and 4.5 (complex) claimed in the text.
Summary: The paper proposes a novel way (truncated Wirting flow) for solving a quadratic system of equations. Under Gaussian design, the approach is shown to provably optimal (in sample complexity) in the noiseless case, stable to noise, and runs in linear-time complexity.

Author Feedback
Author rebuttal: We would like to thank the reviewers for their comments and suggestions, and appreciate their positive appraisal of our approach and results. Below we address the major comments, some of which concern interesting extensions of the current work.

Reviewer 3:

A theoretical understanding on c_0 (the ratio of the sample size to the number of unknowns) is certainly very important and interesting. Unfortunately, our current analysis framework does not yield an accurate estimate on c_0. One possible approach is to follow the machinery of approximate message passing as introduced by Bayati and Montanari, which we will leave for future study.

Besides, thanks for pointing out imprecise legends and text in Fig. 5, which we will correct in the revised version.

Reviewer 4:

Numerical comparison with [2,13,15,16]: According to the plot presented by Netrapalli as well as our own experiment, the alternating minimization approach [15] takes around 90 iterations to attain an accuracy of 10^(-6), where each iteration consists in solving one least-square problem. In comparison, the total runtime of our approach is about 4 times that of solving one least-square problem, which indicates the advantage of our algorithm in terms of practical computational cost. Fienup's algorithm [2] and Gerchberg's algorithm [13] are essentially equivalent to the alternating minimization approach [15] without clever initialization, which have been shown to be slower than the version [15] with spectral initialization. In addition, [16] is mainly designed to exploit additional sparsity information about the signal, which is not accounted for in our current approach. We will be happy to discuss these practical comparisons in the revision.

Uiversality of the factor-4 scaling: While our theory does no yield an accurate estimate on this factor, similar phenomena arise in various experiments that we have conducted under Gaussian design and noiseless data, which accounts for various problem sizes (e.g. when the sample size ranges from 6n to 20n). In fact, this factor gain even improves slightly as the sample size increases. We would also like to clarify that this observation is specific to the algorithmic parameters we recommend throughout the paper (i.e. mu=0.2, alpha_lb = 0.3, alpha_ub = alpha_h = 5), and might change when the parameters vary (e.g. the runtime would be different whenever we change the step size mu). We will add more discussion and explanation in order to make sure our statement is precise and clear.

Reviewer 5:

Robustness under gross errors: This is a very interesting extension for future study. One natural approach is to include an extra regularization term in the non-convex objective functional (e.g. an ell-1 norm enforced on the error term) to handle a constant fraction of gross errors. This might also inspire non-convex paradigms for solving robust PCA.

Reviewer 6:

Indeed, our analysis framework is currently developed for the tractable Gaussian design instead of the CDP design. In the CDP model, there is much less randomness that one can exploit, which makes the analysis significantly more complicated. Fortunately, based on all our empirical experiments, the CDP and the Gaussian design exhibit very similar phase transition diagrams. For this reason, we hope that our theory for Gaussian design can still provide some insights for the CDP model.